# P²-DPO: Grounding Hallucination in Perceptual Processing via Calibration Direct Preference Optimization

**Ruipeng Zhang**[1,3]**, Zhihao Li**[1,2,3]**, Haozhang Yuan**[1,2,3]**, C. L. Philip Chen**[1,2,3]**, Tong Zhang**[1,2,3*]

[1] Guangdong Provincial Key Laboratory of Computational AI Models and Cognitive Intelligence, School of Computer Science & Engineering, South China University of Technology

[2] Pazhou Lab, Guangzhou, China

[3] Engineering Research Center of the Ministry of Education on Health Intelligent Perception and Paralleled Digital-Human, Guangzhou, China

{ruipengzhang.zrp, lizhihaohenry}@gmail.com
{philipchen, tony}@scut.edu.cn

## Abstract

Hallucination has recently garnered significant research attention in Large Vision-Language Models (LVLMs). Direct Preference Optimization (DPO) aims to learn directly from the corrected preferences provided by humans, thereby addressing the hallucination issue. Despite its success, this paradigm has yet to specifically target the perceptual bottleneck in attended regions or address insufficient Visual Robustness against image degradation. Furthermore, existing preference pairs are often vision-agnostic and their inherently off-policy nature limits their effectiveness in guiding model learning. To address these challenges, we propose Perceptual Processing Direct Preference Optimization (P²-DPO), a novel training paradigm in which the model generates and learns from its own preference pairs, thereby directly addressing the identified visual bottlenecks while inherently avoiding the issues of vision-agnostic and off-policy data. It introduces: (1) an on-policy preference pairs construction method targeting Focus-and-Enhance perception and Visual Robustness, and (2) a well-designed Calibration Loss to precisely align visual signals with the causal generation of text. Experimental results demonstrate that with a comparable amount of training data and cost, P²-DPO outperforms strong baselines that rely on costly human feedback on benchmarks. Furthermore, evaluations on Attention Region Fidelity (ARF) and image degradation scenarios validate the effectiveness of P²-DPO in addressing perceptual bottleneck in attended regions and improving Visual Robustness against degraded inputs.[1]

## 1 Introduction

The rapid evolution of Large Language Models (LLMs) (Zhao et al., 2023a; Achiam et al., 2023; Touvron et al., 2023; Guo et al., 2025) has catalyzed the development of Large Vision-Language Models (LVLMs) (Li et al., 2023a; Zhu et al., 2023; Liu et al., 2024b), which have demonstrated impressive capabilities in cross-modal understanding. Despite these advances, hallucination remains a critical challenge: the generated text lacks faithful grounding in visual input (Liu et al., 2024a; Guan et al., 2024). Existing research (Xie et al., 2024; Leng et al., 2024) attributes this to a fundamental architectural imbalance where powerful linguistic priors often suppress visual signals, resulting in hallucination. We can broadly categorize the visual sources of hallucination into two distinct classes: failures in *Perception* and *Perceptual Processing*. The former represents a failure that pushes the model to its upper limits, typically stemming from insufficient knowledge

---

[*]Corresponding author.
[1]The code for model training and testing is available at: https://github.com/ZrpChuang/P2-DPO

gained during pre-training or the inherent limitations of the visual encoder. The latter, in contrast, is considered a failure where the model possesses the latent potential to answer correctly without external knowledge, as it has already captured the key visual evidence but still fails in the subsequent processing stage, leading to hallucination (Yang et al., 2023). While Perception has received significant attention, we argue that Perceptual Processing remains an equally critical but underexplored bottleneck for mitigating hallucinations. Specifically, we identify two core manifestations of this Perceptual Processing failure. The first is a *Perceptual Bottleneck* in Attended Regions, where the model accurately localizes its attention but fails to get the right answer (Figure 1, Left). The second is a *Lack of Robustness*, which reveals the model's high sensitivity to image quality. As shown in Figure 1 (Right), even minor noise or image degradation can cause the model to generate a hallucinatory output. A crucial insight from these failure modes is that the model is often on the cusp of a correct answer, failing not in a fundamental failure in *Perception*, but in its final processing step, which makes it an ideal candidate for self-correction.

The predominant approach to mitigating hallucinations in LVLMs is preference optimization (Rafailov et al., 2023; Yu et al., 2024b), which reduces model hallucination using human or synthetic preferences. Existing data sources follow two main approaches: (1) post-hoc textual revisions of model outputs guided by human or AI feedback (Schulman et al., 2017; Yu et al., 2024b; Lee et al.); and (2) synthetic injection of hallucinations to generate contrastive pairs (Zhou et al., 2024). Despite their demonstrated efficacy (Ouyang et al., 2022; Rafailov et al., 2023), we contend that these methods, which

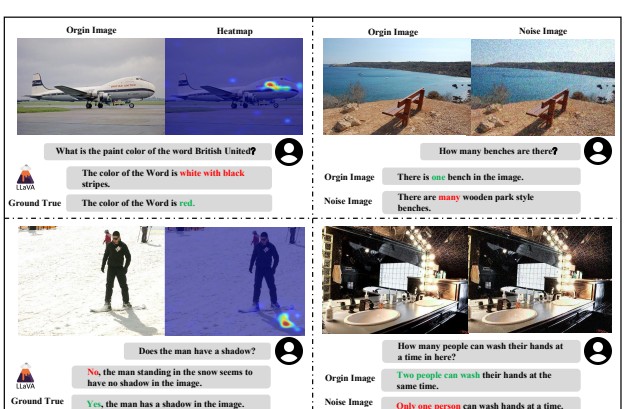

Figure 1: Left: Perceptual Bottleneck. Attention heatmaps visualize that the model correctly focuses on the key entity, yet still hallucinates. Right: Lack of Robustness. The model hallucinates when presented with an image containing noise imperceptible to humans. *All shown answers are generated by LLaVA-1.5-7B.*

we term *Post-hoc Semantic Correction (PSC)*, suffer from a critical limitation: by constructing preference pairs based on textual discrepancies alone, they are inherently **vision-agnostic**. Consequently, the resulting gradients are poorly targeted, failing to teach the model how to resolve the Perceptual Bottleneck or how to address the Lack of Robustness. Moreover, their reliance on external feedback makes the data inherently *off-policy*, which is inefficient for DPO's KL-constrained objective (Azar et al., 2024).

To address the gap, we propose a novel self-correcting method: Perceptual Processing Direct Preference Optimization (P$^2$-DPO). Our core idea is to have the model generate its own on-policy, vision-aware preference data, which provides a direct and targeted gradient signal for its vision-dominant parameters. Specifically, we introduce two new types of preference pairs, each tailored to one of the core deficits: (1) a *Focus-and-Enhance Preference Pair* to resolve the *Perceptual Bottleneck* by contrasting outputs from enhanced versus degraded fine-grained details; and (2) a *Visual Robustness Preference Pair* to boost *Visual Robustness* by contrasting outputs from processing clean versus noisy signals. To further enhance training efficacy, our framework incorporates an *adaptive optimization process*, featuring a Dynamic Deficit-Weighting mechanism and an auxiliary Calibration loss. This results in an efficient learning loop that improves the visual grounding of LVLMs with high precision, obviating the need for costly human feedback.

In summary, our main contributions are as follows.

- We categorize visual causes of hallucination in LVLMs into Perception and Perceptual Processing failures, and identify two overlooked manifestations of the latter. We also critique the existing preference pairs construction methods as vision-agnostic and off-policy.
- We propose Perceptual Processing Direct Preference Optimization (P$^2$-DPO), a self-correction framework that synergistically combines the generation of on-policy, vision-aware preference pairs with a purpose-built optimization strategy, including a dynamic weighting mechanism and a novel Calibration loss, to directly and efficiently rectify perceptual deficits.

- We experimentally show that P²-DPO achieves competitive results on challenging hallucination benchmarks, outperforming costly human-feedback methods without using any such annotations.

## 2  RELATED WORK

**General Capability Enhancement**  A significant body of research aims to mitigate LVLMs hallucination (Yue et al., 2024; Han et al., 2024; Yu et al., 2024a; Huang et al., 2025). One prominent line of work focuses on improving the model's ability to perceive visual evidence by integrating more powerful vision encoders (Lin et al., 2024; Wu & Xie, 2024; Shen et al., 2024), leveraging higher-resolution imagery (Li et al., 2024; Hong et al., 2024), or designing refined attention mechanisms (Zhong et al., 2024; Wu et al., 2024). While these methods can reduce hallucination, they often incur substantial computational costs and engineering overhead due to their architecture-level modifications. More importantly, such architecture-centric solutions are less portable across LVLMs backbones and deployment constraints, and they constitute general-purpose enhancements rather than targeted interventions for the specific visual processing failures we address.

**DPO-based Alignment and Its Data-Centric Limitations**  Direct Preference Optimization (DPO) (Rafailov et al., 2023; An et al., 2023; Zhang et al., 2024) has become the primary paradigm for aligning LVLMs, with multimodal extensions such as mDPO (Wang et al., 2024) and fine-grained data strategies like RLHF-V (Yu et al., 2024b) further advancing the field. However, the efficacy of DPO is fundamentally contingent on the origin and construction of its preference pairs. Current data generation largely falls into two categories: (1) *post-hoc textual revision*, where a model's hallucinatory output $(y^l)$ is corrected by humans or a stronger AI to form the winning response $(y^w)$ (Wang et al., 2023b; Liu et al., 2023; Sun et al., 2024), and (2) *synthetic injection*, where textual errors are deliberately introduced to create contrastive pairs (Zhou et al., 2024).

While these strategies have achieved some success, they are insufficient for resolving the specific visual deficits we target. This insufficiency stems from a core methodological issue: both the DPO loss and its accompanying data construction methods are often inherited from text-only domains without critical adaptation for multimodality. This results in three critical flaws. First, the corrective signal, derived from textual differences, is not targeted at the core visual failures we identify—the Perceptual Bottleneck and insufficient Visual Robustness. Second, the data is inherently *off-policy* due to its reliance on external sources, which is inefficient for DPO's KL-constrained objective. Finally, this dependence on external feedback makes the process prohibitively expensive and difficult to scale.

## 3  PRELIMINARIES: CRITICAL ISSUES IN PREFERENCE DATA

The efficacy of Direct Preference Optimization (DPO) is highly sensitive to the properties of the preference data used for training. Below, we dissect two critical axes of concern that motivate our work: the visual grounding of the data and its relation to the model's own policy.

### 3.1  VISION-AGNOSTIC VS. VISION-GROUNDED DATA

We formalize why Visually-Grounded Contrastive Preference Generation (VCPG) strategy provides a more effective optimization signal for the vision-dominant parameters $\theta_1$ compared to traditional Post-hoc Semantic Correction (PSC) strategies, which are vision-agnostic.

**Gradient of Vision-Dominant Parameters.** We decouple the model parameters $\theta$ into two distinct sets: $\theta_1$, the vision-dominant set, and $\theta_2$, the language-dominant set, as detailed in Appendix A. The gradient for $\theta_1$ in DPO is driven by the difference between the log-likelihood gradients of the winning and losing texts:

$$\Delta(\theta_1) \triangleq \nabla_{\theta_1} \log \pi_\theta(y^w \mid I_{\text{orig}}, P) - \nabla_{\theta_1} \log \pi_\theta(y^l \mid I_{\text{orig}}, P) \tag{1}$$

Thus, the effectiveness of a preference data strategy hinges on its ability to maximize the expected norm of this gradient difference term, $\mathbb{E}[\|\Delta(\theta_1)\|]$.

**Information-Theoretic Analysis via Visual Dependency.** To understand how the properties of preference data influence the gradient norm $\|\Delta(\theta_1)\|$, we introduce the concept of Visual Infor-

mation Dependency (VID): Let $(I, P) \sim \mathcal{D}$, $F_v = g_\theta(I)$, and $Y \sim \pi_\theta(\cdot \mid I, P)$; we define VID $\triangleq I(Y; F_v \mid P, \theta)$. This quantity quantifies how much additional information the visual features $F_v$ provide for generating the text $Y$, beyond what is known from the prompt.

The expected norm $\mathbb{E}[\|\nabla_{\theta_1} \log \pi_\theta(Y \mid I, P)\|]$ is positively associated with VID (motivated by Information Bottleneck views (Alemi et al., 2016)). This proposition reveals a limitation of PSC methods: since the preferred and dispreferred outputs are typically induced by similar visual evidence, we have $I(Y^w; F_v \mid P, \theta) \approx I(Y^l; F_v \mid P, \theta)$, so gradient cancellation occurs, leading to a smaller value for $\|\Delta(\theta_1)\|$. In contrast, VCPG generates pairs with significantly different VIDs, specifically by creating a Visual Information Disparity ($I(Y^w; F_v \mid P, \theta) \gg I(Y^l; F_v \mid P, \theta)$), which maximizes $\|\Delta(\theta_1)\|$, providing a stronger optimization signal.

**Information Geometry Analysis via Fisher Information.** Next, we connect the gradient dynamics to the Fisher Information Matrix (FIM), a key tool for measuring parameter sensitivity. [Information-geometric interpretation (heuristic)] The Fisher information for the vision-dominant parameters $\theta_1$, as measured by $\text{Tr}(F_{\theta_1})$, tends to increase with the KL-divergence between the representation posteriors induced by the preference pair $(y^w, y^l)$:

$$\text{Tr}(F_{\theta_1}) \text{ tends to increase with } \mathbb{E}_\mathcal{D}\left[D_{KL}\big(p_\theta(F_v \mid y^w, P) \,\|\, p_\theta(F_v \mid y^l, P)\big)\right]. \tag{2}$$

This provides a key insight: increasing Fisher information in parameter space corresponds to increasing the separation in the representation space. By generating preference pairs with a large visual disparity between $y^w$ and $y^l$, VCPG tends to push the posteriors $p_\theta(F_v \mid y^w, P)$ and $p_\theta(F_v \mid y^l, P)$ farther apart, thereby increasing the KL-divergence and the induced Fisher information on $\theta_1$. Details and assumptions are provided in Appendix A.

## 3.2 On-Policy vs. Off-Policy Data

Given that DPO ensures the trained policy remains close to the reference policy ($\pi_{\text{ref}}$), we define a response $y$ as on-policy if it has a non-trivial probability under the reference model: $\pi_{\text{ref}}(y \mid I, P) > \epsilon$, where $\epsilon$ is a small positive constant (Yang et al., 2025).

The core of this issue lies in DPO's objective function, which constrains the trained policy $\pi_\theta$ to remain close to the reference policy $\pi_{\text{ref}}$ via a KL-divergence penalty. This introduces a fundamental limitation for strictly off-policy data. If a preferred response $y^w$ lies outside the support of the reference model, denoted as $\text{supp}(\pi_{\text{ref}})$, such that $\pi_{\text{ref}}(y^w \mid I, P) \to 0$, then the KL-divergence term $D_{KL}(\pi_\theta \| \pi_{\text{ref}}) \to \infty$. This effectively confines the optimized policy's support within that of the reference model, i.e., $\text{supp}(\pi_\theta) \subseteq \text{supp}(\pi_{\text{ref}})$, making it mathematically impossible to learn responses that $\pi_{\text{ref}}$ would never generate.

This theoretical limitation manifests as a vanishing gradient during training. The DPO gradient is proportional to a sigmoid weighting factor, $\nabla_\theta \mathcal{L}_{\text{DPO}} \propto \sigma(\hat{r}_l - \hat{r}_w) \cdot \Delta_\nabla$, where $\Delta_\nabla$ is the log-likelihood gradient difference. For a strictly off-policy $y^w$, the implicit reward $\hat{r}_w \triangleq \beta \log(\pi_\theta(y^w \mid I, P)/\pi_{\text{ref}}(y^w \mid I, P))$ diverges to infinity as its denominator approaches zero:

$$\pi_{\text{ref}}(y^w \mid I, P) \to 0 \implies \hat{r}_w \to +\infty$$

Consequently, the argument of the sigmoid function becomes infinitely negative, driving the weighting factor to zero:

$$\hat{r}_l - \hat{r}_w \to -\infty \implies \sigma(\hat{r}_l - \hat{r}_w) \to 0$$

This effectively nullifies the entire gradient update for the most informative, off-policy samples, rendering the learning process highly inefficient. This analysis highlights the critical need for on-policy data generation, a cornerstone of our $P^2$-DPO framework.

## 4 Methodology

Recognizing that Perceptual Processing failures are often "last-mile" errors amenable to self-correction, we introduce $P^2$-DPO, a fully self-driven DPO framework that operates without reliance on external feedback. It couples a mechanism for the autonomous generation of preference pairs with a tailored optimization process. This approach enables the targeted construction of on-policy, vision-aware data, which circumvents the core inefficiencies of conventional, post-hoc preference collection.

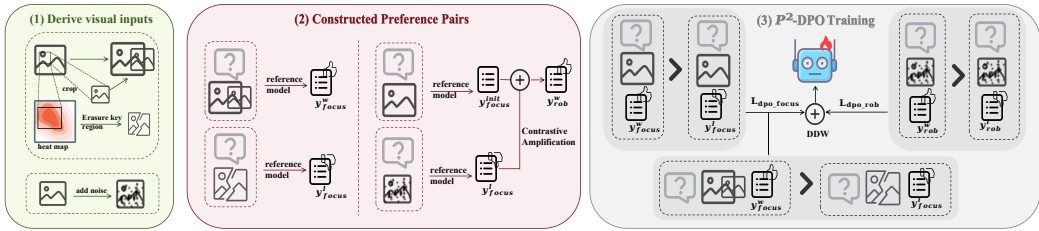

Figure 2: An overview of our proposed P²-DPO framework. The process flows from left to right: (1) Derive Visual Inputs: Based on an initial forward pass to obtain an attention map ($\mathcal{A}$), we create an enhanced input via cropping ($I_{\text{aug}}$), a degraded input via erasure ($I_{\text{deg}}$), and a noisy input ($I_{\text{noise}}$), alongside the original image ($I$). (2) Generate Preference Pairs: The reference model generates two orthogonal preference pairs. (3) P²-DPO Training: Difference losses are dynamically combined.

## 4.1 VISION-AWARE ON-POLICY DATA GENERATION

To address the targeted visual deficits, our approach moves beyond indirect textual revisions by directly intervening on the visual input to generate preference pairs. This causal generation process inherently yields data aligned with our Visually-Grounded Contrastive Preference Generation (VCPG) strategy. Furthermore, because these pairs are derived entirely from the model's own responses without external feedback, the resulting data is on-policy by definition. Our pipeline is designed to generate two distinct types of preference pairs from a single image-prompt instance, a strategy that maximizes data efficiency and obviates the need for human annotation. These generated pairs are then subjected to a rigorous filtering process to ensure their quality.

### 4.1.1 FOCUS-AND-ENHANCE PREFERENCE PAIRS

To overcome the Perceptual Bottleneck, we build upon the crucial observation that a model's attention maps often correctly localize the relevant region even when it is hallucinating. Inspired by previous work (Zhang et al., 2025), which shows that supplementing the input with cropped salient regions can mitigate such errors, we adopt a similar strategy to generate our preference pairs.

Our process begins by using carefully crafted prompts (see Appendix B for details), designed to guide the model's focus towards key visual areas, to perform a forward pass through the reference model $M_{\text{ref}}$. We obtain the final answer-to-image attention maps $\mathcal{A}$ by composing two intermediate, head-averaged attention matrices. Specifically, for both the answer-to-token attention ($\mathcal{A}_{\text{tok}}$) and the token-to-image attention ($\mathcal{A}_{\text{img}}$), we aggregate the scores from all $H$ heads at each layer via averaging, i.e., $\bar{\mathcal{A}} = \frac{1}{H} \sum_{h=1}^{H} \mathcal{A}_h$. The final attention map, quantifying the relevance of each of the $N^2$ input image patches to the answer, is then computed via their tensor product: $\mathcal{A} = \mathcal{A}_{\text{tok}} \otimes \mathcal{A}_{\text{img}} \in \mathbb{R}^{L \times L_c \times 1 \times N^2}$, where $L$ and $L_c$ denote the number of layers in the LLM and the connector, respectively. From these maps, we employ an adaptive cropping algorithm to extract a salient region, $I_{\text{crop}}$, that optimally isolates the most prominent visual entity (see Appendix C for algorithm details). Based on this crop, we derive two distinct visual inputs. The Enhanced Input ($I_{\text{aug}}$) is formed by combining the original image $I$ with its salient crop $I_{\text{crop}}$, denoted as $I_{\text{aug}} = \text{Combine}(I, I_{\text{crop}})$. Conversely, the Degraded Input ($I_{\text{deg}}$) is created by applying a mild erasure within the bounding box of the crop on the original image, denoted as $I_{\text{deg}} = \text{Erase}(I, \text{Bbox}(I_{\text{crop}}))$.

With these two visual contexts established, we generate the preference pair ($y_{\text{focus}}^w, y_{\text{focus}}^l$). The winning response, $y_{\text{focus}}^w = M_{\text{ref}}(I_{\text{aug}}, P_{\text{enh}})$, is generated by conditioning the model on the enhanced input and a meticulously designed prompt template. The losing response, $y_{\text{focus}}^l = M_{\text{ref}}(I_{\text{deg}}, P)$, is generated from the degraded input and the original prompt. This procedure yields a preference pair where the winning response is derived from a state of heightened perceptual clarity, while the losing response stems from a state of targeted visual impairment.

### 4.1.2 VISUAL ROBUSTNESS PREFERENCE PAIRS

We follow a carefully designed three-stage process. The procedure begins with a single prompt $P$ and an original image $I$.

First, we create a degraded version of the image, $I_{\text{noise}} = \text{Noise}(I)$, by perturbing it with Gaussian noise. The losing response, $y_{\text{rob}}^l = M_{\text{ref}}(I_{\text{noise}}, P)$, is then generated by conditioning our reference model $M_{\text{ref}}$ on this corrupted visual input. This response represents the model's output when faced

with low-fidelity signals. Next, an initial, higher-quality response, $y_{\text{rob}}^{\text{init}} = M_{\text{ref}}(I, P)$, is generated by conditioning the model on the original, high-fidelity image $I$.

To refine this initial response into the definitive winning response $y_{\text{rob}}^w$, we introduce *Contrastive Amplification*. This decoding strategy, building on contrastive principles, has been shown to reduce output hallucinations (Leng et al., 2024). The method designates the model conditioned on the original image $I$ as an *expert* (EP) and on the degraded image $I_{\text{noise}}$ as an *amateur* (AT). At each decoding step $t$, it amplifies the difference between their logits, but critically, only within a candidate set $\mathcal{V}_{\text{head}}$ of plausible tokens pre-determined by the expert:

$$y_t \sim \text{softmax}\left((1 + \lambda_{\text{ca}}) \cdot \text{logits}_{\text{EP}}(y_t) - \lambda_{\text{ca}} \cdot \text{logits}_{\text{AT}}(y_t)\right) \tag{3}$$

subject to $y_t \in \mathcal{V}_{\text{head}}(y_{<t})$, where $\lambda_{\text{ca}}$ is a hyperparameter. This process steers generation towards visual fidelity while preserving linguistic coherence.

The output of our data generation pipeline for a single input $(I, P)$ is a set of two orthogonal preference pairs: the Focus-and-Enhance Pair, denoted as $D_{\text{focus}} = (y_{\text{focus}}^w, y_{\text{focus}}^l)$, and the Visual Robustness Pairs, $D_{\text{rob}} = (y_{\text{rob}}^w, y_{\text{rob}}^l)$. To ensure both the quality and utility of this raw dataset, we subsequently apply a principled filtering process. A pair is retained only if it meets a two-fold criterion: (1) the perplexity (PPL) of both responses must fall below a fluency threshold $\tau_{\text{ppl}}$, and (2) its log-probability margin, $M = \log p_{\text{ref}}(y^w) - \log p_{\text{ref}}(y^l)$, must lie within an effective learning range $[\theta_{\text{low}}, \theta_{\text{high}}]$. Further threshold choices are provided in Appendix D.

## 4.2 CALIBRATION DIRECT PREFERENCE OPTIMIZATION

Grounded in the insight that Perceptual Processing failures are on-policy errors requiring probability redistribution, our training objective is a composite DPO loss. It synergistically integrates signals from two orthogonal preference pairs—targeting the Perceptual Bottleneck and Visual Robustness respectively—and unifies them via a dynamic weighting mechanism that adapts to the model's evolving deficits.

### 4.2.1 OBJECTIVE FOR THE PERCEPTUAL BOTTLENECK

To address the Perceptual Bottleneck, we begin with the standard DPO loss on our Focus-and-Enhance pairs ($D_{\text{focus}}$), denoted as $\mathcal{L}_{\text{dpo\_focus}}$:

$$\mathcal{L}_{\text{dpo\_focus}} = -\mathbb{E}_{(y_{\text{focus}}^w, y_{\text{focus}}^l) \sim D_{\text{focus}}}\left[\log \sigma\left(\beta \log \frac{\pi_\theta(y_{\text{focus}}^w|I, P)}{\pi_{\text{ref}}(y_{\text{focus}}^w|I, P)} - \beta \log \frac{\pi_\theta(y_{\text{focus}}^l|I, P)}{\pi_{\text{ref}}(y_{\text{focus}}^l|I, P)}\right)\right] \tag{4}$$

Even though our Focus-and-Enhance pairs provide some visual grounding, the standard DPO loss learns a purely correlational signal, rewarding the preference for $y_{\text{focus}}^w$ over $y_{\text{focus}}^l$ without attributing it to the causal impact of the visual intervention. To instill this causality more robustly, we introduce a complementary Calibration Loss ($\mathcal{L}_{\text{Calib}}$). This objective is derived from a preference model over the Perceptual Confidence Gain, $\Delta_\pi(y) \triangleq \log \pi(y|I_{\text{aug}}) - \log \pi(y|I_{\text{deg}})$. As formally proven in Appendix E, minimizing $\mathcal{L}_{\text{Calib}}$ is equivalent to maximizing $\Delta_{\pi_\theta}(y_{\text{focus}}^w)$, which in turn maximizes the Visual Information Dependency (VID) of the winning response, defined as the mutual information $I(Y_{\text{focus}}^w; F_v^+ \mid P, \theta)$. The resulting loss function is:

$$\mathcal{L}_{\text{Calib}} = -\mathbb{E}_{(y_{\text{focus}}^w, y_{\text{focus}}^l) \sim D_{\text{focus}}}\left[\log \sigma\left(\beta \log \frac{\pi_\theta(y_{\text{focus}}^w|I, I_{\text{crop}}, P)}{\pi_\theta(y_{\text{focus}}^w|I_{\text{deg}}, P)} - \beta \log \frac{\pi_{\text{ref}}(y_{\text{focus}}^l|I, I_{\text{crop}}, P)}{\pi_{\text{ref}}(y_{\text{focus}}^l|I_{\text{deg}}, P)}\right)\right] \tag{5}$$

The complete objective for the Perceptual Bottleneck, $\mathcal{L}_{\text{focus}}$, is a weighted sum of these two components, balanced by a hyperparameter $\lambda_{\text{calib}}$:

$$\mathcal{L}_{\text{focus}} = \mathcal{L}_{\text{dpo\_focus}} + \lambda_{\text{calib}} \cdot \mathcal{L}_{\text{Calib}} \tag{6}$$

### 4.2.2 OBJECTIVE FOR VISUAL ROBUSTNESS

Symmetrically, to tackle the Lack of Robustness, we apply the DPO loss to our Visual Robustness pairs ($D_{\text{rob}}$). This loss, $\mathcal{L}_{\text{dpo\_rob}}$, trains the model to see through noise by favoring the ideal response $y_{\text{rob}}^w$ over the corrupted one $y_{\text{rob}}^l$, even when conditioned on the same noisy input $I_{\text{noise}}$:

$$\mathcal{L}_{\text{dpo\_rob}} = -\mathbb{E}_{(y_{\text{rob}}^w, y_{\text{rob}}^l) \sim D_{\text{rob}}}\left[\log \sigma\left(\beta \log \frac{\pi_\theta(y_{\text{rob}}^w|I_{\text{noise}}, P)}{\pi_{\text{ref}}(y_{\text{rob}}^w|I_{\text{noise}}, P)} - \beta \log \frac{\pi_\theta(y_{\text{rob}}^l|I_{\text{noise}}, P)}{\pi_{\text{ref}}(y_{\text{rob}}^l|I_{\text{noise}}, P)}\right)\right] \tag{7}$$

### 4.2.3 Dynamic Deficit-Weighting

While both preference pairs originate from the same input $(I, P)$, they address different scenarios. Focus-and-Enhance excels at improving local perception, whereas Visual Robustness targets the global robustness, meaning their training signals should be dynamically balanced. To address this, we introduce a novel mechanism called *Dynamic Deficit-Weighting* (DDW). The idea behind DDW is that if the cropped key region has a higher CLIP score with the text compared to the original image, it indicates that the cropped region is highly relevant to the query, and thus its training proportion can be increased. This mechanism first computes a *Perceptual Gain Ratio*, $r$, using a pre-trained CLIP model to diagnose the primary deficit for a given sample, where $r = \frac{\text{CLIPScore}(P, I_{\text{crop}})}{\text{CLIPScore}(P, I)}$. This ratio, which indicates if the bottleneck ($r > 1$) or robustness is the dominant issue, is then mapped to an adjustment factor $\alpha = \alpha_{\max} \cdot \tanh\left(\frac{r-1.0}{\tau}\right)$ that dynamically allocates weights as $w_{\text{focus/robust}} = w_{\text{base}} \pm \alpha$. Our final unified objective, $\mathcal{L}_{\text{total}}$, is the dynamically weighted sum over the mini-batch, enabling tailored corrective pressure for each sample:

$$\mathcal{L}_{\text{total}} = \mathbb{E}\left[w_{\text{focus}} \cdot \mathcal{L}_{\text{focus}} + w_{\text{robust}} \cdot \mathcal{L}_{\text{dpo\_rob}}\right] \tag{8}$$

## 5 Experiments

Our comprehensive evaluation assesses both the effectiveness and the underlying mechanisms of $P^2$-DPO. First, we perform targeted validation of our method's ability to rectify the specific Perceptual Processing deficits we identified. Second, we broaden the scope to a benchmark comparison, evaluating $P^2$-DPO against prominent alignment methods under fair and controlled conditions. Third, we assess data efficiency by examining whether our on-policy, vision-aware data is more sample-efficient for DPO than traditional off-policy sources. Finally, to understand the contributions of each component within our framework, we perform an ablation study to isolate their individual effects.

### 5.1 Experimental Setup

**Base Models.** Our experiments are primarily built upon the open-source **LLaVA-1.5-7B** (Liu et al., 2024b). To demonstrate the generalizability of our method across different model architectures and sizes, we also conduct experiments on the recently released **Qwen2.5-VL-7B/3B** (Bai et al., 2025).

**Data Generation.** Our preference pairs are generated using prompts (image-question) sourced from the **RLHF-V** dataset (Yu et al., 2024b). We **do not** use any of the provided human preference labels, ensuring that our data generation requires no human feedback or annotation.

**Evaluation Benchmarks.** We conduct a comprehensive evaluation on a suite of standard hallucination and trustworthiness benchmarks, strictly adhering to the official evaluation protocol for each.

- **POPE** (Li et al., 2023b): Probes for object-level hallucinations through binary (yes/no) questions about object presence.
- **HallusionBench** (Guan et al., 2024): Evaluates fine-grained discrimination, testing a model's ability to distinguish objects from their contexts and attributes.
- **MMHal-Bench** (Sun et al., 2023): A question-answering benchmark that leverages GPT-4 for nuanced scoring of factual consistency regarding detailed object properties.
- **AMBER** (Wang et al., 2023a): A multi-dimensional benchmark that assesses model trustworthiness by evaluating its generated text against rich, object-level annotations.
- **TextVQA** (Singh et al., 2019): Leveraged for our targeted Perceptual Bottleneck analysis, its ground-truth bounding boxes are essential to quantitatively measuring attention alignment.

### 5.2 Targeted Validation of Perceptual Deficits

We first conduct two targeted experiments to directly validate our method's efficacy in resolving the specific Perceptual Processing failures identified in the introduction.

**Validating Perceptual Bottleneck Mitigation.** To evaluate the Perceptual Bottleneck, we select TextVQA (Singh et al., 2019) for its ground-truth bounding boxes, which facilitate

Table 1: Evaluation on AFR and Processing Accuracy (AFR > 14.0).

| Model | AFR (Avg. ↑) | P-Acc. (% ↑) |
|---|---|---|
| LLaVA-1.5-7B | 14.73 | 66.29 |
| + DPO$_{\text{RLHF}}$ | 15.57 ↑0.84 | 65.71 ↓0.58 |
| **+ P$^2$-DPO** | **18.71** ↑3.98 | **70.10** ↑3.81 |

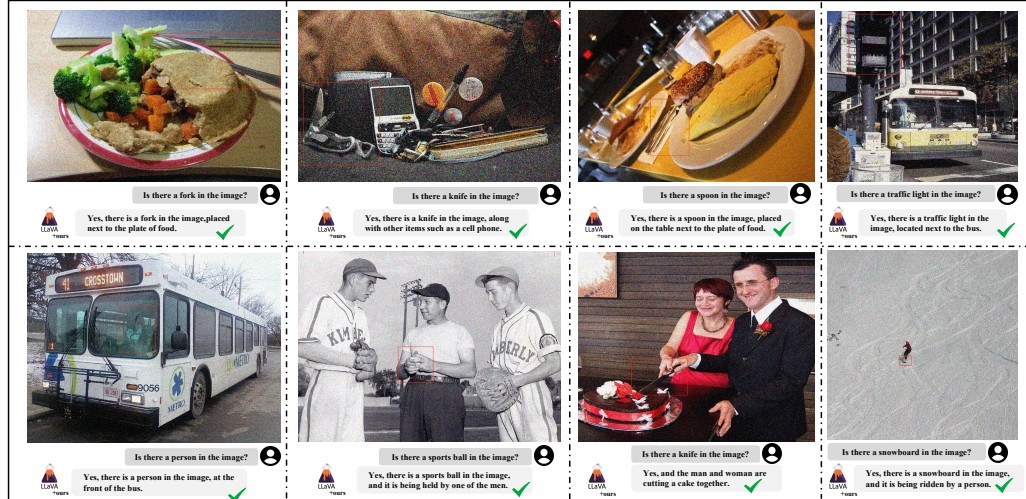

Figure 3: The model's response effect for blurry images($\sigma = 0.20$)

precise quantitative measurement. Inspired by prior work for evaluating visual grounding (Zhang et al., 2025), we use the *Attention Focus Ratio* (AFR)—the ratio of mean attention inside vs. outside the answer's box—to quantify visual *perception*. To isolate subsequent *processing* failures, we then measure *Processing Accuracy* (P-Acc.) in a high-focus subset where the AFR already exceeds our baseline's average (AFR > 14.0). The results in Table 1 reveal baselines exhibit high AFR but poor P-Acc., indicating that attention is accurately localized, but it still gives a wrong answer. In stark contrast, P$^2$-DPO significantly boosts both metrics, effectively bridging this perception-processing gap and confirming it rectifies the bottleneck.

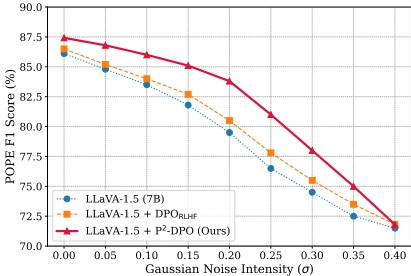

Figure 4: Model accuracy on the POPE benchmark under increasing levels of Gaussian noise ($\sigma$).

**Robustness to Visual Perturbations.** We evaluate Visual Robustness via a systematic stress test on the POPE benchmark (Li et al., 2023b). Gaussian noise with increasing standard deviation ($\sigma$) is added to all images, and the resulting F1 scores are measured. Figure 4 shows that performance degrades with stronger perturbations, but robustness to light noise—common in real-world settings—is especially critical. We compare original LLaVA-1.5-7B, standard DPO on RLHF-V data, and our method. While all models decline under heavy noise, P$^2$-DPO demonstrates significantly higher stability at low-to-moderate noise levels, outperforming LLaVA-1.5-7B by over 4 F1 points at $\sigma = 0.20$. As shown in Figure 3, we find that the model provides correct answers even when visual degradation is difficult for humans to recognize. These results validate our design improves early-stage robustness and mitigates premature degradation under minor perturbations.

## 5.3 BENCHMARK COMPARISON

The key advantage of P$^2$-DPO is its ability to achieve *competitive* performance without relying on costly human or external AI annotations. To ensure fairness, we compare against prior work with similar data scales and compute budgets. As shown in Table 2, our method consistently improves hallucination-sensitive metrics. On LLaVA-1.5-7B, P$^2$-DPO not only achieves top scores across POPE, HallusionBench, and MMHal-Bench, but also demonstrates a particularly dramatic improvement on AMBER's relational reasoning, boosting the $F1_R$ score by a substantial **+8.5 points** over the base model.

We further validate the generalization ability on Qwen2.5-VL-3B, where P$^2$-DPO again shows consistent gains. Notably, it achieves the most significant improvement on the challenging MMHal-Bench, reducing the hallucination rate by **-3.13%** while simultaneously boosting the AMBER relational reasoning score to **80.9 (+3.0)**. These results demonstrate three key advantages: (i) consistent gains on hallucination-sensitive benchmarks, (ii) stable improvements across different architectures,

Table 2: Evaluation of P²-DPO against prominent alignment methods on core hallucination benchmarks. Baselines include the training-free VCD (Leng et al., 2024), the AI-feedback-driven HA-DPO (Zhao et al., 2023c), and methods trained on human preferences (DPO$_{RLHF-V}$, V-DPO$_{RLHF-V}$) from the RLHF-V dataset (Yu et al., 2024b). Our method, using only self-generated data, achieves highly competitive or superior results. **Arrows show deltas vs. the corresponding *Base* in each block:** green = improvement, red = degradation; for ↓ metrics (MMHal Hal, AMBER CHAIRi/Hal), smaller is better so green arrows point downward.

| Model | Pairs Source | POPE (F1)↑ | | | | HallusionBench (Acc)↑ | | | MMHal-Bench | | AMBER | | | | |
|---|---|---|---|---|---|---|---|---|---|---|---|---|---|---|---|
| | | Adv. | Pop. | Rand. | Avg. | qAcc | fAcc | aAcc | Hal↓ | Score↑ | CHAIRi↓ | Hal↓ | $F1_E$↑ | $F1_A$↑ | $F1_R$↑ |
| LLaVA-1.5-7B | Base | 81.80 | 84.36 | 89.12 | 85.10 | 13.90 | 20.23 | 48.16 | 0.62 | 1.97 | 7.8 | 36.4 | 83.2 | 64.1 | 62.4 |
| + VCD | None | 81.33 | 85.06 | 87.16 | 84.51 | — | — | — | 0.58 | 2.12 | — | — | — | — | — |
| + HA-DPO | AI | 82.54 | 87.81 | **90.25** | 86.86 | — | — | — | 0.60 | 1.97 | 6.7 | 30.9 | 88.1 | 66.1 | 68.8 |
| + V-DPO$_{RLHF-V}$ | Human | 84.05 | 87.91 | 89.90 | 87.28 | 17.36 | 19.94 | 51.63 | 0.56 | 2.16 | **5.6** | 27.3 | 91.5 | 73.7 | 64.1 |
| + V-DPO$_{SAD}$ | AI | 83.77 | 87.62 | 89.57 | 86.98 | 22.20 | 21.10 | 55.31 | **0.53** | 2.36 | 6.6 | 30.5 | **95.2** | **76.1** | 61.1 |
| + DPO$_{RLHF-V}$ | Human | 84.03 | 85.94 | 89.12 | 86.38 | 16.70 | 20.81 | 51.31 | 0.60 | 2.08 | 6.4 | 34.5 | 90.7 | 72.6 | 64.6 |
| + P²-DPO (Ours) | Self | **84.53** ↑2.73 | **87.91** ↑3.55 | 89.87 ↑0.75 | **87.44** ↑2.34 | **26.37** ↑12.47 | **30.05** ↑9.82 | **55.62** ↑7.46 | 0.56 ↓0.06 | **2.43** ↑0.46 | 5.9 ↓1.9 | **26.7** ↓9.7 | 92.6 ↑9.4 | 71.7 ↑7.6 | **70.9** ↑8.5 |
| Qwen2.5-VL-3B | Base | 86.26 | 88.05 | 89.79 | 88.03 | 28.13 | 34.10 | 58.10 | 0.42 | 3.38 | 6.8 | 39.1 | **93.2** | 83.7 | 77.9 |
| + DPO$_{RLHF-V}$ | Human | 86.41 | **88.59** | 90.40 | 88.46 | 29.89 | 33.53 | **59.08** | 0.40 | 3.41 | 6.8 | 39.5 | 93.0 | 83.5 | 78.1 |
| + P²-DPO (Ours) | Self | **86.88** ↑0.62 | 88.48 ↑0.43 | **91.50** ↑1.71 | **88.95** ↑0.92 | **30.11** ↑1.98 | **34.68** ↑0.58 | 58.64 ↑0.54 | **0.39** ↓0.03 | **3.49** ↑0.11 | **6.6** ↓0.2 | **38.2** ↓0.9 | 92.7 ↓0.5 | **84.2** ↑0.5 | **80.9** ↑3.0 |
| Qwen2.5-VL-7B | Base | 84.92 | 86.78 | 89.01 | 86.90 | 31.87 | 37.57 | 61.03 | 0.349 | 3.47 | 5.4 | 29.0 | 97.3 | **83.8** | 72.6 |
| + DPO$_{RLHF-V}$ | Human | 84.99 | 86.81 | 89.72 | 87.17 | 34.95 | 39.60 | 62.62 | 0.345 | 3.63 | 4.2 | 29.3 | **98.5** | 82.5 | 72.2 |
| + P²-DPO (Ours) | Self | **86.79** ↑1.87 | **87.79** ↑1.01 | **90.05** ↑1.04 | **88.21** ↑1.31 | **36.70** ↑4.83 | **41.62** ↑4.05 | **65.19** ↑4.16 | **0.32** ↓0.03 | **3.94** ↑0.47 | **3.9** ↓1.5 | **24.9** ↓4.1 | 98.5 ↑1.2 | 83.4 ↑0.4 | **72.9** ↑0.3 |

and (iii) annotation efficiency, since all benefits are obtained without external supervision. We also evaluate Qwen2.5-VL-7B and observe similar trends. Taken together, they confirm that by enhancing *Perceptual Processing*, P²-DPO enables models to better exploit inherent capabilities and deliver robust improvements at no additional annotation cost.

## 5.4 ANALYSIS OF ON-POLICY DATA

As argued in the preliminaries, preference data from off-policy sources (e.g., human feedback) creates a significant policy gap, which is challenging for DPO's KL-constrained objective to bridge. To verify this, we employ the Implicit Preference Strength (IPS) metric (Zhao et al., 2023b) to measure the policy divergence between various preference pairs and the base model. A large negative IPS signifies a sub-

Table 3: IPS analysis reveals a severe policy gap in off-policy data, contrasted with the strong alignment of our on-policy methods.

| Metric | Off-Policy (RLHF-V) | Focus-Enhance | Visual Robust. |
|---|---|---|---|
| Avg. IPS | -58.52 | 12.35 | 45.58 |
| Std. Dev. | 147.08 | 25.61 | 60.56 |
| Neg. Ratio | 68.3% | 8.5% | 44.2% |

stantial policy gap, predicting a difficult and inefficient learning process for DPO. Table 3 provides a stark contrast. The standard **Off-Policy** data exhibits a severely negative average IPS of **-58.52** with a **68.3%** negative ratio, quantifying the substantial learning difficulty it imposes. In contrast, our on-policy data with positive IPS and low variance is not particularly difficult to learn and offers a stable learning signal.

## 5.5 ABLATION STUDY

We evaluate our two-part design—data construction, consisting of Focus-and-Enhance pairs (FEPs) and Visual Robustness pairs (VRPs), and training strategy, including the calibration loss ($\mathcal{L}_{Calib}$) and Dynamic Deficit-Weighting (DDW)—across four benchmarks (Table 4). Training only with VRPs under standard DPO leads to a performance drop, and similarly, training only with FEPs also degrades results, indicating that both data types are necessary and complementary. On the optimiza-

Table 4: Ablation study of our method's components. We report performance on key hallucination and reasoning benchmarks. Each component contributes positively to the final performance.

| Configuration | POPE F1↑ | AMBER Hal↓ | MMHal Score↑ | HallusionBench qAcc↑ | fAcc↑ | aAcc↑ |
|---|---|---|---|---|---|---|
| P²-DPO | **87.42** | **29.27** | **2.43** | **26.37** | **30.05** | **55.62** |
| *Ablating Pairs:* | | | | | | |
| w/o FEPs | 85.84 | 34.70 | 2.30 | 23.56 | 26.62 | 51.10 |
| w/o VRPs | 85.27 | 33.60 | 2.26 | 21.02 | 25.71 | 52.10 |
| *Ablating Strategy:* | | | | | | |
| w/o $\mathcal{L}_{Calib}$ | 86.17 | 33.10 | 2.33 | 23.56 | 27.75 | 53.41 |
| w/o DDW | 86.68 | 31.20 | 2.39 | 23.96 | 28.61 | 52.70 |

tion side, discarding $\mathcal{L}_{Calib}$ also leads to overall performance degradation, demonstrating the effectiveness of our loss design, while replacing Dynamic Deficit-Weighting with equal static weighting consistently degrades results, validating that different losses should carry different weights for different types of errors. Overall, each component contributes positively to the final performance.

## 6 CONCLUSION

We identify that LVLMs have a failure in Perceptual Processing—a critical "last-mile" problem. To address this, we propose P²-DPO, a self-correcting framework that generates on-policy, vision-

aware preference data to directly target these failures. Experiments confirm that P$^2$-DPO achieves performance comparable to or exceeding other training paradigms that rely on human annotation, across a range of hallucination benchmarks, while demonstrating significant advantages in learning efficiency and robustness.

## 7 ACKNOWLEDGEMENTS

This work was funded in part by the National Natural Science Foundation of China grant under number 62536004, and in part by the Key-Area Research and Development Program of Guangdong Province under number 2023B0303030001, and in part by the Science and Technology Program of Guangzhou under number 2024A04J6310, and in part by the Fundamental Research Funds for the Central Universities 2025ZYGXZR021.

### STATEMENTS

**LLM Usage Statement**  In adherence to the ICLR 2026 policy, we disclose that Large Language Models (LLMs), including services based on GPT-4, were utilized in the preparation of this manuscript. Their role was primarily that of a writing assistant, grammar correction, and polishing the phrasing of sentences to improve clarity and readability. The core scientific ideas, experimental design, results analysis, and final conclusions were conceived and articulated exclusively by the human authors.

**Ethics Statement**  We have considered the ethical implications of this work. Our research is built upon publicly available datasets and open-source models, and we do not anticipate any direct negative societal impacts. The preference data generated by our P$^2$-DPO framework is derived from the model's own outputs on these public datasets and does not involve any private or sensitive user information. While our work aims to mitigate model hallucination, which is a step towards safer AI, we acknowledge that no method is perfect, and the potential for misuse of Large Vision-Language Models remains a broader community concern. To foster transparency and enable further research into the safety and reliability of LVLMs, We commit to releasing the implementation of our method and the evaluation scripts upon publication.

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

# A  THEORETICAL ANALYSIS: TARGETED PARAMETER OPTIMIZATION BY MAXIMIZING VISUAL INFORMATION DISPARITY

This section aims to formally show that the Visually-Grounded Contrastive Preference Generation (VCPG) strategy more effectively concentrates the optimization pressure of Direct Preference Optimization (DPO) onto the targeted vision-dominant parameters $\theta_1$ than the Post-hoc Semantic Correction (PSC) strategy. We begin with formal definitions of the model and DPO gradients, introduce information-theoretic tools, and then connect the statistical properties of preference pairs to parameter sensitivity measured by the Fisher Information Matrix (FIM).

## A.1  FORMAL DEFINITIONS: MODEL, PARAMETERS, AND DPO GRADIENT

### A.1.1  PARAMETER DECOUPLING

We assume LVLMs exhibit functional specialization: different parameter subsets tend to serve different functions. For tractability, we adopt a coarse two-way partition: a vision-dominant subset that is more responsive to visual evidence and thus more plastic under vision-heavy supervision, and a language-dominant subset that predominantly supports textual reasoning. This specialization is supported by evidence from three complementary perspectives—architecturally, via purpose-built bridging modules such as the Q-Former in BLIP-2 Li et al. (2023a); empirically, via PEFT results (e.g., LLaMA-Adapter Zhang et al. (2023)) suggesting that visual integration can be localized to a sparse parameter subset; and mechanistically, via interpretability analyses that associate visually grounded knowledge with specific neural pathways Fuchi & Takagi (2024). Guided by this evidence, we partition the parameters $\theta$ into two sets:

- $\theta_1$ **(Vision-Dominant Parameter Set):** the subset of parameters whose updates are predominantly driven by visual evidence, i.e., they exhibit larger log-likelihood gradients when conditioning on the image $I$ (and the corresponding fused visual representation $F_v$) varies.
- $\theta_2$ **(Language-Dominant Parameter Set):** the complementary subset whose updates are predominantly driven by linguistic signals, primarily supporting textual understanding and reasoning. These parameters operate over linguistic inputs as well as the fused visual representation $F_v$; while chiefly governing text generation, they may still interact with visual information depending on task demands.

### A.1.2  DPO LOSS AND GRADIENT DERIVATION

For a single training instance comprising an original image $I_{\text{orig}}$, a text prompt $P$, and a preference pair $(y^w, y^l)$, the DPO loss is:

$$\mathcal{L}(\theta) = -\log \sigma \left( \hat{r}_w - \hat{r}_l \right), \tag{9}$$

where the implicit reward is

$$\hat{r}_y = \beta \log \frac{\pi_\theta(y \mid I_{\text{orig}}, P)}{\pi_{\text{ref}}(y \mid I_{\text{orig}}, P)}. \tag{10}$$

Applying the chain rule:

$$\nabla_\theta \mathcal{L}(\theta) = -\frac{\partial \log \sigma(z)}{\partial z}\bigg|_{z=\hat{r}_w-\hat{r}_l} \cdot \nabla_\theta(\hat{r}_w - \hat{r}_l). \tag{11}$$

Using $\frac{d \log \sigma(z)}{dz} = 1 - \sigma(z) = \sigma(-z)$ and noting $\pi_{\text{ref}}$ is fixed, we obtain:

$$\nabla_\theta \mathcal{L}(\theta) = -\sigma(\hat{r}_l - \hat{r}_w) \cdot \beta \left( \nabla_\theta \log \pi_\theta(y^w \mid I_{\text{orig}}, P) - \nabla_\theta \log \pi_\theta(y^l \mid I_{\text{orig}}, P) \right). \tag{12}$$

Let $A = \beta \cdot \sigma(\hat{r}_l - \hat{r}_w) > 0$. Projecting onto the $\theta_1$ subspace yields:

$$\nabla_{\theta_1} \mathcal{L}(\theta) = -A \cdot \left[ \nabla_{\theta_1} \log \pi_\theta(y^w \mid I_{\text{orig}}, P) - \nabla_{\theta_1} \log \pi_\theta(y^l \mid I_{\text{orig}}, P) \right]. \tag{13}$$

Define the $\theta_1$-gradient difference term

$$\Delta(\theta_1) \triangleq \nabla_{\theta_1} \log \pi_\theta(y^w \mid I_{\text{orig}}, P) - \nabla_{\theta_1} \log \pi_\theta(y^l \mid I_{\text{orig}}, P). \tag{14}$$

Thus, the effectiveness of a preference strategy hinges on its ability to increase the magnitude of $\Delta(\theta_1)$ (in expectation over preference pairs).

A.2 INFORMATION-THEORETIC INTERPRETATION OF GRADIENT DYNAMICS

[Visual Information Dependency, VID] Let $(I, P) \sim \mathcal{D}$ be an image–prompt pair drawn from the data distribution, $F_v = g_\theta(I)$ be the fused visual representation, and $Y \sim \pi_\theta(\cdot \mid I, P)$ be the model output. We define the Visual Information Dependency (VID) under parameters $\theta$ as the conditional mutual information

$$\mathrm{VID}(\theta) \triangleq I(Y; F_v \mid P, \theta), \qquad \text{where } (I, P) \sim \mathcal{D}, \ F_v = g_\theta(I), \ Y \sim \pi_\theta(\cdot \mid I, P). \qquad (15)$$

Equivalently,

$$\mathrm{VID}(\theta) = \mathbb{E}\left[\log \frac{p_\theta(Y \mid F_v, P)}{p_\theta(Y \mid P)}\right] \qquad (16)$$

under the induced joint distribution of $(Y, F_v, P)$.

[**Assumption:** Gradient–VID association] Under the induced joint distribution of $(Y, F_v, P)$, the expected norm of the vision-parameter log-likelihood gradient is positively associated with VID:

$$\mathbb{E}[\|\nabla_{\theta_1} \log \pi_\theta(Y \mid I, P)\|] \text{ tends to increase with } \mathrm{VID}(\theta) = I(Y; F_v \mid P, \theta). \qquad (17)$$

**Setting of the two strategies (basic cases).** We define the two preference construction strategies as follows.

- **PSC:** $Y^l \sim \pi_\theta(\cdot \mid I_{\mathrm{orig}}, P)$ is generated from the original image and prompt; $Y^w = \mathrm{Edit}(Y^l)$ is obtained by post-hoc editing without access to additional visual evidence.
- **VCPG:** There exist an enhanced image $I^+$ and a degraded image $I^-$ such that $Y^w \sim \pi_\theta(\cdot \mid I^+, P)$ and $Y^l \sim \pi_\theta(\cdot \mid I^-, P)$. We assume a degradation channel $I^- = \mathcal{D}(I^+)$ (noise/lossy transform), and $I_{\mathrm{orig}}$ is equal or close to $I^+$ (the training image). When optimizing DPO, we form the training tuple $(I_{\mathrm{orig}}, P, Y^w, Y^l)$ and score *both* responses under the same context $(I_{\mathrm{orig}}, P)$ in the DPO loss.

[DPI-based VID ordering]

(a) (**PSC**) Let $F_v = g_\theta(I_{\mathrm{orig}})$. Assume the edit operator is conditionally independent of $F_v$ given $Y^l$, i.e., $F_v \to Y^l \to Y^w$ forms a Markov chain under $(P, \theta)$. Then by the data processing inequality (DPI),

$$I(Y^w; F_v \mid P, \theta) \leq I(Y^l; F_v \mid P, \theta).$$

If the edit is small (token-level perturbation), we assume the VID reduction is bounded:

$$I(Y^l; F_v \mid P, \theta) - I(Y^w; F_v \mid P, \theta) \leq \varepsilon_{\mathrm{VID}}$$

for a small $\varepsilon_{\mathrm{VID}} > 0$.

(b) (**VCPG**) Let $I^- = \mathcal{D}(I^+)$ be a degradation channel and $F_v^\pm = g_\theta(I^\pm)$. Let $Y^+ \sim \pi_\theta(\cdot \mid I^+, P)$ and $Y^- \sim \pi_\theta(\cdot \mid I^-, P)$. We assume degradation reduces visual dependence *in expectation*:

$$I(Y^+; F_v^+ \mid P, \theta) \geq I(Y^-; F_v^- \mid P, \theta).$$

If $I_{\mathrm{orig}} = I^+$ (or is sufficiently close), evaluating the quantities against $I_{\mathrm{orig}}$ preserves this ordering up to a small mismatch $\delta_{\mathrm{img}}$.

[Local smoothness of gradient field] Assume $\nabla_{\theta_1} \log \pi_\theta(y \mid I_{\mathrm{orig}}, P)$ is $L$-Lipschitz in $y$ under a task-appropriate distance $d(\cdot, \cdot)$:

$$\left\| \nabla_{\theta_1} \log \pi_\theta(y_1 \mid I_{\mathrm{orig}}, P) - \nabla_{\theta_1} \log \pi_\theta(y_2 \mid I_{\mathrm{orig}}, P) \right\| \leq L \, d(y_1, y_2).$$

[Gradient-difference behavior under PSC vs. VCPG] Let $g(y) \triangleq \nabla_{\theta_1} \log \pi_\theta(y \mid I_{\mathrm{orig}}, P)$ and define the random gradient-difference

$$\Delta^{(\cdot)}(\theta_1) \triangleq g(Y^w) - g(Y^l),$$

where $(Y^w, Y^l)$ are constructed by the corresponding strategy and then both are scored under $(I_{\mathrm{orig}}, P)$ in $g(\cdot)$. Under the basic cases specified above:

(i) **PSC:** If $Y^w = \text{Edit}(Y^l)$ with small edit distance $d(Y^w, Y^l) \leq \varepsilon$, then by Lemma A.2,

$$\|\Delta^{(\text{PSC})}(\theta_1)\| = \|g(Y^w) - g(Y^l)\| \leq L\varepsilon. \tag{18}$$

Moreover, Lemma A.2(a) implies $I(Y^w; F_v \mid P, \theta)$ is close to $I(Y^l; F_v \mid P, \theta)$ up to $\varepsilon_{\text{VID}}$, and by Assumption A.2, the corresponding expected gradient norms are close (up to a constant factor). Hence PSC yields a small gradient-difference in equation 18.

(ii) **VCPG:** Let $I^+$ be enhanced and $I^-$ degraded with $I^- = \mathcal{D}(I^+)$, and assume $I_{\text{orig}} = I^+$ (or the mismatch is bounded by $\delta_{\text{img}}$). Then by Lemma A.2(b) and Assumption A.2, there exist $\alpha > 0$ and small $\varepsilon \geq 0$ such that

$$\mathbb{E}[\|g(Y^w)\|] \geq \alpha, \qquad \mathbb{E}[\|g(Y^l)\|] \leq \varepsilon, \tag{19}$$

where $\alpha$ tends to increase as the preferred output becomes more visually dependent, and $\varepsilon$ tends to decrease as the degradation strengthens. Therefore, by the triangle inequality,

$$\mathbb{E}\left[\|\Delta^{(\text{VCPG})}(\theta_1)\|\right] = \mathbb{E}\left[\|g(Y^w) - g(Y^l)\|\right] \geq \mathbb{E}[\|g(Y^w)\|] - \mathbb{E}[\|g(Y^l)\|] \geq \alpha - \varepsilon. \tag{20}$$

In the limiting case $\varepsilon \to 0$, we have $\mathbb{E}\|\Delta^{(\text{VCPG})}(\theta_1)\| \to \mathbb{E}\|g(Y^w)\|$, i.e., the gradient-difference magnitude is *significantly amplified* as the degraded sample becomes visually uninformative.

*Proof.* (i) For PSC, the Lipschitz smoothness in Lemma A.2 directly gives equation 18 when the edit distance is small. The DPI statement in Lemma A.2(a) formalizes that post-hoc editing without extra visual evidence cannot increase visual dependence; with small edits, the VID gap is bounded by $\varepsilon_{\text{VID}}$. Assumption A.2 then connects this bounded VID gap to a bounded gap in expected vision-gradient norms.

(ii) For VCPG, Lemma A.2(b) states that degradation reduces visual dependence in expectation. Combined with Assumption A.2, this induces a gap between $\mathbb{E}\|g(Y^w)\|$ and $\mathbb{E}\|g(Y^l)\|$ as in equation 19. Applying the triangle inequality yields the expectation-level lower bound equation 20. QED

**Consequences for DPO updates.** Combining Theorem A.2 with the DPO gradient on $\theta_1$, $\nabla_{\theta_1}\mathcal{L}(\theta) = -A \cdot \Delta(\theta_1)$, we obtain: PSC produces a small update due to equation 18; VCPG yields a larger-magnitude update due to equation 20, thereby *increasing* $\|\Delta(\theta_1)\|$ (in expectation) compared to PSC.

### A.3 LINKING STATISTICAL PROPERTIES OF PREFERENCE PAIRS TO THE FIM

We now connect the above analysis to the Fisher Information Matrix (FIM). For a preference observation $(y^w \succ y^l)$ under $(I_{\text{orig}}, P)$, define the DPO preference likelihood as

$$\mathcal{P}_\theta(y^w \succ y^l \mid I_{\text{orig}}, P) \triangleq \sigma(\hat{r}_w - \hat{r}_l). \tag{21}$$

The score function restricted to $\theta_1$ is

$$s_1(\theta) \triangleq \nabla_{\theta_1} \log \mathcal{P}_\theta(y^w \succ y^l \mid I_{\text{orig}}, P). \tag{22}$$

We are interested in the trace of $F_{\theta_1}$. From the DPO gradient derivation, we have $s_1(\theta) = A \cdot \Delta(\theta_1)$, where $A = \beta\,\sigma(\hat{r}_l - \hat{r}_w)$. Therefore,

$$\text{Tr}(F_{\theta_1}) = \mathbb{E}_{\mathcal{D}}\left[\|s_1(\theta)\|^2\right] = \mathbb{E}_{\mathcal{D}}\left[A^2\,\|\Delta(\theta_1)\|^2\right]. \tag{23}$$

To relate $\text{Tr}(F_{\theta_1})$ to the statistical structure of a preference pair, we consider conditional distributions over the visual representation space induced by the model and data, denoted as $p_\theta(F_v \mid y, P)$.

[Information-geometric interpretation (heuristic)] Within the DPO framework, $\text{Tr}(F_{\theta_1})$ tends to increase when the preference pair induces a larger separation between the representation posteriors:

$$\text{Tr}(F_{\theta_1}) \text{ tends to increase with } \mathbb{E}_{\mathcal{D}}\left[D_{KL}\big(p_\theta(F_v \mid y^w, P) \,\|\, p_\theta(F_v \mid y^l, P)\big)\right]. \tag{24}$$

**Definition of** $p_\theta(F_v \mid y, P)$. Under the induced joint distribution $(I, P) \sim \mathcal{D}$, $F_v = g_\theta(I)$, $Y \sim \pi_\theta(\cdot \mid I, P)$, we use

$$p_\theta(F_v \mid y, P) \propto p_\theta(y \mid F_v, P)\, p_\theta(F_v \mid P)$$

as the posterior over fused visual representations associated with generating $y$ under prompt $P$.

**Heuristic explanation.** The term $\Delta(\theta_1)$ can be interpreted as the parameter-space force that shifts the model toward representation patterns consistent with $y^w$ rather than $y^l$. From an information-geometric viewpoint Amari & Nagaoka (2000), larger KL separation between the two posteriors typically implies a larger required parameter displacement, which manifests as a larger $\|\Delta(\theta_1)\|$ and thus a larger $\mathrm{Tr}(F_{\theta_1})$.

The VCPG strategy, by design, generates preference pairs $(y^w, y^l)$ that are separated by a wide visual-dependency gap. A visually-rich preferred text $y^w$ and a visually-agnostic dispreferred text $y^l$ tend to require substantially different ideal visual features, meaning the posteriors $p_\theta(F_v \mid y^w, P)$ and $p_\theta(F_v \mid y^l, P)$ are typically farther apart. Therefore, VCPG increases the Fisher information in the vision-parameter subspace by enlarging the separation in the representation space, yielding a more targeted and effective optimization.

# B  OPTIMIZING ATTENTION MAPS VIA PROMPT ENGINEERING

The efficacy of our adaptive cropping algorithm is critically dependent on the quality of the answer-to-image attention maps $\mathcal{A}$ produced by the reference model $M_{\mathrm{ref}}$. To elicit concentrated and well-aligned maps, we first run $M_{\mathrm{ref}}$ with carefully crafted prompts that explicitly ask the model to *locate the relevant region before answering*. From this forward pass we extract two intermediate, head-averaged attention tensors: the answer-to-token attention $\mathcal{A}_{\mathrm{tok}}$ and the token-to-image attention $\mathcal{A}_{\mathrm{img}}$. For each module and layer we average over the $H$ heads, $\bar{\mathcal{A}} = \frac{1}{H}\sum_{h=1}^{H}\mathcal{A}_h$, and compose the final answer-to-image map via their tensor product:

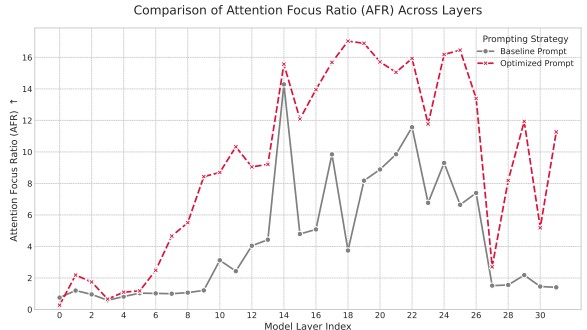

Figure 5: Layer-wise comparison of the Attention Focus Ratio (AFR) between the baseline and our optimized prompt. The optimized prompt (in red) consistently achieves a higher AFR, particularly in the mid-to-high layers (7-26), indicating a substantially improved ability to localize relevant visual information. This enhanced focus provides a high-quality signal for our subsequent cropping algorithm.

$$\mathcal{A} = \mathcal{A}_{\mathrm{tok}} \otimes \mathcal{A}_{\mathrm{img}} \in \mathbb{R}^{L \times L_c \times 1 \times N^2},$$

where $L$ and $L_c$ denote the number of layers in the LLM and the connector, respectively, and $N^2$ is the number of image patches. This $\mathcal{A}$ quantifies the relevance of each of the $N^2$ patches to the predicted answer.

Given $\mathcal{A}$, we derive a saliency map over the $N \times N$ patch grid and feed it to our adaptive cropping procedure to obtain a salient region $I_{\mathrm{crop}}$ (Appendix C). Concretely, a multi-scale sliding window searches over candidate boxes and maximizes a sharpness score—the normalized difference between the summed attention inside a window and that of its adjacent neighborhood—while enforcing image-bound constraints. The selected window is then mapped from patch indices to pixel coordinates to yield a tight bounding box. This crop is subsequently used to construct the Focus-and-Enhance visual contexts introduced in the main text: the enhanced input $I_{\mathrm{aug}} = \mathrm{Combine}(I, I_{\mathrm{crop}})$ and the degraded input $I_{\mathrm{deg}} = \mathrm{Erase}(I, \mathrm{Bbox}(I_{\mathrm{crop}}))$, which together form vision-grounded preference pairs for P$^2$-DPO.

## B.1  EXPERIMENTAL SETUP

**Dataset and Metric.** To quantitatively evaluate attention localization, we use the **TextVQA** (Singh et al., 2019) dataset, which provides ground-truth bounding boxes for textual answers. Our evaluation metric is the **Attention Focus Ratio (AFR)** (Zhang et al., 2025), defined as the ratio of the mean attention intensity inside the ground-truth box to the mean attention intensity outside. A higher AFR score indicates a more accurate and focused localization.

**Prompting Strategies.** We compare two strategies:

- **Baseline Prompt:** The default user-assistant conversation template used by LLaVA (Liu et al., 2024b), where the question is presented directly.

- **Optimized Prompt:** A structured, task-decomposing prompt we designed to explicitly guide the model's focus before answering.

## B.2 RESULTS AND ANALYSIS

Our experiments demonstrate that the optimized prompt significantly enhances the model's ability to focus on relevant visual evidence. The comparative results are visualized in Figure 5, which plots the average AFR score across all 32 layers of the vision-language backbone for both prompting strategies.

The analysis of the plot reveals several key insights:

**Superior Overall Performance:** The "Optimized Prompt" curve (in red) lies substantially above the "Baseline Prompt" curve for the vast majority of layers, indicating a consistent and significant improvement in attention focusing.

**Impact on Deeper Layers:** The most dramatic improvements occur in the mid-to-high layers of the model (approximately layers 7 through 26). These layers are critical for abstract reasoning and semantic understanding. The high AFR in these layers suggests that our prompt is not just affecting low-level visual processing, but is successfully guiding the model's deeper cognitive processes to ground themselves in the correct visual evidence.

**Justification for Cropping:** The substantially higher AFR scores validate our approach. By using the optimized prompt, we can reliably generate high-quality, focused attention maps. These maps serve as an excellent foundation for our adaptive cropping algorithm to accurately isolate the most salient visual entity, which is the cornerstone of our Focus-and-Enhance data generation pipeline.

## B.3 FINAL OPTIMIZED PROMPT TEMPLATE

Based on this quantitative validation, we adopted the following template for generating all attention maps in our pipeline. It modifies the standard LLaVA instruction format to include a "chain-of-thought" style directive.

```
USER: <image>
Your task is to answer the question by first locating the
relevant region in the image.  Question:  {question}
ASSISTANT:
```

Here, {question} is the placeholder for the original question from the dataset. This structure compels the model to prioritize the localization task, resulting in the improved attention focus shown above.

## C    CROPPING ALGORITHM

In this section, we provide the detailed methodology for our adaptive cropping algorithm, which is responsible for extracting the salient region $I_{\text{crop}}$ from a given image $I$ based on the model's internal attention maps $\mathcal{A}$.

## C.1 ALGORITHM MOTIVATION AND OVERVIEW

The primary goal of this algorithm is to identify and isolate the most prominent visual entity that the model is attending to. A simple approach of taking the single point of maximum attention is often too noisy and fails to capture the full extent of an object. Therefore, our algorithm employs a more robust, multi-scale sliding window approach. It evaluates candidate bounding boxes of various sizes, selecting the one that not only contains high attention scores but also exhibits the sharpest contrast with its surrounding regions. This sharpness criterion, measured as a normalized attention difference, is key to ensuring that the crop is tightly focused on a single, salient entity rather than a diffuse area of mild attention. The entire process is detailed in Algorithm 1.

## C.2 KEY STEPS EXPLAINED

---

**Algorithm 1** Adaptive Cropping from Attention Map

---

**Require:** Attention map $\mathcal{A}$, image size $(W, H)$, base box shape $(b_w, b_h)$
**Require:** Scale ratios $R = \{1.0, 1.2, \ldots, 2.0\}$
**Ensure:** Bounding box coordinates $(x_1, y_1, x_2, y_2)$
 1: Initialize $D_{\text{best}} \leftarrow -\infty$, $B_{\text{best}} \leftarrow \text{None}$
 2: **for** each ratio $r \in R$ **do**
 3:     *// Compute crop dimensions*
 4:     $w_r \leftarrow b_w \cdot r; \quad h_r \leftarrow b_h \cdot r$
 5:     $N_w \leftarrow \lfloor w_r / (W/\dim_1(\mathcal{A})) \rfloor$
 6:     $N_h \leftarrow \lfloor h_r / (H/\dim_0(\mathcal{A})) \rfloor$
 7:     *// Sliding window attention sum*
 8:     $\mathcal{A}_{\text{sum}} \leftarrow \text{SlidingWindowSum}(\mathcal{A}, (N_h, N_w))$
 9:     *// Locate attention peak*
10:     $(y_{\max}, x_{\max}) \leftarrow \arg\max(\mathcal{A}_{\text{sum}})$
11:     $a_{\max} \leftarrow \mathcal{A}_{\text{sum}}[y_{\max}, x_{\max}]$
12:     $a_{\text{adj}} \leftarrow \text{MeanAdjacentAttention}(\mathcal{A}_{\text{sum}}, (y_{\max}, x_{\max}))$
13:     *// Score attention sharpness*
14:     $D_r \leftarrow (a_{\max} - a_{\text{adj}})/(N_w \cdot N_h)$
15:     **if** $D_r > D_{\text{best}}$ **then**
16:         $D_{\text{best}} \leftarrow D_r$
17:         $B_{\text{best}} \leftarrow ((y_{\max}, x_{\max}), (N_h, N_w), (h_r, w_r))$
18:     **end if**
19: **end for**
20: *// Decode best crop box*
21: $((y^*, x^*), (N_h^*, N_w^*), (h^*, w^*)) \leftarrow B_{\text{best}}$
22: $x_c \leftarrow x^* \cdot (W/\dim_1(\mathcal{A})) + w^*/2$
23: $y_c \leftarrow y^* \cdot (H/\dim_0(\mathcal{A})) + h^*/2$
24: $x_c \leftarrow \text{clip}(x_c, w^*/2, W - w^*/2)$
25: $y_c \leftarrow \text{clip}(y_c, h^*/2, H - h^*/2)$
26: *// Convert to box coordinates*
27: $x_1 \leftarrow x_c - w^*/2; \quad y_1 \leftarrow y_c - h^*/2$
28: $x_2 \leftarrow x_c + w^*/2; \quad y_2 \leftarrow y_c + h^*/2$
29: **return** $(\text{int}(x_1), \text{int}(y_1), \text{int}(x_2), \text{int}(y_2))$ =0

---

**Lines 2-6:** The core of the algorithm is a loop over several scale ratios. For each ratio, it calculates the target crop dimensions $(w_r, h_r)$ and converts them into the corresponding number of blocks $(N_w, N_h)$ in the attention map. This allows the algorithm to search for objects of different sizes.

**Line 8:** A 2D summed-area table (or a direct sliding window sum) is used to efficiently calculate the total attention within every possible rectangular window of size $(N_h, N_w)$. This creates a new map $\mathcal{A}_{\text{sum}}$ where each point represents the attention sum of a candidate region.

**Lines 14-18:** This is the crucial step for selecting the best scale. Instead of picking the window with the absolute highest attention sum (which would favor larger windows), we calculate a normalized difference score $D_r$ (Line 14). This score measures how much "sharper" the peak attention is compared to its immediate neighbors. By dividing by the area of the window ($N_w \times N_h$), we normalize for size and use the subsequent `if` block (Lines 15-18) to select the scale that produces the most focused, unambiguous attention peak.

**Lines 21-29:** Once the best-performing scale ratio and window position are identified, the algorithm unpacks this information (Line 21) and converts it back into image-space coordinates. It calculates the center of the bounding box (Lines 22-23), clips it to ensure it stays within the image boundaries (Lines 24-25), and finally computes and returns the top-left and bottom-right coordinates (Lines 27-29).

# D    DATA FILTERING IMPLEMENTATION DETAILS

To ensure the quality and utility of our self-generated preference pairs, we applied a principled, data-driven filtering process. The raw dataset, consisting of 5,733 pairs, was analyzed based on two key metrics derived from the reference model ($\pi_{\text{ref}}$): Perplexity (PPL) for fluency, and Log-Probability Margin for learning utility. The goal was to remove outliers and retain approximately 95% of the data, focusing on samples that were both fluent and presented a meaningful learning challenge. The 95% retention target was chosen to balance two competing goals: filtering out low-quality or uninformative samples, while preserving sufficient data diversity and volume for effective training. This range also aligns with standard statistical practices (e.g., 2.5th–97.5th percentiles) for outlier exclusion, ensuring that the retained dataset reflects the core distribution without being skewed by extreme values.

Table 5: Statistical summary of filtering metrics for the 5,733 raw preference pairs. Thresholds were determined based on this distribution to retain approximately 95% of the data.

| Statistic | PPL (Win) | PPL (Lose) | Log-Prob Margin |
|---|---|---|---|
| Mean | 1.76 | 1.79 | 0.017 |
| Std. Dev. | 0.16 | 0.15 | 0.094 |
| Min | 1.34 | 1.24 | -0.294 |
| 25% (Q1) | 1.66 | 1.69 | -0.039 |
| 50% (Median) | 1.74 | 1.77 | 0.015 |
| 75% (Q3) | 1.85 | 1.87 | 0.075 |
| Max | 2.48 | 2.78 | 0.402 |
| **Final Threshold** | $\max(\cdot) < 2.50$ | | $[-0.174, 0.208]$ |

Our filtering process is based on a two-fold criterion applied to each instance:

**1. Fluency Control via Perplexity (PPL).**   Perplexity measures the fluency of a generated response. A high PPL often indicates incoherent or grammatically incorrect text. To remove such outliers, we specified a maximum perplexity threshold, $\tau_{\text{ppl}}$, and discarded any pair where either the winning or losing response exceeded this threshold. Based on our data distribution (see Table 5), where the 99.9th percentile for the maximum PPL was 2.48, we set a conservative threshold to avoid removing valid samples.

- **Criterion:** $\max(\text{PPL}(y^w), \text{PPL}(y^l)) < \tau_{\text{ppl}}$
- **Selected Value:** $\tau_{\text{ppl}} = 2.50$

**2. Learning Utility via Log-Probability Margin.**   The log-probability margin, defined as $M = \log p_{\text{ref}}(y^w) - \log p_{\text{ref}}(y^l)$, quantifies the reference model's preference strength and indicates the learning difficulty of a pair. Pairs with extreme margin values (either very large and positive, or very negative) are less effective for training. To select for samples in the sweet spot of learning, we retained only those pairs whose margin fell within the central 95% of the data distribution. This was achieved by setting the lower and upper bounds, $\theta_{\text{low}}$ and $\theta_{\text{high}}$, to the 2.5th and 97.5th percentiles of the margin distribution, respectively.

- **Criterion:** $\theta_{\text{low}} \leq M \leq \theta_{\text{high}}$
- **Selected Values:** $\theta_{\text{low}} = -0.174$, $\theta_{\text{high}} = 0.208$

## D.1    SUMMARY OF DATA DISTRIBUTION AND FILTERING

Table 5 summarizes the statistical properties of the key metrics in our raw dataset. The final filtering criteria were chosen based on this analysis to be robust and reproducible. After applying both criteria, 5,446 pairs (94.9%) were retained for the final preference pairs.

# E    DERIVATION OF THE CALIBRATION LOSS ($\mathcal{L}_{\text{CALIB}}$)

The standard Direct Preference Optimization (DPO) framework is derived from a preference model over pairs of textual responses $(y^w, y^l)$. Our novel Calibration Loss, $\mathcal{L}_{\text{Calib}}$, extends this principle to a new domain: it models a preference over *perceptual confidence gains*. This appendix provides a first-principles derivation of $\mathcal{L}_{\text{Calib}}$ from a reward modeling perspective, analogous to the derivation of the standard DPO loss.

### E.1 A Reward Model for Perceptual Confidence Gain

First, we define the core quantity of interest: the **Perceptual Confidence Gain**. For a given policy $\pi$ and a response $y$, we define its gain $\Delta_\pi(y)$ as the increase in log-probability when the model is provided with the salient visual crop $I_{\text{crop}}$ versus the degraded context $I_{\text{deg}}$:

$$\Delta_\pi(y) = \log \pi(y|I, I_{\text{crop}}, P) - \log \pi(y|I_{\text{deg}}, P) \tag{25}$$

This value, $\Delta_\pi(y)$, quantifies how much the policy's confidence in response $y$ is *calibrated* by the specific visual evidence in $I_{\text{crop}}$.

Our goal is to train the policy $\pi_\theta$ to achieve a high confidence gain on the *correct* response ($y^w$), while ensuring this gain is meaningfully larger than any spurious gain the model might have on an *incorrect* response ($y^l$). To formalize this, we define a hybrid preference model. We posit that the confidence gain from our policy $\pi_\theta$ on the winning response $y^w$ should be preferable to the confidence gain from the reference policy $\pi_{\text{ref}}$ on the losing response $y^l$.

Using a Bradley-Terry-like model, we can express this preference $p^*$ as:

$$p^*(\Delta_{\pi_\theta}(y^w) \succ \Delta_{\pi_{\text{ref}}}(y^l)) = \sigma(r(\Delta_{\pi_\theta}(y^w)) - r(\Delta_{\pi_{\text{ref}}}(y^l))) \tag{26}$$

where $r(\cdot)$ is an implicit reward function over the confidence gains.

### E.2 Deriving the Loss Objective

For simplicity and directness, we define the reward of a given confidence gain as being directly proportional to the gain itself, scaled by the DPO temperature parameter $\beta$:

$$r(\Delta_\pi(y)) = \beta \cdot \Delta_\pi(y) \tag{27}$$

Substituting this reward definition back into our preference model, we get:

$$p^*(\Delta_{\pi_\theta}(y^w) \succ \Delta_{\pi_{\text{ref}}}(y^l)) = \sigma(\beta \cdot \Delta_{\pi_\theta}(y^w) - \beta \cdot \Delta_{\pi_{\text{ref}}}(y^l))$$

Now, we expand the gain terms $\Delta$ using their definitions:

$$p^*(\dots) = \sigma\left(\beta\left(\log \frac{\pi_\theta(y^w|I, I_{\text{crop}}, P)}{\pi_\theta(y^w|I_{\text{deg}}, P)}\right) - \beta\left(\log \frac{\pi_{\text{ref}}(y^l|I, I_{\text{crop}}, P)}{\pi_{\text{ref}}(y^l|I_{\text{deg}}, P)}\right)\right)$$

The final step is to formulate the training objective using Maximum Likelihood Estimation (MLE). We aim to find the policy $\pi_\theta$ that maximizes the log-likelihood of this preference data. The negative log-likelihood loss for a single preference pair is therefore:

$$\mathcal{L} = -\log p^*(\Delta_{\pi_\theta}(y^w) \succ \Delta_{\pi_{\text{ref}}}(y^l)) \tag{28}$$

This gives us precisely the form of our Calibration Loss, $\mathcal{L}_{\text{Calib}}$, averaged over the dataset $D_{\text{focus}}$:

$$\mathcal{L}_{\text{Calib}} = -\mathbb{E}_{(y^w, y^l) \sim D_{\text{focus}}}\left[\log \sigma\left(\beta \log \frac{\pi_\theta(y^w|I, I_{\text{crop}}, P)}{\pi_\theta(y^w|I_{\text{deg}}, P)} - \beta \log \frac{\pi_{\text{ref}}(y^l|I, I_{\text{crop}}, P)}{\pi_{\text{ref}}(y^l|I_{\text{deg}}, P)}\right)\right] \tag{29}$$

This derivation demonstrates that $\mathcal{L}_{\text{Calib}}$ is not an ad-hoc objective, but a principled loss grounded in a reward model tailored to instill visual causality by maximizing the confidence gain on correct responses, regularized by the baseline gain on incorrect ones.

### E.3 Connection to Visual Information Dependency (VID)

We now analyze how minimizing $\mathcal{L}_{\text{Calib}}$ implicitly maximizes the Visual Information Dependency (VID) of the winning response $y^w$. Recall that VID is defined as the conditional mutual information $I(Y; F_v \mid P, \theta)$, where $Y \sim \pi_\theta(\cdot \mid I, P)$ and $F_v = g_\theta(I)$. The objective of $\mathcal{L}_{\text{Calib}}$ is to maximize the term inside the sigmoid, particularly the perceptual confidence gain for the winning response:

$$\Delta_{\pi_\theta}(y^w) = \log \pi_\theta(y^w|I, I_{\text{crop}}, P) - \log \pi_\theta(y^w|I_{\text{deg}}, P) \tag{30}$$

Let's analyze the two components of this term. The enhanced context $(I, I_{\text{crop}})$ provides a rich visual representation, which we denote as $F_v^+$. Conversely, the degraded context $I_{\text{deg}}$ provides a noisy and

uninformative representation, $F_v^-$. Therefore, maximizing $\Delta_{\pi_\theta}(y^w)$ is equivalent to maximizing the difference in log-likelihoods given these two representations:

$$\max_\theta \left( \log \pi_\theta(y^w|F_v^+) - \log \pi_\theta(y^w|F_v^-) \right) \tag{31}$$

From an information-theoretic viewpoint, maximizing $\log \pi_\theta(y|F_v)$ is closely related to increasing the dependence between the output random variable and the visual representation, i.e., $I(Y; F_v \mid P, \theta)$ (Alemi et al., 2016). Thus, the optimization forces the model to learn a policy $\pi_\theta$ that satisfies two conditions simultaneously:

1. **High Information Utilization:** To maximize $\log \pi_\theta(y^w|F_v^+)$, the model must learn to extract and utilize a large amount of information from the rich visual features $F_v^+$. This directly pushes for a higher conditional mutual information $I(Y; F_v^+ \mid P, \theta)$.
2. **Low Information Reliance on Noise:** To minimize $\log \pi_\theta(y^w|F_v^-)$, the model must learn to *distrust* the noisy features $F_v^-$ and reduce its reliance on them for generating $y^w$.

By jointly optimizing for both conditions, the training process explicitly rewards the model for making its generation of $y^w$ more dependent on the high-quality visual evidence present in $F_v^+$. This directly translates to an increase in Visual Information Dependency (VID), effectively teaching the model to "see" rather than merely "guess".

## F  IMPLEMENTATION DETAILS

This section provides a comprehensive overview of the experimental setup and hyperparameters used to obtain the results presented in the main paper.

### F.1  GENERAL TRAINING CONFIGURATION

All models are fine-tuned using the AdamW optimizer (Loshchilov & Hutter, 2017). We employ a cosine learning rate schedule with a peak learning rate of $2 \times 10^{-5}$, a linear warm-up phase of 100 steps, and a final learning rate annealed to zero. The weight decay is set to 0.1. To ensure computational and memory efficiency, all fine-tuning is performed using Low-Rank Adaptation (LoRA) (Hu et al., 2022). We apply LoRA adapters with a rank of $r = 64$ and an alpha scaling factor of $\alpha = 64$ to all linear projection layers within the attention blocks of the language model. For all Direct Preference Optimization (DPO) based loss components, the temperature parameter $\beta$ is consistently set to 0.1.

All experiments were conducted on a system equipped with 2 NVIDIA A100 (80GB) GPUs. Our implementation is built upon the PyTorch framework and leverages the Hugging Face Transformers and PEFT libraries.

### F.2  $P^2$-DPO SPECIFIC HYPERPARAMETERS

The key hyperparameters unique to our $P^2$-DPO framework are detailed below. These values were selected based on preliminary experiments and held constant across all main results.

- **Calibration Loss Weight ($\lambda_{\mathbf{calib}}$):** The balancing coefficient for the Calibration Loss is set to $\lambda_{\text{calib}} = 0.3$.
- **Contrastive Amplification Coefficient ($\lambda_{\mathbf{ca}}$):** For generating the *Visual Robustness Pairs*, the coefficient for Contrastive Amplification is set to $\lambda_{\text{ca}} = 0.1$.
- **Dynamic Deficit-Weighting (DDW) Parameters:**
  - Base weight ($w_{\text{base}}$): 0.5
  - Maximum adjustment factor ($\alpha_{\text{max}}$): 0.2
  - Temperature ($\tau$): 0.1

## G  ADDITIONAL EXPERIMENTS ON DATA QUALITY

In this section, we present two additional experiments that specifically examine the quality of our constructed training data: (1) an external validation of the PPL + Margin filtering strategy using

Table 6: CLIPScore statistics for attention-guided vs. inverse cropping.

| Group | Metric | Count | Mean | Median | Std | Min | Max |
|---|---|---|---|---|---|---|---|
| att_crop | improvement_ratio | 500 | 1.22 | 1.16 | 0.20 | 1.06 | 3.17 |
| att_crop | improvement_diff | 500 | 3.64 | 3.09 | 1.85 | 1.85 | 12.25 |
| att_inverse | improvement_ratio | 500 | 0.93 | 0.93 | 0.18 | 0.22 | 1.55 |
| att_inverse | improvement_diff | 500 | -1.76 | -1.29 | 3.75 | -15.15 | 8.95 |

human-annotated references, and (2) an analysis of the reasonableness of CLIPScore as a dynamic weighting signal for attention-guided crops.

### G.1 EFFECTIVENESS OF PPL + MARGIN FILTERING

Our preference pairs are constructed from the prompts and images in the RLHF-V dataset. During training, we use only the prompt and image and do *not* use the human preference annotations provided by RLHF-V. To validate the quality of our self-constructed preference pairs and to check whether the PPL + Margin filter truly identifies "problematic pairs", we construct an external evaluation task based on RLHF-V's human annotations.

**Evaluation task with human references.** Each RLHF-V sample contains an image, a question, and two human responses: a high-quality answer $A_{HQ}$ (High-Quality Answer) and a low-quality answer $A_{LQ}$ (Low-Quality Answer). In this experiment, these human annotations are used only as evaluation references and do not participate in training.

For each self-constructed preference pair $(y^w, y^l)$, we construct an evaluation prompt that contains: (i) the original question; (ii) the human high-quality answer as a golden reference; (iii) the human low-quality answer as a negative reference; and (iv) two model responses, denoted as *Response A* and *Response B*, corresponding to $y^w$ and $y^l$, randomly assigned to A/B.

A strong general-purpose language model (Qwen3-Max) is then used as an evaluator to assign a quality score in $\{1, 2, 3\}$ to each of A and B, based on their semantic proximity to the human high-/low-quality answers:

- **3 points**: semantically close to $A_{HQ}$, with only minor descriptive differences and correctly capturing the main idea;
- **2 points**: basically reasonable but not fully aligned, e.g., missing some information or adding harmless verbosity;
- **1 point**: closer to $A_{LQ}$ or clearly wrong / hallucinatory.

**Standard-answer-based invalid-sample criterion.** Within this framework, we label a preference pair as an *invalid sample that should be removed* if either of the following holds:

- the response we mark as the winner, $y^w$, receives the lowest score (1 point), indicating that it is semantically closer to the human low-quality answer;
- $y^w$ receives a strictly lower score than $y^l$, indicating that the evaluator believes that the true preference direction is opposite to our constructed win/lose relation.

This "standard-answer filter" depends only on the question, the human high-/low-quality answers, and our generated responses. It is completely independent of PPL, Margin, or attention maps, and thus serves as an external semantic-level reference.

**Quantitative comparison with PPL + Margin.** According to the above standard-answer filter:

- the Qwen-based evaluator marks **510** preference pairs as invalid samples;
- our PPL + Margin filtering alone removes **287** samples;
- the intersection between these two sets contains **236** samples.

Thus, among the 287 samples removed by PPL + Margin, we have

$$\frac{236}{287} \approx 82\%$$

that are *also* judged invalid by the independent standard-answer filter (Qwen + human high/low-quality answers).

Two observations follow:

- **High overlap indicates that the rules are not arbitrary.** More than 80% of the samples removed by PPL + Margin are simultaneously flagged as problematic by an independent evaluator that explicitly compares against human high-/low-quality answers. This suggests that the PPL + Margin filter is highly consistent with a human-anchored semantic criterion, rather than pruning examples at random.

- **The thresholds are intentionally conservative.** Although the standard-answer filter identifies 510 suspicious samples in total, PPL + Margin removes only a subset of them (287 samples) and retains roughly 95% of the overall data. PPL + Margin therefore acts as a *fallback cleaner for extremely bad samples*, instead of aggressively deleting all borderline cases, and thus achieves a balance between data quality and data diversity.

Overall, this external evaluation confirms that our PPL + Margin filtering strategy reliably identifies and removes a large portion of low-quality or preference-direction-incorrect samples, while remaining conservative in scale.

### G.2 REASONABLENESS OF CLIPSCORE-BASED DYNAMIC WEIGHTING

We further design a controlled experiment to verify the reasonableness of using CLIPScore as a dynamic weighting signal in our data construction pipeline.

**Setup.** We randomly sample 500 examples from the training set and compute CLIPScore between the prompt $P$ and: (i) the original image $I$, and (ii) the attention-guided cropped image $I_{\text{crop}}$. We define

$$\text{diff} = \text{CLIPScore}(P, I_{\text{crop}}) - \text{CLIPScore}(P, I),$$

$$\text{ratio} = \frac{\text{CLIPScore}(P, I_{\text{crop}})}{\text{CLIPScore}(P, I)}.$$

At the same time, we construct a *reverse-cropping* control group: for the same set of examples, we crop from the complementary (low-attention) region of the attention map to obtain $I_{\text{inv}}$, and compute the corresponding $\text{diff}$ and $\text{ratio}$ in the same way. We denote the attention-guided cropping group as `att_crop`, and the reverse-cropping group as `att_inverse`.

**Results.** Table 6 summarizes the statistics, and the full distributions (omitted here for space) show consistent trends.

In the `att_crop` group, the mean of improvement_ratio is about $1.22$, and the median is about $1.16$, indicating that attention-guided cropping systematically improves CLIPScore, with roughly a 20% relative gain on average. The mean of improvement_diff is about $3.64$, and the median is about $3.09$, showing that the semantic similarity between the cropped image and the prompt is consistently increased for most samples.

In contrast, in the `att_inverse` group, the mean of improvement_ratio is about $0.93$, and the median is also about $0.93$, both below 1, meaning that when we deliberately crop from regions not attended by the model, CLIPScore decreases in a systematic way. The mean of improvement_diff is about $-1.76$, and the median is about $-1.29$, exhibiting a clear negative shift.

Although the standard deviations indicate some variation across individual samples, the signs and magnitudes of the means and medians make the overall pattern clear: when cropping along high-response attention regions (`att_crop`), CLIPScore typically *improves*; when cropping from complementary regions (`att_inverse`), CLIPScore typically *degrades*.

**Implications for dynamic weighting.** These results show that CLIPScore is highly sensitive to whether the crop focuses on semantically relevant regions. When the crop aligns with high-response attention regions, most samples satisfy $\text{ratio} > 1$ and $\text{diff} > 0$; when the crop lies in complementary regions, most samples satisfy $\text{ratio} < 1$ and $\text{diff} < 0$. Using $\text{ratio}$ and $\text{diff}$ to drive the dynamic weighting of $w_{\text{focus}}$ and $w_{\text{robust}}$ therefore allows us, in a statistical sense, to amplify the contribution of "well-cropped" samples and suppress the impact of "badly cropped" samples, which is precisely the intended behavior of our weighting mechanism.

## H  ANALYSIS OF ATTENTION REDISTRIBUTION AND FOCUSING

To quantify how training reshapes the balance between textual and visual evidence inside the LVLMs, we analyze (i) global and layer-wise shifts in attention from text tokens to image tokens, and (ii) the evolution of the Attention Focusing Ratio (AFR), which measures how tightly attention concentrates on ground-truth object regions.

### H.1  GLOBAL AND LAYER-WISE SHIFT FROM TEXTUAL TO VISUAL ATTENTION

We first measure how much attention answer tokens allocate to visual tokens versus textual tokens across training. For each checkpoint (base model and epochs 1–4), we compute two aggregated statistics over the entire preference-training split (more than 5,000 multimodal examples):

- **Layer-wise mean image ratio**: for each layer, we compute the fraction of attention mass from answer tokens to image tokens (`image_sum_ratio`), then average this ratio across all layers.

- **Global image ratio across all layers**:

$$\text{Global Image Ratio} = \frac{\text{image-sum}}{\text{image-sum} + \text{text-sum}},$$

  where `image-sum` and `text-sum` aggregate attention weights to image/text tokens over all layers and heads.

Both aggregation schemes show a consistent global shift toward visual evidence: the mean image ratio increases from **0.1971** (base) to **0.2055** (epoch 4), a relative gain of about **+4.3%**; the global image ratio increases from **0.2204** to **0.2291**, a relative gain of about **+4.0%**. Although the absolute change ($\approx 0.0087$) is small on the $[0, 1]$ scale, these metrics are averaged over all layers, heads, and answer tokens, and are computed on the full training split, indicating a stable redistribution of attention probability mass from text to image tokens rather than noise on a small subset. The epoch-wise trend remains consistently above the base level and aligns with the reduction of hallucination metrics reported in the main results, suggesting that improved visual grounding is accompanied by a measurable shift in attentional reliance toward image tokens.

To better understand how this shift is organized inside the network, we further break down the visual attention ratio layer-wise by grouping the 32 Transformer layers into low (0–7), mid (8–17), and high (18–31) segments. The change is not a uniform "flat boost" of visual attention, but a structured redistribution:

- In the **low segment** (layers 0–7), the average visual ratio increases only modestly (about **+1.5%** relatively), and in some shallow layers (e.g., layers 2–5) it even slightly decreases.

- In the **mid segment** (layers 8–17) and **high segment** (layers 18–31), the average visual ratio increases more strongly, by roughly **+5.5%** and **+6.1%** respectively.

- Several mid/high layers exhibit the most prominent gains, for example

$$\text{layer 21: } 0.2535 \rightarrow 0.2710, \quad \text{layer 30: } 0.1058 \rightarrow 0.1283,$$

For some upper layers (e.g., layers 22, 27, 30), the visual attention ratio exhibits a nearly monotonic increase from epoch 0 to epoch 4, while shallow layers remain comparatively stable. This pattern indicates that the model is reallocating attention budget from early layers toward mid/high layers, which have been identified as key stages for visual–language fusion and hallucination mitigation

in prior analyses of LVLMs internals [2,3,4]. Importantly, our training objective does not impose any explicit layer-wise rescaling; the observed redistribution emerges from optimization under the calibration-based preference objectives, which are designed to make preference differences depend on visual interventions rather than purely textual paraphrasing.

## H.2 ATTENTION FOCUSING RATIO (AFR): FOCUSING ON GROUND-TRUTH OBJECT REGIONS

Global shifts toward image tokens do not, by themselves, guarantee better grounding. To verify that the model is increasingly focusing on *correct* regions, we revisit the Attention Focusing Ratio (AFR) defined in Sec. 5.2 of the main paper. Using TextVQA, which provides ground-truth bounding boxes.Higher $AFR_l$ indicates a stronger concentration of attention on the ground-truth object region relative to background or irrelevant areas.

In the main paper, AFR was reported only for the base model and final model. Here we include intermediate checkpoints at epochs 1–3 to track the full training trajectory. The *average AFR over all layers* increases from **5.54** (base) to **5.64** (epoch 4), a relative gain of about **+1.8%**, indicating a moderate but consistent strengthening of attention to key object regions. Grouping layers into low/mid/high segments yields:

- **Low**: $0.75 \to 0.79 \to 0.79 \to 0.78 \to 0.78$ (absolute change $\approx$ **+0.03**);
- **Mid**: $7.05 \to 7.03 \to 7.10 \to 7.12 \to 7.16$ (absolute change $\approx$ **+0.11**);
- **High**: $7.20 \to 7.30 \to 7.32 \to 7.32 \to 7.33$ (absolute change $\approx$ **+0.13**).

The largest absolute gains thus occur in mid/high segments, while the low segment fluctuates slightly around its base value. The overall shape of the AFR curve over layers (i.e., which layers act as peaks and valleys) remains almost unchanged: "visual–semantic" layers such as layer 14 already have high AFR in the base model and remain the main peaks, but their peak values are further amplified. For example,

$$\text{layer 14: } 14.74 \to 17.10 \to 18.02 \to 18.44 \to 18.72,$$

a relative increase of about **+27%**. Thus, the layers that were originally most responsible for focusing on ground-truth object regions become even more concentrated on these regions under our training, rather than being replaced by a different subset of layers.

Taken together, the global, layer-wise, and AFR analyses indicate that the proposed training scheme preserves the original hierarchical division of labor across layers while gently but directionally reallocating attention from shallow layers and background regions toward mid/high layers and ground-truth object regions. The model not only "looks more at the image" in aggregate, but does so in a more structured and semantically focused manner, consistent with the reductions in hallucination observed in the main quantitative results.

## I ADDITIONAL EXAMPLES

We present several additional examples to visually demonstrate the effectiveness of P$^2$-DPO in rectifying the specific perceptual failures we identified. These case studies highlight our model's ability to overcome the Perceptual Bottleneck, its enhanced robustness against visual noise, and its overall superior performance in complex reasoning compared to the baseline LLaVA-1.5-7B.

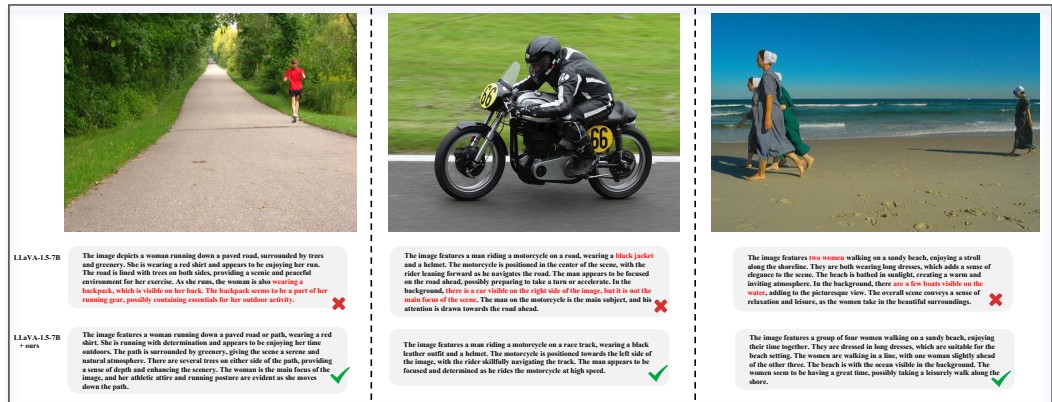

Figure 6: Example of improved overall performance in a complex scene. Given a general prompt such as "Describe this image", P$^2$-DPO produces a more accurate and detailed description than the baseline, showcasing the holistic improvements brought by our method.

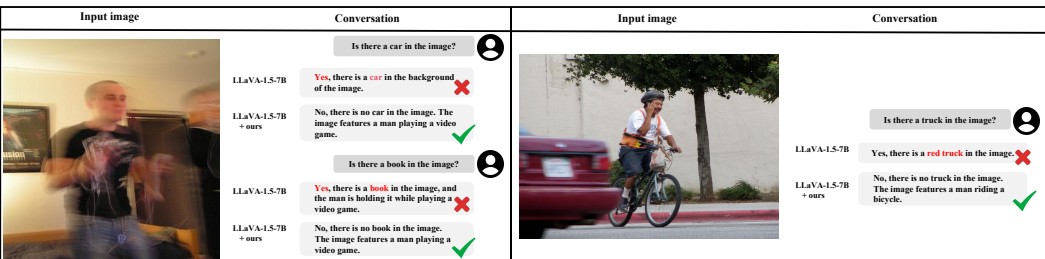

Figure 7: Visual robustness under motion blur. P$^2$-DPO correctly rejects hallucinated objects (e.g., "car", "book", "truck") that the baseline model incorrectly predicts, demonstrating improved resilience to visual degradation.

