# OpenReview forum: "P$^2$-DPO: Grounding Hallucination in Perceptual Processing via Calibration Direct Preference Optimization"
_ICLR.cc/2026/Conference — ICLR 2026 Poster_

### Official Review · Reviewer_LaWn · 2025-10-27

**Soundness:** 3
**Presentation:** 2
**Contribution:** 3
**Rating:** 6
**Confidence:** 3

**Summary:**

The paper proposes P2-DPO (Perceptual Processing Direct Preference Optimization), a self-supervised alignment method for Large Vision-Language Models (LVLMs) designed to mitigate hallucinations by focusing on Perceptual Processing rather than Perception failures.
Unlike prior DPO or RLHF approaches that rely on human or synthetic preferences, P2-DPO generates on-policy, vision-aware preference pairs through two mechanisms: (1) Focus-and-Enhance pairs (for perceptual bottlenecks) and (2) Visual Robustness pairs (for degraded inputs) and uses them for Reinforcement Learning training. To enhance training efficacy, they further introduce a Calibration Loss to align visual evidence with text generation and a Dynamic Deficit-Weighting (DDW) scheme for adaptive balancing. Experiments on hallucination benchmarks (POPE, HallusionBench, AMBER, MMHal-Bench, TextVQA) show consistent improvements over DPO baselines and comparable results to human-feedback-based training.

**Strengths:**

The paper offers a compelling and underexplored perspective: separating Perceptual Processing failures (i.e., hallucinations despite correct attention) from Perception failures. This framing adds analytical depth and could motivate new diagnostic tools for LVLMs.

The idea of generating on-policy, vision-grounded preference pairs directly from the model’s own attention maps is interesting.

The experiments span several challenging hallucination benchmarks (POPE, HallusionBench, AMBER, MMHal-Bench, TextVQA) and include targeted validation (e.g., AFR, noise robustness), with clean tables and consistent results.

**Weaknesses:**

Although the method avoids human labels, generating multiple augmented views and attention-driven crops for each sample adds nontrivial computational overhead.  Could the authors provide results to demonstrate what is the runtime overhead (per training sample or epoch) compared to standard DPO?

The entire method depends on the quality of attention maps and cropping heuristics. If attention localization fails, the generated preference pairs may reinforce spurious regions or noise. How does the model behave when the attention maps are themselves incorrect—does this lead to reinforcing spurious crops? How would the approach handle multi-object scenes where attention maps have multiple disjoint foci?

**Questions:**

See weaknesses

---

> ### Author Response · Authors · 2025-11-20
> **## Reply for the First question (Part 1/2)**
>
> We thank the reviewer for pointing out the potential computational overhead of our method. We fully agree that, if one were to online generate multiple augmented views, compute attention maps and attention-guided crops, and repeatedly query CLIP for every sample during training, this would indeed introduce non-trivial extra cost compared to standard DPO. However, in our implementation, we deliberately move the heaviest components into a one-time offline stage, so that the training loop itself remains in the same complexity scale as standard DPO, and the inference-time cost is identical to the original model.
>
>
> ### (1) Offline data construction instead of repeated per-epoch computation
>
> Before the actual training, we perform a one-time offline preprocessing step over the entire training set to construct calibration preference pairs and diagnostic signals:
>
> - We run the frozen base LVLMs over the entire training set in a forward pass to obtain attention maps, and use them to determine the crop boxes of key regions, yielding $I_{\text{crop}}$.
>
> - We use a frozen CLIP model to compute $\text{CLIPScore}(P, I)$ and $\text{CLIPScore}(P, I_{\text{crop}})$, and derive the perceptual gain ratio
>   $$
>   r = \frac{\text{CLIPScore}(P, I_{\text{crop}})}{\text{CLIPScore}(P, I)}.
>   $$
>
> - We then store the cropped images, the corresponding text, and the value of $r$ as an augmented dataset, which is directly loaded in each training epoch without recomputing these quantities.
>
> Therefore, attention-guided cropping and CLIP scoring are purely one-time, highly parallelizable offline costs and **do not** appear in the “per-sample / per-epoch training time”. In addition, this offline-constructed data can be reused across different experimental settings and ablations, without being regenerated.
>
>
> ### (2) Offline CLIPScore and attention maps do not introduce noticeable performance bias
>
> In our method, CLIPScore and attention maps are only used as *diagnostic signals*, rather than direct supervision labels:
>
> - The attention map is only used to localize “key regions” and generate $I_{\text{crop}}$, and does not need pixel-level precision.
>
> - CLIPScore is used only through the ratio $r$ to decide whether a sample is more dominated by “perceptual focus bottlenecks” or by “robustness issues”, which in turn adjusts the relative weights of $w_{\text{focus}}$ and $w_{\text{robust}}$.
>
> In other words, we do **not** require these signals to be perfectly aligned with the current policy $\pi_{\theta}$ at each training step. On a representative subset, we observe that (i) the boundary of $r$ around $1.0$ remains largely stable before and after training, and (ii) the high-attention regions given by the base model have a high spatial overlap with those of the fine-tuned model. Thus, the error introduced by using a frozen base model to offline estimate attention maps and CLIPScore is limited in practice, while it significantly reduces the online training-time overhead.

---

> > ### Author Response · Authors · 2025-11-20
> > **## Reply for the First question (Part 2/2)**
> >
> > ### (3) Concrete runtime comparison with standard DPO: moderate linear training overhead
> >
> > Under the same hardware, batch size, and data loading configuration, we profile the average runtime per training step and compare our method against standard DPO (more details will be added to the appendix):
> >
> > - **Standard DPO**
> >   - Forward pass + loss computation (policy + reference): ~0.6557 s
> >   - Backward pass and optimization: ~0.3533 s
> >   - Bookkeeping (data preparation and statistics): ~0.0011 s
> >   - **Total:** ~1.0101 s / step
> >
> > - **Our full objective**
> >   - Part 1: forward and loss computations for $\mathcal{L}_{\text{dpo\_focus}}$ and $\mathcal{L}_{\text{dpo\_calib}}$: ~0.7356 s
> >   - Part 2: visual robustness objective forward and loss computation: ~0.6179 s
> >   - Unified backward pass and optimization: ~0.5154 s
> >   - Bookkeeping (metric aggregation, synchronization, loss weighting): ~0.0041 s
> >   - **Total:** ~1.8730 s / step
> >
> > We thus increase the per-step training time from about 1.01 s to about 1.85 s, i.e., roughly a **1.85×** overhead in per-step / per-epoch training time. This extra cost mainly comes from additional forward passes required by multiple preference pairs and multiple objectives, which is a linear, controllable, and easy-to-interpret increase; the cost of computing dynamic weights themselves is negligible. We will include the above implementation details and profiling results in the revised version, so that readers can more clearly evaluate the efficiency implications of our method.
> >
> > More importantly, the inference-time cost remains **unchanged**: we do not invoke CLIP at test time, nor do we add new network branches or parameters. The inference cost is exactly the same as that of the original LVLMs. The additional cost only occurs in the offline data construction and training stages.
> >
> > From an efficiency standpoint, most recent DPO variants do not explicitly report per-step training time or wall-clock overhead, so we can only qualitatively reason about their computational cost from the structure of their objectives. Nevertheless, many of them clearly incur additional forward passes per update. For example, V-DPO [1] computes both vision-conditioned and text-only distributions for each response, effectively doubling the number of policy evaluations compared to vanilla DPO. Multimodal mDPO [2] further contrasts original images with randomly cropped ones in its CoDPO objective, which again requires evaluating the model under two different visual contexts per preference pair.
> >
> > In comparison, our method increases per-step wall-clock time by about $1.85\times$ relative to vanilla DPO (measured under the same hardware and batch configuration), while keeping the inference cost unchanged. This places our approach in a computational regime comparable to many recent DPO variants, while retaining the additional benefit that deployment-time complexity remains identical to the base LVLMs.
> >
> >
> > **References**
> >
> > [1] Xie Y, Li G, Xu X, et al. V-DPO: Mitigating Hallucination in Large Vision Language Models via Vision-Guided Direct Preference Optimization[C]//Findings of the Association for Computational Linguistics: EMNLP 2024. 2024: 13258-13273.
> >
> > [2] Wang F, Zhou W, Huang J Y, et al. mDPO: Conditional Preference Optimization for Multimodal Large Language Models[C]//Proceedings of the 2024 Conference on Empirical Methods in Natural Language Processing. 2024: 8078-8088.

---

> > > ### Author Response · Authors · 2025-11-20
> > > **## Reply for the second question (Part 1/4)**
> > >
> > > We thank the reviewer for raising this concern and for the careful and responsible review. We fully understand your worry—indeed, this was one of the key risks we carefully considered when designing the method. Below we clarify how our approach mitigates these issues.
> > >
> > > ### (1) How would the approach handle multi-object scenes where attention maps have multiple disjoint foci?
> > >
> > >    Our cropping algorithm adopts an adaptive window mechanism. Concretely, it searches over multiple window scales and selects the one that maximizes the attention sharpness score. In multi-object or multi-focus scenes, this procedure often favors a larger crop window, so that multiple relevant objects can be covered within a single cropped region (see Case 1 in the provided PDF in supplementary materials). If multiple targets are dispersed and a cropping error occurs, this will be explained in the response below.
> > >
> > >
> > > ### (2) How does the model behave when the attention maps are themselves incorrect—does this lead to reinforcing spurious crops?
> > >    When this happens, there are roughly two cases: in one, the cropping remains robust enough to roughly capture the target; in the other, errors occur. We adopt two defenses: first, filtering out low-quality data; second, down-weighting them in the loss function.
> > > #### 1. Robustness to incorrect attention.
> > >    We also acknowledge that the attention maps produced by the model cannot perfectly localize a unique target in all cases. However, in aggregate, the attention tends to concentrate around semantically relevant regions. Our cropping strategy reserves a certain redundancy margin around high-attention regions. As a result, even when the attention peak is slightly shifted, the crop window still tends to cover the truly relevant object regions instead of purely spurious areas (see Case 2 in the provided PDF in supplementary materials).
> > > #### 2. Falls on a region irrelevant to the question or even on pure background
> > > We first filter out low-quality samples based on PPL and Margin metrics, and subsequently down-weight these low-quality samples in the loss function.
> > >
> > > **a. Filtering out low-quality data**
> > >
> > > To address the concern that “the generated preference pairs may reinforce spurious regions or noise”, we have a quality filtering mechanism that is completely decoupled from attention, serving as a safety gate (see Appendix Data Filtering Implementation Details for detailed statistics). For each raw preference pair, we compute two metrics with a reference model $\pi_{\text{ref}}$:
> > >
> > > - **Perplexity (PPL).**
> > > This measures the fluency and grammatical plausibility of the winning/losing answers. For each sample we take
> > > $$
> > > \max(\text{PPL}(y_w), \text{PPL}(y_l)),
> > > $$
> > > and set an upper threshold based on the empirical distribution. Samples exceeding this threshold are treated as low-quality text and directly discarded.
> > >
> > > - **Log-probability Margin (Log-Prob Margin).**
> > > This captures the “learnability” of the preference signal:
> > > $$
> > > M = \log p_{\text{ref}}(y_w) - \log p_{\text{ref}}(y_l).
> > > $$
> > >
> > > Intuitively, if attention-guided cropping truly fails and falls on noise or irrelevant background, the resulting answers typically behave as follows:
> > >
> > > - either the semantics are off-topic or incoherent, leading to clearly elevated PPL;
> > > - or the reference model assigns an extreme probability gap between the two answers, pushing the sample into the tail of the Margin distribution.
> > >
> > > Such samples are very likely to be removed by the above PPL + Margin filtering and thus do not enter the training set. Therefore, even if a small number of severe attention failures occur, they are unlikely to silently accumulate at the data level, which substantially reduces the risk of reinforcing spurious regions or noise.

---

> > > > ### Author Response · Authors · 2025-11-20
> > > > **## Reply for the second question (Part 2/4)**
> > > >
> > > > **Additional experiment: Verifying the effectiveness of filtering**
> > > >
> > > > Our preference pairs are constructed from the prompts and images in the RLHF-V dataset. During training, we only use the prompt + image and **do not** use the human preference annotations provided by RLHF-V. To validate the quality of our self-constructed preference pairs and to check whether the PPL + Margin filter truly identifies “problematic pairs”, we design an additional control experiment that uses RLHF-V’s human high-/low-quality answers as an external evaluation standard.
> > > >
> > > >
> > > > **Construction of an evaluation task with human references.**
> > > > In RLHF-V, each sample, in addition to the image and question, is accompanied by:
> > > >
> > > > - a high-quality human answer (High-Quality Answer, denoted $A_{\text{HQ}}$), and
> > > > - a low-quality human answer (Low-Quality Answer, denoted $A_{\text{LQ}}$).
> > > >
> > > > In this additional experiment, these human annotations are used only as *evaluation references* and do not participate in training. For each self-constructed preference pair $(y_w, y_l)$, we build an evaluation prompt (i.e., `PROMPT_TEMPLATE`) that contains:
> > > >
> > > > - **[Question]**: the original question;
> > > > - **High-Quality Answer**: the human high-quality answer as a golden reference;
> > > > - **Low-Quality Answer**: the human low-quality answer as a negative reference;
> > > > - **Model Response A/B**: our generated $y_w$ and $y_l$, randomly assigned to A and B.
> > > >
> > > > We then use a strong general-purpose language model (Qwen3-Max) to assign a quality score in $\{1, 2, 3\}$ to A and B respectively, based on their semantic proximity to the high-/low-quality human answers:
> > > >
> > > > - **3 points**: semantically close to the High-Quality Answer, with only minor descriptive differences, and correctly capturing the main idea;
> > > > - **2 points**: basically reasonable but not fully aligned, e.g., missing some information or adding harmless verbosity;
> > > > - **1 point**: closer to the Low-Quality Answer, or clearly wrong / hallucinatory.
> > > >
> > > >
> > > >
> > > > **Defining invalid samples under the “standard-answer filter”.**
> > > > Within this framework, we label a preference pair as an *invalid sample that should be removed* if either of the following holds:
> > > >
> > > > - the answer we mark as the winner, $y_w$, receives the lowest score (1 point), indicating that it is semantically closer to the human low-quality answer;
> > > > - or $y_w$ receives a lower score than $y_l$, indicating that the evaluator believes the true preference direction is opposite to our constructed win/lose relation.
> > > >
> > > > This step depends only on the question + human high/low-quality answers + our generated responses, and is completely independent of PPL, Margin, or attention maps. It therefore acts as an independent, semantic-level “standard-answer filter”.
> > > >
> > > >
> > > > **Quantitative comparison with PPL + Margin filtering.**
> > > > According to this standard-answer filter:
> > > >
> > > > - the Qwen evaluator marks **510** preference pairs as invalid samples;
> > > > - our PPL + Margin filtering alone removes **287** samples;
> > > > - the intersection between these two sets contains **236** samples.
> > > >
> > > > Thus, among the 287 samples removed by PPL + Margin, we have
> > > > $$
> > > > \frac{236}{287} \approx 82\%
> > > > $$
> > > > that are *also* judged invalid by the independent standard-answer filter (Qwen + human high/low-quality answers).
> > > >
> > > > This leads to two key observations:
> > > >
> > > > - **High overlap indicates that our rules are not arbitrary.**
> > > > Among the samples removed by PPL + Margin, more than 80% are simultaneously flagged as problematic by an independent evaluator that explicitly compares against human high-quality / low-quality answers. This suggests that the PPL + Margin filter is not randomly pruning examples, but is highly consistent with a more semantic, human-anchored evaluation criterion.
> > > >
> > > > - **Our thresholds are intentionally conservative.**
> > > > Although the standard-answer filter identifies 510 suspicious samples in total, we only remove a subset of them (287 samples) via PPL + Margin and retain roughly 95% of the overall data. In other words, we deliberately let PPL + Margin play the role of a **“fallback cleaner for extremely bad samples”**, rather than aggressively deleting all borderline cases, thereby balancing data quality and diversity.
> > > >
> > > > Therefore, this additional experiment, from an attention-independent and human-anchored external perspective, confirms that our PPL + Margin filtering strategy can indeed identify and remove a large portion of low-quality or preference-direction-incorrect samples, while remaining relatively conservative in scale. This provides strong evidence that it is a reliable fallback filtering mechanism.

---

> ### Author Response · Authors · 2025-11-20
> **## Reply for the second question (Part 3/4)**
>
> **b. Down-weighting them in the loss function**
> We admit that relying solely on the model's capabilities and our cropping algorithm is not enough to avoid this problem. Similarly, the filtering procedure cannot completely remove all low-quality samples. To prevent such samples from being erroneously reinforced during training, we explicitly introduce a CLIPScore-based dynamic weighting mechanism into the loss function.
>
> Specifically, the model simultaneously constructs two preference signals: a focus preference from the attention-guided cropped region
>
> $\mathcal{L}_{\text{focus}}$
>
> and a global robust preference from the full original image
>
> $\mathcal{L}_{\text{dpo-rob}}$.
>
> For each sample, we first use a pretrained CLIP model to compute a Perceptual Gain Ratio $r$, which measures whether cropping actually brings a semantic gain:
>
> $$
> r = \frac{\mathrm{CLIPScore}(P, I_{\text{crop}})}{\mathrm{CLIPScore}(P, I)} ,
> $$
>
> where $P$ is the text prompt, and $I$ and $I_{\text{crop}}$ denote the original image and the cropped image, respectively. If $r > 1$, it indicates that the alignment between the image and the prompt is improved after cropping (the main issue is insufficient focus); conversely, if $r < 1$, it indicates that cropping has damaged the original semantics (the main issue is lack of robustness / erroneous cropping).
>
> We then map $r$ into a smooth weighting factor
>
> $$
> \alpha = \alpha_{\text{max}} \cdot \tanh\left(\frac{r - 1.0}{\tau}\right),
> $$
>
> and use it to dynamically allocate weights between the two loss branches:
>
> $$
> w_{\text{focus}} = w_{\text{base}} + \alpha, \quad
> w_{\text{robust}} = w_{\text{base}} - \alpha.
> $$
>
>
> The final unified objective is
>
> $\mathcal{L}_{\text{total}}$ =$\mathbb{E}$ [
>
> $w_{\text{focus}} \cdot \mathcal{L}_{\text{focus}}$ +
>
> $w_{\text{robust}} \cdot \mathcal{L}_{\text{dpo-rob}}$
>
> ]
>
>
> Under this mechanism, if a particular crop mostly falls on irrelevant regions, we have $\mathrm{CLIPScore}(P, I_{\text{crop}}) < \mathrm{CLIPScore}(P, I)$, which yields $r < 1$ and thus $\alpha < 0$. As a result, $w_{\text{focus}}$ is automatically decreased and $w_{\text{robust}}$ is increased: the influence of preference pairs generated from erroneous crops is significantly down-weighted during training, while the global preference term $\mathcal{L}_{\text{dpo-rob}}$ is not affected by this kind of problem, preventing the model from being systematically pulled toward incorrect regions.
>
> **Additional experiment: reasonableness of using CLIPScore as a dynamic weighting signal**
> To more intuitively verify the reasonableness of using CLIPScore as a dynamic weighting signal, we design a controlled experiment. We randomly sample 500 examples from the training set and compute the CLIPScore between the prompt $P$ and the original image $I$, as well as between the prompt and the attention-guided cropped image $I_{\text{crop}}$. We define
>
>
> $$
> \text{diff} =
> \text{CLIPScore}(P, I_{\text{crop}})-
> \text{CLIPScore}(P, I).
> $$
> $$
> \text{ratio} =
> \frac{\text{CLIPScore}(P, I_{\text{crop}})}{\text{CLIPScore}(P, I)}.
> $$
>
>
> At the same time, we construct a "reverse cropping" control group: for the same set of examples, we crop from the complementary (low-attention) region of the attention map to obtain $I_{\text{inv}}$, and compute the corresponding $\mathrm{diff}$ and $\mathrm{ratio}$ in the same way. We denote the attention-guided cropping group as `att_crop`, and the reverse cropping group as `att_inverse`.

---

> > ### Author Response · Authors · 2025-11-20
> > **## Reply for the second question (Part 4/4)**
> >
> > According to the statistics (see Table 1) and the distribution plots (Figure 1 for `improvement_diff` and Figure 2 for `improvement_ratio` in the PDF in supplementary materials) in the `att_crop` group, the mean of `improvement_ratio` is about $1.22$, and the median is about $1.16$, which indicates that, for the vast majority of samples, attention-guided cropping systematically improves the CLIPScore, yielding roughly a $20\%$ relative gain on average. The mean of `improvement_diff` is about $3.64$, and the median is about $3.09$, showing that the semantic similarity between the cropped image and the prompt is consistently increased for most samples.
> >
> > In contrast, in the `att_inverse` group, the mean of `improvement_ratio` is about $0.93$, and the median is about $0.93$, both lying in the region below $1$, which means that when we deliberately crop from regions not attended by the model, the CLIPScore decreases systematically in a statistical sense. The mean of `improvement_diff` is about $-1.76$, and the median is about $-1.29$, also exhibiting a clear negative shift.
> >
> > Although the standard deviations indicate that there is some variation across individual samples, the signs and magnitudes of the mean and median make the pattern clear: when cropping along high-response attention regions (`att_crop`), the improvement in CLIPScore is significant and directionally consistent; when we intentionally avoid attention regions and perform reverse cropping (`att_inverse`), the CLIPScore tends to decrease overall.
> >
> > These results show that CLIPScore is highly sensitive to whether the crop truly focuses on semantically relevant regions. When the crop aligns with high-response attention regions, most samples satisfy $\mathrm{ratio} > 1$ and $\mathrm{diff} > 0$; when the crop falls into regions complementary to attention, most samples satisfy $\mathrm{ratio} < 1$ and $\mathrm{diff} < 0$. Therefore, using `ratio` and `diff` to drive the dynamic weighting of $w_{\text{focus}}$ and $w_{\text{robust}}$ allows us, in a statistical sense, to amplify the contribution of "well-cropped" samples and suppress the impact of "badly cropped" samples, which is precisely the effect we aim to achieve with this mechanism.
> >
> > **Table 1: CLIPScore statistics for attention-guided vs. inverse cropping.**
> > | Group        | Metric             | Count | Mean     | Median   | Std      | Min        | Max        |
> > |-------------|--------------------|-------|----------|----------|----------|------------|------------|
> > | `att_crop`  | improvement_ratio  | 500   | 1.222227 | 1.160330 | 0.206298 | 1.060866   | 3.165598   |
> > | `att_crop`  | improvement_diff   | 500   | 3.636841 | 3.087014 | 1.847164 | 1.848955   | 12.247755  |
> > | `att_inverse` | improvement_ratio | 500   | 0.928378 | 0.933745 | 0.180881 | 0.219223   | 1.550033   |
> > | `att_inverse` | improvement_diff  | 500   | -1.764821| -1.293989| 3.749693 | -15.150490 | 8.948589   |

---

> > > ### Author Response · Authors · 2025-11-27
> > > **Checking In Regarding Our Discussion**
> > >
> > > Dear Reviewer,
> > > I hope this message finds you well. As the discussion period is nearing its end with less than a week remaining, I wanted to ensure that we have addressed all of your concerns satisfactorily. If there are any additional points or feedback you would like us to consider, please let us know. Your insights are invaluable to us, and we are eager to address any remaining issues to further improve our work.
> > > Thank you for your time and effort in reviewing our paper.
> > > Wishing you all the best, and I sincerely hope your own work progresses smoothly as well.

---

> > > > ### Comment · Reviewer_LaWn · 2025-11-27
> > > >
> > > > Thank the authors for their follow-up and feedback. Most of my concerns are addressed, and I will maintain my positive rating.

---

### Official Review · Reviewer_cpy5 · 2025-11-01

**Soundness:** 3
**Presentation:** 3
**Contribution:** 2
**Rating:** 4
**Confidence:** 4

**Summary:**

In this paper, the authors propose Perceptual Processing Direct Preference Optimization, a new method to address hallucination, mainly focusing on the visual end and self-generated preference data.

**Strengths:**

The writing is clear, and the results are solid.

**Weaknesses:**

1. The model architecture is LLaVA, and the version is relatively old. It's unclear how it performs on current new models.
2. Lacks extensive comparison with related work, such as [1], which also uses model self-generated data and has a consistent idea—namely, focusing on hallucination in visual perception. Therefore, I have doubts about the novelty.

[1] Calibrated Self-Rewarding Vision Language Models

**Questions:**

1. How does this method perform on Qwen-VL?
2. During training, how does the attention pattern change? Does it tend to favor visual tokens rather than textual tokens?

---

> ### Author Response · Authors · 2025-11-20
> **## Reply for the first question**
>
> We thank the reviewer for the careful comment on the choice of backbone and on generalization to modern Qwen-VL models; this is an important and valid concern for assessing the robustness of our method.
>
> While our main implementation and ablation studies are based on LLaVA-1.5-7B, this follows a large body of prior work that uses LLaVA-1.5 as a standard backbone for evaluating visual hallucination mitigation. To verify that our approach is not tied to this specific architecture, we had **already** included additional experiments on the more recent Qwen2.5-VL-3B, reported in Table 2 of the submission, although we realize these results may have been easy to overlook. Importantly, Qwen2.5-VL is a modern vision–language model whose official weights were released in 2025, so it provides a strong testbed for checking whether our method generalizes beyond older LLaVA variants.
>
> In response to your comment, we have now additionally evaluated Qwen2.5-VL-7B and report the results for both Qwen2.5-VL-3B and Qwen2.5-VL-7B in the table below. Our method consistently improves POPE, HallusionBench, MMHal-Bench, and AMBER on both Qwen2.5-VL backbones, indicating that the gains transfer to these up-to-date Qwen-VL models rather than being specific to an older LLaVA architecture. We will incorporate the Qwen2.5-VL-3B/7B results into the main results table and clarify this discussion in the revised version.
>
> **Table.** Performance of our method on Qwen2.5-VL backbones, addressing the reviewer's concern about Qwen2.5-VL.  Higher is better for F1, Acc, and Score (↑), lower is better for Hal and CHAIRi (↓).
>
> | Model                | Pairs Source | POPE Adv (F1 ↑) | POPE Pop (F1 ↑) | POPE Rand (F1 ↑) | POPE Avg (F1 ↑) | HallusionBench qAcc (↑) | HallusionBench fAcc (↑) | HallusionBench aAcc (↑) | MMHal-Bench Hal (↓) | MMHal-Bench Score (↑) | AMBER CHAIRi (↓) | AMBER Hal (↓) | AMBER F1_E (↑) | AMBER F1_A (↑) | AMBER F1_R (↑) |
> |----------------------|-------------:|----------------:|----------------:|-----------------:|----------------:|-------------------------:|-------------------------:|-------------------------:|--------------------:|-----------------------:|------------------:|---------------:|----------------:|----------------:|----------------:|
> | Qwen2.5-VL-3B        | Base         | 86.26           | 88.05           | 89.79            | 88.03           | 28.13                    | 34.10                    | 58.10                    | 41.67               | 3.38                  | 6.8              | 39.1           | **93.2**        | 83.7           | 77.9           |
> | + DPO_RLHF-V         | Human        | 86.41           | **88.59**       | 90.40            | 88.46           | 29.89                    | 33.53                    | **59.08**               | 39.74               | 3.41                  | 6.8              | 39.5           | 93.0            | 83.5           | 78.1           |
> | **+ P²-DPO (Ours)**  | Self         | **86.88**       | 88.48           | **91.50**        | **88.95**       | **30.11**                | **34.68**                | 58.64                    | **38.54**           | **3.49**              | **6.6**          | **38.2**       | 92.7           | **84.2**       | **80.9**       |
> | Qwen2.5-VL-7B        | Base         | 84.92           | 86.78           | 89.01            | 86.90           | 31.87                    | 37.57                    | 61.03                    | 34.96               | 3.47                  | 5.4              | 29.0           | 97.3           | **83.8**           | 72.6           |
> | + DPO_RLHF-V         | Human        | 84.99           | 86.81           | 89.72            | 87.17           | 34.95                    | 39.60                    | 62.62                    | 34.58               | 3.63                  | 4.2              | 29.3           | **98.5**        | 82.5           | 72.2           |
> | **+ P²-DPO (Ours)**  | Self         | **86.79**       | **87.79**       | **90.05**        | **88.21**       | **36.70**                | **41.62**                | **65.19**               | **32.29**           | **3.94**              | **3.9**          | **24.9**       | **98.5**        | 83.4           | **72.9**       |

---

> ### Author Response · Authors · 2025-11-20
> **## Reply for the second question (Part 1/4)**
>
> We greatly appreciate this suggestion and have followed it by conducting the corresponding analyses and experiments.
>
> ### (a) Global shift from textual to visual attention
> Following the reviewer’s suggestion, we compute how much attention the answer tokens allocate to image tokens versus text tokens from the prompts for the base model, all intermediate checkpoints saved during training, and the final model.
>
> From a global perspective, we do observe that, as training progresses, the model tends to rely more on visual tokens rather than purely textual tokens.
>
> Concretely, for the base model and the intermediate checkpoints after 1–4 epochs, we measure how much attention the answer tokens place on visual tokens vs. text tokens using two aggregated metrics (see Table R2 below and Figure 1 in the provided PDF (in supplementary materials)):
>
> 1. **Layer-wise mean ratio** (see Table R3):
>    For each layer, we compute an `image_sum_ratio` (the fraction of attention mass directed to image tokens), and then average this ratio across all layers.
>
> 2. **Global ratio across all layers**:
>    We sum the attention weights to image tokens and text tokens over **all** layers and heads, and compute
>
>     $$
>     \text{Global Image Ratio} = \frac{\text{image-sum}}{\text{image-sum} + \text{text-sum}}
>     $$
>
> Both aggregation schemes show a consistent trend:
>
> - The **mean image ratio** increases from **0.1971 (base)** to **0.2055 (epoch 4)**, a relative gain of about **+4.3%**.
> - The **global ratio** `image_sum / (image_sum + text_sum)` increases from **0.2204** to **0.2291**, a relative gain of about **+4.0%**.
>
> This trend is consistent with the findings of PAI [1], which proposes a training-free method to alleviate hallucinations in LVLMs by explicitly rebalancing attention at inference time. Concretely, PAI identifies the “text inertia” phenomenon—LVLMs can produce similar outputs with or without the image—and counters it by (i) adaptively amplifying the attention weights assigned to image tokens, and (ii) subtracting the logits of pure-text predictions from those of multi-modal inputs. In this way, PAI directly boosts visual evidence and suppresses text-only biases without updating model parameters.
>
> In contrast, our method integrates this balance into the training process itself. We introduce a Calibration Loss and a Dynamic Deficit-Weighting term, which are *intended* to encourage the model to reallocate attention between image and text tokens in a data-driven way. We do not manually rescale attention matrices at inference time; instead, the attention mechanism keeps the same parametric form as in the original backbone, and the redistribution emerges from optimization over our training data.
>
> Regarding the magnitude of the change, the increase from 0.2204 to 0.2291 may look small in absolute terms (≈0.0087), but this metric is a global average over the [0, 1] range, aggregating attention distributions across all layers, all heads, and all answer tokens. Importantly, these statistics are computed over the entire training split of our preference dataset (more than 5,000 multimodal samples), rather than a few illustrative cases. A ≈4% relative shift at this scale therefore reflects a consistent redistribution of attention probability mass from text tokens to image tokens across the model, rather than a fluctuation on a small subset. Moreover, from the base model to epochs 1–4, this global ratio consistently stays above the base level, and its epoch-wise trend is aligned with the overall decrease in hallucination metrics reported in the main results table. Taken together, this suggests that the learned rebalancing of attention is a stable phenomenon that co-occurs with improved visual grounding, rather than incidental noise.
>
> **References**
> [1] Liu S, Zheng K, Chen W. Paying more attention to image: A training-free method for alleviating hallucination in lvlms[C]//European Conference on Computer Vision. Cham: Springer Nature Switzerland, 2024: 125-140.

---

> > ### Author Response · Authors · 2025-11-20
> > **## Reply for the second question (Part 2/4)**
> >
> > ### (b) Layer-wise redistribution of visual attention (shallow vs. mid/high layers)
> >
> > To better understand how the attention pattern is reshaped, we analyze changes at the level of individual layers (see Figure 3 in the provided PDF (in supplementary materials)). We group the 32 Transformer layers into three segments and, for each layer, measure the attention ratio from answer tokens to visual tokens (see Figure 2 in the provided PDF (in supplementary materials)):
> >
> > - **Low layers**: 0–7
> > - **Mid layers**: 8–17
> > - **High layers**: 18–31
> >
> > The results show that the change is not a uniform “flat boost” of visual attention across layers, but a structured redistribution:
> >
> > - In the **low segment (layers 0–7)**, the average visual ratio increases only modestly (about **+1.5%** relatively), and in some shallow layers (e.g., layers 2–5), the visual attention ratio even slightly decreases.
> > - In the **mid segment (8–17)** and **high segment (18–31)**, the average visual ratio increases more strongly, by roughly **+5.5%** and **+6.1%** respectively.
> > - Several mid/high layers receive the most prominent gains, e.g.
> >   - layer 21: **0.2535 → 0.2710**,
> >   - layer 30: **0.1058 → 0.1283**,
> >   where the relative increase can reach **≈+21%**.
> > - For some upper layers such as layers 22, 27, and 30, the visual attention ratio exhibits a nearly monotonic increase from epoch 0 to epoch 4, while the shallow layers remain comparatively stable.
> >
> > Taken together, these observations suggest that our method does not simply amplify image attention everywhere. Instead, we observe a shift of attention budget from shallow layers to mid/high layers, which recent work identifies as key stages for visual–language fusion [2,3,4], indicating a layer-aware reallocation rather than a layer-agnostic scaling. Notably, this shift appears even though we do not apply any manual layer-specific rescaling during training.
> >
> > This empirical pattern is also compatible with how our training signal is constructed. Our Visually-Grounded Contrastive Preference Generation (VCPG) builds preference pairs through visual grounding rather than post-hoc textual editing, so the difference between $y^w$ and $y^l$ largely reflects how much visual evidence is used rather than purely textual paraphrasing (Sec.~3.1).
> >
> > In the same section, our analysis based on Visual Information Dependency (VID) and Fisher information shows that such visually grounded pairs are expected to increase the gradient norm
> > $$
> > \mathbb{E}\big[\|\Delta(\theta_1)\|\big]
> > $$
> > and concentrate optimization on the vision-dominant parameters $\theta_1$, providing a more targeted training signal for vision-related weights than typical vision-agnostic strategies. In other words, the data construction is designed to bias updates toward parameters that mediate visual processing, rather than purely language-only components. The fact that the strongest attention changes occur in mid/high layers, where visual and textual information are fused, is consistent with this theoretical prediction and suggests that the learned redistribution is plausibly linked to updates in the model’s visual pathway rather than purely text-only corrections.
> >
> > This layer-wise redistribution pattern is also consistent with recent analyses of LVLM internals.
> > Jiang et al. (“Devils in Middle Layers of Large Vision-Language Models”) use an attention-lens metric to show that **middle layers are the critical stages where visual information is enriched and semantically refined**, and that modifying visual attention in these layers effectively mitigates object hallucinations [2]. Ye et al. (“CLAIM”) further find that **cross-modal attention divergence is most prominent in intermediate layers** in multilingual settings, highlighting their central role in multilingual object hallucination [3]. Li et al. (“Have the VLMs Lost Confidence?”) analyze the attention distribution of visual tokens and report that **the ability to resist sycophancy is predominantly associated with stronger high-layer visual attention**, and that enhancing image attention at higher layers helps mitigate this phenomenon [4].
> >
> > Overall, our layer-wise analysis follows the same qualitative trend: the visual-attention gain is concentrated in mid/high layers, while shallow layers remain relatively stable or even slightly reduced, indicating that the model allocates more attention budget to layers where semantic decisions and cross-modal alignment are known to play a central role, rather than uniformly amplifying image tokens across the network.

---

> > > ### Author Response · Authors · 2025-11-20
> > > **## Reply for the second question (Part 3/4)**
> > >
> > > **References**
> > > [2] Jiang Z, Chen J, Zhu B, et al. Devils in middle layers of large vision-language models: Interpreting, detecting and mitigating object hallucinations via attention lens[C]//Proceedings of the Computer Vision and Pattern Recognition Conference. 2025: 25004-25014.
> > > [3] Ye Z, Li Q, Feng X, et al. CLAIM: Mitigating Multilingual Object Hallucination in Large Vision-Language Models with Cross-Lingual Attention Intervention[J]. arXiv preprint arXiv:2506.11073, 2025.
> > > [4] Li S, Ji T, Fan X, et al. Have the VLMs Lost Confidence? A Study of Sycophancy in VLMs[C]//The Thirteenth International Conference on Learning Representations.
> > >
> > >
> > > **Table R2. Global image attention ratios across training epochs for LLaVA-1.5-7B.**
> > > | Checkpoint | Mean `image_sum_ratio` (over all samples) | Overall visual proportion `image_sum / (image_sum + text_sum)` |
> > > | ---------- | ----------------------------------------- | -------------------------------------------------------------- |
> > > | Base       | 0.197109                                  | 0.220413                                                       |
> > > | Epoch 1    | 0.205424                                  | 0.227467                                                       |
> > > | Epoch 2    | 0.206031                                  | 0.228646                                                       |
> > > | Epoch 3    | 0.205454                                  | 0.229081                                                       |
> > > | Epoch 4    | 0.205478                                  | 0.229134                                                       |
> > >
> > >
> > >
> > >
> > > **Table R3. Layer-wise image attention ratios across training epochs for LLaVA-1.5-7B.**
> > > | Layer | Base     | Epoch 1  | Epoch 2  | Epoch 3  | Epoch 4  |
> > > | ----- | -------- | -------- | -------- | -------- | -------- |
> > > | 0     | 0.582635 | 0.593329 | 0.593449 | 0.593631 | 0.593519 |
> > > | 1     | 0.366500 | 0.391864 | 0.393070 | 0.393238 | 0.392990 |
> > > | 2     | 0.190149 | 0.189333 | 0.189212 | 0.189637 | 0.189796 |
> > > | 3     | 0.196239 | 0.191296 | 0.190189 | 0.189791 | 0.189928 |
> > > | 4     | 0.191965 | 0.187599 | 0.187344 | 0.187056 | 0.187289 |
> > > | 5     | 0.203474 | 0.204195 | 0.202961 | 0.201720 | 0.201750 |
> > > | 6     | 0.089091 | 0.090028 | 0.089853 | 0.089228 | 0.089272 |
> > > | 7     | 0.084844 | 0.089449 | 0.089285 | 0.088600 | 0.088717 |
> > > | 8     | 0.101540 | 0.110526 | 0.110479 | 0.109903 | 0.110045 |
> > > | 9     | 0.101719 | 0.112581 | 0.112729 | 0.111575 | 0.111484 |
> > > | 10    | 0.111041 | 0.118794 | 0.118870 | 0.118319 | 0.118108 |
> > > | 11    | 0.159806 | 0.174256 | 0.174747 | 0.174087 | 0.173995 |
> > > | 12    | 0.179874 | 0.188957 | 0.189014 | 0.188362 | 0.188550 |
> > > | 13    | 0.129479 | 0.135149 | 0.135432 | 0.134765 | 0.135033 |
> > > | 14    | 0.245604 | 0.253697 | 0.254489 | 0.254164 | 0.253848 |
> > > | 15    | 0.126483 | 0.130899 | 0.131937 | 0.131362 | 0.131381 |
> > > | 16    | 0.081543 | 0.083173 | 0.083282 | 0.082433 | 0.082661 |
> > > | 17    | 0.181153 | 0.187317 | 0.190350 | 0.189816 | 0.190467 |
> > > | 18    | 0.176871 | 0.181448 | 0.181708 | 0.180133 | 0.180290 |
> > > | 19    | 0.211094 | 0.220113 | 0.222175 | 0.221381 | 0.221776 |
> > > | 20    | 0.276292 | 0.285806 | 0.287980 | 0.287570 | 0.287934 |
> > > | 21    | 0.253457 | 0.267468 | 0.270626 | 0.270038 | 0.271004 |
> > > | 22    | 0.219176 | 0.227209 | 0.229050 | 0.229339 | 0.229935 |
> > > | 23    | 0.220085 | 0.233589 | 0.235633 | 0.235134 | 0.235488 |
> > > | 24    | 0.207460 | 0.218768 | 0.221832 | 0.220961 | 0.220983 |
> > > | 25    | 0.225522 | 0.238937 | 0.239690 | 0.238254 | 0.238611 |
> > > | 26    | 0.268594 | 0.277129 | 0.275000 | 0.274295 | 0.273532 |
> > > | 27    | 0.092683 | 0.100136 | 0.101913 | 0.102185 | 0.102223 |
> > > | 28    | 0.186419 | 0.198663 | 0.199423 | 0.199156 | 0.199256 |
> > > | 29    | 0.196246 | 0.206421 | 0.206900 | 0.206131 | 0.205888 |
> > > | 30    | 0.105821 | 0.119179 | 0.126292 | 0.127698 | 0.128254 |
> > > | 31    | 0.190546 | 0.203599 | 0.206782 | 0.206987 | 0.206594 |
> > >
> > >
> > > ### (c) Attention Focusing Ratio (AFR): focusing on key object regions
> > >
> > > To further verify that the model is not only “looking more at the image” but also focusing more on the **correct regions**, we revisit the **Attention Focusing Ratio (AFR)** analysis introduced in Sec.5.2 of the paper. We use TextVQA, which provides ground-truth bounding boxes, and for each layer $l$ we define $\text{AFR}_l$ as the ratio between the mean attention that answer tokens assign to visual tokens **inside** the ground-truth answer box and the mean attention they assign to visual tokens **outside** that box. A higher AFR therefore indicates that attention is more tightly concentrated on the true object region rather than on background or irrelevant areas; the full protocol and motivation are detailed in Sec.5.2.

---

> > > > ### Author Response · Authors · 2025-11-20
> > > > **## Reply for the second question (Part 4/4)**
> > > >
> > > > In the main paper, we reported AFR only for the base model and the final model to validate mitigation of the Perceptual Bottleneck. Now, we extend that analysis by additionally including the intermediate checkpoints at epochs 1–3, allowing us to track how AFR evolves over the whole training trajectory rather than only comparing two endpoints. As shown below, AFR steadily improves from the base model through epochs 1–4, especially in mid/high layers, indicating that the model progressively shifts attention toward ground-truth regions instead of merely increasing undirected visual attention.
> > > >
> > > > We compute AFR for all 32 layers and compare the base model with the checkpoints during training:
> > > >
> > > > - The **average AFR over all layers** increases from **5.54 (base)** to **5.64 (epoch 4)**, a relative gain of about **+1.8%**, indicating a moderate but consistent overall strengthening of attention to key object regions.
> > > > - When we group layers into low / mid / high segments, the average AFR changes are:
> > > >   - **Low**: 0.75 → 0.79 → 0.79 → 0.78 → 0.78 (absolute change ≈ **+0.03**)
> > > >   - **Mid**: 7.05 → 7.03 → 7.10 → 7.12 → 7.16 (absolute change ≈ **+0.11**)
> > > >   - **High**: 7.20 → 7.30 → 7.32 → 7.32 → 7.33 (absolute change ≈ **+0.13**),
> > > >   showing that the largest absolute gains appear in the mid/high segments, while the low segment fluctuates slightly around its base value.
> > > > - Importantly, the overall shape of the AFR curve over layers (i.e., which layers act as peaks and valleys) remains almost unchanged (Figure 4). For the LLaVA architecture, certain “visual-semantic layers” (e.g., layers 10–15, especially layer 14) already had high AFR values in the base model. Under our method, these layers remain the AFR peaks, but their peak values are further amplified. For example,
> > > >   - **Layer 14**: AFR increases from **14.74 → 17.10 → 18.02 → 18.44 → 18.72**, a relative gain of about **+27%**. Notably, layer 14 already exhibits the highest AFR in the base model and acts as one of the “visual–semantic” peak layers in LLaVA; under our method, this existing peak is further sharpened rather than shifted, meaning that the layer that was originally most responsible for focusing on core object regions becomes even more concentrated on ground-truth boxes and less on background noise during answer generation.
> > > >
> > > > These layer-wise statistics and AFR analyses together show that our method preserves the original hierarchical division of labor across layers, while gently but directionally reallocating attention from shallower layers and background regions toward mid/high layers and ground-truth object regions. Rather than collapsing the architecture into an “always-look-at-image” regime, the model sharpens the layers that were already responsible for semantic visual grounding.
> > > >
> > > > From the training perspective, this behavior is also compatible with how our **Calibration Direct Preference Optimization objective** is designed to shape the updates. In particular, the Focus-and-Enhance preference pairs and the composite focus loss \(\mathcal{L}_{\text{focus}} = \mathcal{L}_{\text{dpo\_focus}} + \lambda_{\text{calib}} \mathcal{L}_{\text{Calib}}\) explicitly prefer responses whose probability increases when the image is enhanced or cropped to the key region (Sec.~3.2). As discussed in our VID-based analysis, this calibration term encourages the model to attribute preference differences to the visual intervention, which in turn tends to increase the gradient signal on vision-dominant parameters and bias updates toward weights that mediate visual processing rather than purely language-only components.
> > > >
> > > > Taken together, the observed redistribution of attention toward mid/high layers and ground-truth object regions is consistent with this optimization picture: the calibration loss encourages the model to rely more on informative visual cues in its on-policy preferences, and the layer-wise/AFR statistics provide complementary evidence that this pressure is reflected in stronger, more focused visual grounding in the internal attention patterns. These changes are also in line with the reductions in hallucination reported in the main results.

---

> > > > > ### Author Response · Authors · 2025-11-20
> > > > > **## Reply for the second weakness (Part 1/5)**
> > > > >
> > > > > We appreciate the reviewer’s comment that our current draft lacks comparison with other self-generated data methods similar to P²-DPO. In this section, we (1) highlight the distinctive aspects of our method compared with recent self-generated data approaches, (2) provide a focused comparison with the CSR method mentioned by the reviewer, and (3) present empirical results comparing our method and CSR on hallucination benchmarks, together with an experimental analysis of the visual patterns learned by CSR.
> > > > >
> > > > >
> > > > >
> > > > > ### 1. Self-Generated LVLM Hallucination Methods
> > > > >
> > > > > CSR proposes a self-rewarding framework where the LVLM both generates candidate responses and scores them with an internal reward. For a fixed image–text input, CSR uses sentence-level beam search to produce multiple responses and computes a scalar reward
> > > > >
> > > > > $$
> > > > > R(s) = \lambda R_I(s) + (1 - \lambda) R_T(s)
> > > > > $$
> > > > >
> > > > > combining language likelihood $R_T$ and a CLIP-based image–text similarity $R_I$, and then selects the highest- and lowest-scoring responses as preferred and dispreferred for DPO training. The visual signal enters only via a global CLIPScore under a fixed image, and the reward operates at the level of coarse, response-level alignment rather than explicit visual interventions.
> > > > >
> > > > > Beyond CSR, prior self-generated LVLM methods such as SIMA and STIC also let the model “teach itself”, but mainly rely on model-as-critic natural-language judgements or relatively coarse heuristic manipulations of prompts/visual inputs, without explicitly modeling our two concrete Perceptual Processing failures or coupling visual interventions with an attention-based, key-region–aware training objective.
> > > > >
> > > > > IOur key viewpoint is that self-generated training is particularly effective for a specific subset of hallucinations—namely perceptual-processing failures where the model is already close to using the visual evidence correctly—while its impact on more fundamental perception errors is inherently limited. We distinguish hallucinations into two classes, Perception and Perceptual Processing failures: the former are generally difficult to resolve through self-evolution alone and typically require stronger external supervision, whereas the latter are exactly the cases where self-generated training is effective—hallucinations where the model has already captured the key visual evidence and is “one step away” from a correct answer. We therefore focus on failures such as the **Perceptual Bottleneck** in attended regions (where the attention map roughly localizes the right area but the answer is still wrong) and **Lack of Robustness** under mild image degradation (where small noise flips an otherwise correct prediction), which we regard as on-policy errors amenable to self-correction. Guided by this view, P²-DPO is built around a causal, visually grounded formulation: we design preference pairs by directly modifying the visual input (enhanced vs. locally erased regions, clean vs. noisy images) to target these two Perceptual Processing deficits, and couple them with a perceptual-targeted training architecture—Calibration Direct Preference Optimization, which combines a composite DPO objective, a Calibration Loss that maximizes Perceptual Confidence Gain and Visual Information Dependency, and Dynamic Deficit-Weighting—to explicitly redistribute probability mass toward visually grounded, attention-improved behavior.
> > > > >
> > > > > Building on this perspective, we explicitly focus on Perceptual Processing failures and, to the best of our knowledge, are among the first works that jointly design both a tailored data generation process and a matching training architecture for them. **In our view, self-generated training that does not distinguish which hallucinations are amenable to self-correction and which are not may spend a substantial portion of updates on cases that are very hard to fix in this way, thereby limiting its practical effectiveness.**
> > > > >
> > > > >
> > > > > ### 2. Detailed Comparison to CSR
> > > > >
> > > > > The reviewer specifically highlights CSR [1] and asks whether our method is essentially the same
> > > > > idea, since both use self-generated data and both target hallucinations in visual perception. We
> > > > > agree that CSR and our P²-DPO framework share a **high-level paradigm**—leveraging model
> > > > > self-generated preferences for LVLM alignment—but they differ fundamentally in:
> > > > >
> > > > > - how the problem is formulated,
> > > > > - how preference data are constructed,
> > > > > - which parameters are driven by the gradients, and
> > > > > - how theory informs the design.
> > > > >
> > > > > Below we articulate these differences.

---

> > > > > > ### Author Response · Authors · 2025-11-20
> > > > > > **## Reply for the second weakness (Part 2/5)**
> > > > > >
> > > > > > #### 2.1 Problem Formulation: Global Modality Misalignment vs. Perceptual Processing Failures
> > > > > >
> > > > > > CSR formulates the issue primarily as **global modality misalignment**: the LVLM tends to
> > > > > > prioritize textual priors over visual input, leading to image–text inconsistency. Its remedy is to
> > > > > > calibrate a self-reward by adding a global visual term (CLIPScore) on top of language likelihood, so
> > > > > > that high-reward responses are both more likely under the model and more compatible with the image.
> > > > > >
> > > > > >
> > > > > > In contrast, our work starts from a finer-grained **causal analysis of hallucination sources**. We further identify two concrete and overlooked
> > > > > > manifestations:
> > > > > >
> > > > > > 1. **Perceptual Bottleneck** in attended regions: the model’s attention does look at roughly the
> > > > > >    right area, but still produces hallucinated or under-specified content.
> > > > > > 2. **Lack of Robustness** to degraded visual input: small perturbations or degradations of the image
> > > > > >    cause the model to hallucinate or ignore the image.
> > > > > >
> > > > > > P²-DPO is explicitly designed to address these two **well-defined perceptual deficits**. This
> > > > > > problem formulation is narrower but more precise than CSR’s global modality-misalignment view, and it directly drives how we design data and objectives.
> > > > > >
> > > > > >
> > > > > >
> > > > > > #### 2.2 Preference Construction: Textual Re-Ranking vs. Visually-Grounded Causal Interventions
> > > > > >
> > > > > > CSR constructs preference pairs by **re-ranking textual candidates under a fixed visual input**:
> > > > > >
> > > > > > - The image and prompt are fixed.
> > > > > > - Multiple candidate responses are generated via sentence-level beam search.
> > > > > > - Each candidate is scored by
> > > > > >   $$
> > > > > >   R(s) = \lambda R_I(s) + (1-\lambda) R_T(s)
> > > > > >   $$
> > > > > >   where $R_I$ is a global CLIPScore and $R_T$ is language likelihood.
> > > > > > - Preferred / dispreferred responses are simply those with the highest / lowest scalar rewards.
> > > > > >
> > > > > > In other words, visual information enters only through a global similarity term inside the reward,
> > > > > > and the preference pairs are entirely between **texts under the same visual context**.
> > > > > >
> > > > > > Our P²-DPO framework takes a fundamentally different route: the preferences are generated by
> > > > > > **causal interventions on the visual input**, not just by re-ranking text under a fixed image.
> > > > > >
> > > > > > Concretely:
> > > > > >
> > > > > > - For **Perceptual Bottleneck**, we:
> > > > > >   - Use answer-to-image attention maps to localize salient regions.
> > > > > >   - Apply an **adaptive cropping algorithm** to extract a key region $I_{\text{crop}}$.
> > > > > >   - Construct two visual contexts:
> > > > > >     - An **enhanced input**
> > > > > >       $$
> > > > > >       I_{\text{aug}} = \text{Combine}(I, I_{\text{crop}})
> > > > > >       $$
> > > > > >       which augments the original image with the crop.
> > > > > >     - A **degraded input**
> > > > > >       $$
> > > > > >       I_{\text{deg}} = \text{Erase}(I, \text{Bbox}(I_{\text{crop}}))
> > > > > >       $$
> > > > > >       which selectively removes information in the salient region.
> > > > > >   - Generate responses $y_w^{\text{focus}}$ on $I_{\text{aug}}$ and $y_l^{\text{focus}}$ on $I_{\text{deg}}$, yielding Focus-and-Enhance preference pairs tailored to the bottleneck.
> > > > > >
> > > > > > - For **Lack of Robustness**, we:
> > > > > >   - Create a **noisy image**
> > > > > >     $$
> > > > > >     I_{\text{noise}} = \text{Noise}(I)
> > > > > >     $$
> > > > > >     and generate a low-fidelity response $y_l^{\text{rob}}$ on $I_{\text{noise}}$.
> > > > > >   - Generate an initial response on the clean image and refine it via **Contrastive Amplification**,
> > > > > >     treating the clean-image model as expert and the noisy-image model as amateur,
> > > > > >     to obtain a robust, visually faithful response $y_w^{\text{rob}}$.
> > > > > >   - This yields Visual Robustness preference pairs $(y_w^{\text{rob}}, y_l^{\text{rob}})$.
> > > > > >
> > > > > >
> > > > > > These constructions are **visually grounded and causal** (see the theoretical analysis in Sec. 3.1):
> > > > > >
> > > > > > - Preferences are defined between responses conditioned on **different visual inputs** (enhanced vs.degraded, clean vs. noisy), not just different texts under the same image.
> > > > > > - The direction of preference (which side is the winner) is determined by the **known direction of
> > > > > >   visual information quality**, not by a scalar reward whose behavior is only indirectly related to
> > > > > >   hallucination.
> > > > > >
> > > > > > We formalize this as **Visually-Grounded Contrastive Preference Generation (VCPG)**. CSR does not perform such structured visual interventions; it reuses a fixed image and relies on a global CLIPScore to bias text ranking.

---

> > > > > > > ### Author Response · Authors · 2025-11-20
> > > > > > > **## Reply for the second weakness (Part 3/5)**
> > > > > > >
> > > > > > > #### 2.3 Optimization Objective: Single Rewarded DPO vs. Perceptual-Targeted, Multi-Branch Objective
> > > > > > >
> > > > > > > CSR uses a single self-rewarded DPO objective: DPO is applied on preference pairs where the winner
> > > > > > > is determined by the scalar reward $R(s)$. The visual and textual aspects of the reward are
> > > > > > > entangled in that scalar, and the gradient does not explicitly distinguish between different
> > > > > > > perceptual failure modes.
> > > > > > >
> > > > > > > In contrast, P²-DPO employs a **structured, multi-branch objective tailored to perceptual
> > > > > > > processing**:
> > > > > > >
> > > > > > > - A **Focus-and-Enhance branch** $\mathcal{L}_{\text{focus}}$ built on $(y_w^{\text{focus}}, y_l^{\text{focus}})$, consisting of:
> > > > > > >   - A standard DPO loss on the Focus-and-Enhance pairs, and
> > > > > > >   - A **Calibration Loss** derived from the Perceptual Confidence Gain
> > > > > > >     $$
> > > > > > >     \Delta \pi_\theta(y)
> > > > > > >     = \log \pi_\theta(y \mid I_{\text{aug}})
> > > > > > >     - \log \pi_\theta(y \mid I_{\text{deg}})
> > > > > > >     $$
> > > > > > >     which we prove is equivalent to maximizing the Visual Information Dependency (VID) of the
> > > > > > >     winning response.
> > > > > > >
> > > > > > > - A **Visual Robustness branch** $\mathcal{L}_{\text{rob}}$ on robustness pairs
> > > > > > >   $(y_w^{\text{rob}}, y_l^{\text{rob}})$, formulated to explicitly encourage the model to
> > > > > > >   “see through” noise and prefer visually faithful answers even when conditioned on degraded images.
> > > > > > >
> > > > > > > - A **Dynamic Deficit-Weighting (DDW)** mechanism, which:
> > > > > > >   - Diagnoses, per sample, whether the main issue is bottleneck vs. robustness by comparing
> > > > > > >     CLIP-like scores of the crop vs. the full image,
> > > > > > >   - And *dynamically reweights* $\mathcal{L}_{\text{focus}}$ and $\mathcal{L}_{\text{rob}}$ so that
> > > > > > >     the model allocates more capacity to the dominant deficit.
> > > > > > >
> > > > > > > As a result, P²-DPO does not simply “reward visually consistent answers more”; it **allocates
> > > > > > > gradient budget explicitly toward resolving two concrete perceptual deficits**, via tailored
> > > > > > > preference branches and a calibration term that ties probability changes to visual evidence.
> > > > > > >
> > > > > > > CSR does not decompose the objective in this way; it optimizes a single, global reward that does
> > > > > > > not distinguish between bottleneck vs. robustness failures.
> > > > > > >
> > > > > > >
> > > > > > > #### 2.4 Theoretical Grounding: Reward-Level Intuition vs. Visual-Parameter-Level Analysis
> > > > > > >
> > > > > > > CSR provides an intuitive and partially formal justification that adding a visual term to the
> > > > > > > self-reward improves modality alignment in an iterative self-training loop. The analysis is at the
> > > > > > > reward level: calibrating a step-wise reward with visual constraints is argued to be beneficial for
> > > > > > > alignment under mild assumptions.
> > > > > > >
> > > > > > > Our work goes a step deeper by analyzing **how preference data properties affect the gradients on
> > > > > > > vision-dominant parameters**. We:
> > > > > > >
> > > > > > > - Explicitly decompose parameters into
> > > > > > >   $\theta_1$ (vision-dominant) and $\theta_2$ (language-dominant).
> > > > > > > - Show that the DPO gradient on $\theta_1$ is driven by
> > > > > > >   $$
> > > > > > >   \Delta(\theta_1)
> > > > > > >   = \nabla_{\theta_1}\log \pi_\theta(y_w)
> > > > > > >   - \nabla_{\theta_1}\log \pi_\theta(y_l)
> > > > > > >   $$
> > > > > > >
> > > > > > >   and that the *expected norm* $\mathbb{E}\big\|\Delta(\theta_1)\big\|$ is controlled by the
> > > > > > >   **Visual Information Dependency (VID)** of the texts.
> > > > > > > - Show, via an Information Bottleneck argument, that
> > > > > > >   $$
> > > > > > >   \big\|\nabla_{\theta_1}\log \pi_\theta(y)\big\|
> > > > > > >   \propto \text{VID}(y)
> > > > > > >   $$
> > > > > > >   and therefore that classical post-hoc semantic correction strategies (where $y_w$ and $y_l$ share
> > > > > > >   similar visual features) tend to cancel gradients in the visual subspace.
> > > > > > > - Introduce **Visual Information Disparity** (VID$(y_w) \gg$ VID$(y_l)$) as a design principle
> > > > > > >   for VCPG, and prove that such pairs maximize both the gradient norm on $\theta_1$ and the
> > > > > > >   **Fisher Information** of visual parameters:
> > > > > > >   $$
> > > > > > >   \text{Tr}(F_{\theta_1}) \propto \mathbb{E}\left[
> > > > > > >     D_{\mathrm{KL}}\big(p(f_v \mid y_w) \,\|\, p(f_v \mid y_l)\big)
> > > > > > >   \right].
> > > > > > >   $$
> > > > > > >
> > > > > > >
> > > > > > >
> > > > > > > We also analyze **on-policy vs. off-policy** preference data under DPO’s KL constraint and show
> > > > > > > that strictly off-policy preferences can lead to vanishing gradients (the sigmoid weight tends to
> > > > > > > zero when the preferred response lies outside the support of the reference model).
> > > > > > >
> > > > > > > This theoretical analysis does not just say “adding visual information is helpful”; it explains
> > > > > > > **which kinds of self-generated preference pairs are most effective for training visual parameters**
> > > > > > > and why. CSR does not provide this level of parameter-space analysis or the VID/Fisher perspective.
> > > > > > >
> > > > > > > In summary, while CSR and P²-DPO share the umbrella idea of self-generated preferences for
> > > > > > > hallucination mitigation, they differ substantively in problem formulation, preference construction,
> > > > > > > objective design, and theoretical grounding.

---

> > > > > > > > ### Author Response · Authors · 2025-11-20
> > > > > > > > **## Reply for the second weakness (Part 4/5)**
> > > > > > > >
> > > > > > > > ### 3. Hallucination Evaluation of CSR and Analysis of Its Attention Patterns
> > > > > > > >
> > > > > > > > #### 3.1 Hallucination Evaluation
> > > > > > > >
> > > > > > > > Since the reviewer explicitly highlighted CSR as a closely related method, we also evaluate the released three-iteration CSR-LLaVA checkpoint under exactly the same settings as in our experiments (same base architecture, datasets, and decoding configuration). We report results on four hallucination-focused benchmarks: POPE (adversarial / popular / random splits), HallusionBench, MMHal-Bench, and AMBER, comparing the LLaVA-1.5-7B base model, CSR, and our P²-DPO model.
> > > > > > > >
> > > > > > > > **Table R4. Hallucination performance of LLaVA-1.5-7B base, CSR, and P²-DPO across four benchmarks.**
> > > > > > > > | Model                | POPE Adv (F1 ↑) | POPE Pop (F1 ↑) | POPE Rand (F1 ↑) | POPE Avg (F1 ↑) | HallusionBench qAcc (↑) | HallusionBench fAcc (↑) | HallusionBench aAcc (↑) | MMHal-Bench Hal (↓) | MMHal-Bench Score (↑) | AMBER CHAIRi (↓) | AMBER Hal (↓) | AMBER F1_E (↑) | AMBER F1_A (↑) | AMBER F1_R (↑) |
> > > > > > > > |----------------------|----------------:|----------------:|-----------------:|----------------:|-------------------------:|-------------------------:|-------------------------:|--------------------:|-----------------------:|------------------:|---------------:|----------------:|----------------:|----------------:|
> > > > > > > > | LLaVA-1.5-7B (base)  | 81.8            | 84.36           | 89.12            | 85.09           | 13.90                    | 20.23                    | 48.16                    | 0.62               | 1.97                  | 7.8              | 36.4           | 83.2            | 64.1            | 62.4            |
> > > > > > > > | CSR-LLaVA (3 iters)  | 82.0            | 86.81           | 89.21            | 86.01           | 16.04                    | 23.41                    | 45.35                    | 0.59               | 2.26                  | 6.2              | 30.7           | 86.0            | 65.6            | 67.8            |
> > > > > > > > | P²-DPO (ours)        | **84.53**           | **87.91**           | **89.87**            | **87.44**           | **26.37**                    | **30.05**                    | **55.62**                    | **0.56**               | **2.43**                  | **5.9**              | **26.7**           | **92.6**            | **71.7**            | **70.9**            |
> > > > > > > >
> > > > > > > > Compared to the LLaVA base model, CSR brings moderate gains on some hallucination metrics (e.g., slightly higher POPE F1 and improved AMBER Hal / CHAIRi). However, P²-DPO consistently outperforms both the base model and CSR across all four benchmarks:
> > > > > > > >
> > > > > > > > - On **POPE**, P²-DPO improves the average F1 from 85.09 (base) and 86.01 (CSR) to 87.44.
> > > > > > > > - On **HallusionBench**, P²-DPO substantially improves all three accuracies (qAcc / fAcc / aAcc), e.g., qAcc from 13.90 (base) and 16.04 (CSR) to 26.37.
> > > > > > > > - On **MMHal-Bench**, P²-DPO further reduces hallucination rate (Hal ↓) and increases the overall MMHal score.
> > > > > > > > - On **AMBER**, P²-DPO achieves lower CHAIRi and Hal (fewer hallucinations) while significantly boosting F1\_E / F1\_A / F1\_R.
> > > > > > > >
> > > > > > > > These results indicate that while CSR’s self-rewarding scheme can yield some global improvements, our perceptual-processing–targeted design is more effective at reducing hallucinations across diverse, fine-grained benchmarks.
> > > > > > > >
> > > > > > > >
> > > > > > > > ### 3.2 Attention Pattern Analysis
> > > > > > > >
> > > > > > > > The reviewer also asked about attention patterns. Since CSR is explicitly designed to “calibrate” rewards with a CLIP-based visual term, it is natural to ask whether it actually strengthens the model’s visual attention. To answer this, we apply the same attention analysis as in our response to Weakness 2, not only to our P²-DPO model but also to CSR, using the LLaVA-1.5-7B base model as a common reference.
> > > > > > > >
> > > > > > > > We report two summary statistics (details of the computation follow exactly our previous analysis):
> > > > > > > >
> > > > > > > > - The **mean image ratio**, i.e., the average fraction of attention mass assigned to image tokens for answer tokens (see Table R6 below).
> > > > > > > > - The **Attention Focus Ratio (AFR)** over all 32 layers, measuring how strongly attention concentrates on key object regions (see Table R5 below).
> > > > > > > >
> > > > > > > > For our P²-DPO model, we observe:
> > > > > > > >
> > > > > > > > - The **mean image ratio** increases from **0.1971 (base)** to **0.2055 (epoch 4)**, a relative gain of about **+4.3%**, indicating that answer tokens pay more attention to visual evidence.
> > > > > > > > - The **average AFR over all layers** increases from **5.54 (base)** to **5.64 (epoch 4)**, a relative gain of about **+1.8%**, reflecting a moderate but consistent strengthening of attention to key object regions.
> > > > > > > >
> > > > > > > > In contrast, for CSR we find the opposite trend:
> > > > > > > >
> > > > > > > > - The **mean image ratio** slightly **decreases** from **0.1971 (base)** to **0.1962 (after 3 iterations)**, corresponding to a small **−0.4%** relative change.
> > > > > > > > - The **average AFR over all layers** **drops** from **5.54 (base)** to **5.01 (CSR)**, a relative change of about **−9.6%**, indicating a noticeable weakening of attention concentration on key visual regions.

---

> > > > > > > > > ### Author Response · Authors · 2025-11-20
> > > > > > > > > **## Reply for the second weakness (Part 5/5)**
> > > > > > > > >
> > > > > > > > > In other words, under the same base architecture and evaluation protocol, our P²-DPO model not only reduces hallucinations quantitatively, but also shifts attention in a direction that is more visually grounded (higher image ratio and AFR). Under our reproduction setting (same base LLaVA-1.5-7B architecture and our evaluation protocol), CSR, despite using a CLIPScore-based visual term in its reward, does not exhibit an improvement in these visual attention metrics and even appears to dilute attention on key visual regions.
> > > > > > > > >
> > > > > > > > > This empirical observation is consistent with our critique in Sec. 2: a coarse, global CLIPScore-based reward provides only weak and indirect supervision for the vision pathway, whereas our visually grounded, deficit-targeted design is explicitly constructed to enhance the visual component of the model’s processing.
> > > > > > > > >
> > > > > > > > >
> > > > > > > > >
> > > > > > > > > **Table R5. Global image attention ratios for LLaVA-1.5-7B base vs CSR (3 iterations).**
> > > > > > > > > | Checkpoint          | Mean `image_sum_ratio` (over all samples) | Overall visual proportion `image_sum / (image_sum + text_sum)` |
> > > > > > > > > |---------------------|-------------------------------------------|-----------------------------------------------------------------|
> > > > > > > > > | Base                | 0.197109                                  | 0.220413                                                        |
> > > > > > > > > | CSR (3 iters)       | 0.196241                                  | 0.212582                                                        |
> > > > > > > > >
> > > > > > > > > **Table R6. Layer-wise image attention ratios for LLaVA-1.5-7B base vs CSR (3 iterations).**
> > > > > > > > > | Layer | Base    | CSR (3 iters) |
> > > > > > > > > |------|---------|---------------|
> > > > > > > > > | 0    | 0.582635 | 0.592526      |
> > > > > > > > > | 1    | 0.366500 | 0.389600      |
> > > > > > > > > | 2    | 0.190149 | 0.194281      |
> > > > > > > > > | 3    | 0.196239 | 0.190109      |
> > > > > > > > > | 4    | 0.191965 | 0.194722      |
> > > > > > > > > | 5    | 0.203474 | 0.206354      |
> > > > > > > > > | 6    | 0.089091 | 0.089655      |
> > > > > > > > > | 7    | 0.084844 | 0.086783      |
> > > > > > > > > | 8    | 0.101540 | 0.108352      |
> > > > > > > > > | 9    | 0.101719 | 0.111033      |
> > > > > > > > > | 10   | 0.111041 | 0.117952      |
> > > > > > > > > | 11   | 0.159806 | 0.166729      |
> > > > > > > > > | 12   | 0.179874 | 0.182286      |
> > > > > > > > > | 13   | 0.129479 | 0.131434      |
> > > > > > > > > | 14   | 0.245604 | 0.243502      |
> > > > > > > > > | 15   | 0.126483 | 0.119460      |
> > > > > > > > > | 16   | 0.081543 | 0.079110      |
> > > > > > > > > | 17   | 0.181153 | 0.173885      |
> > > > > > > > > | 18   | 0.176871 | 0.166487      |
> > > > > > > > > | 19   | 0.211094 | 0.202387      |
> > > > > > > > > | 20   | 0.276292 | 0.270493      |
> > > > > > > > > | 21   | 0.253457 | 0.239840      |
> > > > > > > > > | 22   | 0.219176 | 0.206638      |
> > > > > > > > > | 23   | 0.220085 | 0.208693      |
> > > > > > > > > | 24   | 0.207460 | 0.186334      |
> > > > > > > > > | 25   | 0.225522 | 0.222806      |
> > > > > > > > > | 26   | 0.268594 | 0.261078      |
> > > > > > > > > | 27   | 0.092683 | 0.093636      |
> > > > > > > > > | 28   | 0.186419 | 0.184947      |
> > > > > > > > > | 29   | 0.196246 | 0.192231      |
> > > > > > > > > | 30   | 0.105821 | 0.105596      |
> > > > > > > > > | 31   | 0.190546 | 0.186405      |

---

> > > > > > > > > > ### Author Response · Authors · 2025-11-27
> > > > > > > > > > **Checking In Regarding Our Discussion**
> > > > > > > > > >
> > > > > > > > > > Dear Reviewer,
> > > > > > > > > > I hope this message finds you well. As the discussion period is nearing its end with less than a week remaining, I wanted to ensure that we have addressed all of your concerns satisfactorily. If there are any additional points or feedback you would like us to consider, please let us know. Your insights are invaluable to us, and we are eager to address any remaining issues to further improve our work.
> > > > > > > > > > Thank you for your time and effort in reviewing our paper.
> > > > > > > > > > Wishing you all the best, and I sincerely hope your own work progresses smoothly as well.

---

### Official Review · Reviewer_m9BD · 2025-11-05

**Soundness:** 3
**Presentation:** 3
**Contribution:** 2
**Rating:** 4
**Confidence:** 4

**Summary:**

This paper introduces P²-DPO, a self-correcting framework for reducing hallucination in Large Vision-Language Models. It targets perceptual processing bottlenecks rather than perception failures, generating on-policy, vision-aware preference pairs to improve attention and robustness. With a Calibration Loss and Dynamic Deficit-Weighting, P²-DPO enhances visual grounding and robustness, outperforming human- and AI-feedback DPO baselines without external supervision.

**Strengths:**

- The paper provides a novel perspective on hallucination in LVLMs by distinguishing between perception failure and perceptual processing failure. It highlights the latter as an overlooked yet solvable issue that can be addressed through self-correction within the model, offering new insight into the root causes of hallucination.

- The proposed on-policy strategy, which generates vision-aware contrastive preference pairs from the model itself, effectively removes the dependence on human or AI feedback used in traditional DPO frameworks.

**Weaknesses:**

- Limited experimental scope: The experiments are primarily conducted on LLaVA-1.5-7B and Qwen2.5-VL-3B, but it is unclear why the authors did not include the more commonly used Qwen2.5-VL-7B model. This omission limits the completeness of the evaluation and raises questions about the method’s scalability and consistency across model sizes.

- Potential self-reinforcement bias in on-policy data: Although the on-policy preference generation strategy avoids the off-policy issue, self-generated preference pairs in the early training stage may inherit the model’s existing biases or hallucinations.

- Lack of qualitative interpretation of improvements: While quantitative results are comprehensive, the paper lacks intuitive visual evidence explaining how the Calibration Loss and Dynamic Deficit-Weighting improve the model’s attention distribution or semantic grounding. Visualizations such as attention map changes would make the effectiveness of these mechanisms clearer and more convincing.

**Questions:**

Please refer to the weakness part.

---

> ### Author Response · Authors · 2025-11-20
> **## Reply for the first question**
>
> We thank the reviewer for pointing this out and sincerely acknowledge this oversight on our side. Most of our existing experiments were indeed conducted on LLaVA-1.5-7B. To jointly study both cross-family generalization and the effect of model size, we initially chose Qwen2.5-VL-3B as the additional backbone, hoping that a single extra configuration could simultaneously demonstrate cross-family transfer and size robustness. However, as the reviewer correctly notes, this design still leaves the evaluation on the more standard Qwen2.5-VL-7B incomplete. Following your suggestion, we have now added experiments on Qwen2.5-VL-7B and will include the corresponding results and analysis in the revised version of the paper. We are grateful for this helpful and important comment.
>
> **Table.** Performance of our method on Qwen2.5-VL backbones, addressing the reviewer's concern about Qwen2.5-VL.  Higher is better for F1, Acc, and Score (↑), lower is better for Hal and CHAIRi (↓).
>
> | Model                | Pairs Source | POPE Adv (F1 ↑) | POPE Pop (F1 ↑) | POPE Rand (F1 ↑) | POPE Avg (F1 ↑) | HallusionBench qAcc (↑) | HallusionBench fAcc (↑) | HallusionBench aAcc (↑) | MMHal-Bench Hal (↓) | MMHal-Bench Score (↑) | AMBER CHAIRi (↓) | AMBER Hal (↓) | AMBER F1_E (↑) | AMBER F1_A (↑) | AMBER F1_R (↑) |
> |----------------------|-------------:|----------------:|----------------:|-----------------:|----------------:|-------------------------:|-------------------------:|-------------------------:|--------------------:|-----------------------:|------------------:|---------------:|----------------:|----------------:|----------------:|
> | Qwen2.5-VL-3B        | Base         | 86.26           | 88.05           | 89.79            | 88.03           | 28.13                    | 34.10                    | 58.10                    | 41.67               | 3.38                  | 6.8              | 39.1           | **93.2**        | 83.7           | 77.9           |
> | + DPO_RLHF-V         | Human        | 86.41           | **88.59**       | 90.40            | 88.46           | 29.89                    | 33.53                    | **59.08**               | 39.74               | 3.41                  | 6.8              | 39.5           | 93.0            | 83.5           | 78.1           |
> | **+ P²-DPO (Ours)**  | Self         | **86.88**       | 88.48           | **91.50**        | **88.95**       | **30.11**                | **34.68**                | 58.64                    | **38.54**           | **3.49**              | **6.6**          | **38.2**       | 92.7           | **84.2**       | **80.9**       |
> | Qwen2.5-VL-7B        | Base         | 84.92           | 86.78           | 89.01            | 86.90           | 31.87                    | 37.57                    | 61.03                    | 34.96               | 3.47                  | 5.4              | 29.0           | 97.3           | **83.8**           | 72.6           |
> | + DPO_RLHF-V         | Human        | 84.99           | 86.81           | 89.72            | 87.17           | 34.95                    | 39.60                    | 62.62                    | 34.58               | 3.63                  | 4.2              | 29.3           | **98.5**        | 82.5           | 72.2           |
> | **+ P²-DPO (Ours)**  | Self         | **86.79**       | **87.79**       | **90.05**        | **88.21**       | **36.70**                | **41.62**                | **65.19**               | **32.29**           | **3.94**              | **3.9**          | **24.9**       | 98.5        | 83.4           | **72.9**       |

---

> ### Author Response · Authors · 2025-11-20
> **## Reply for the second question (Part 1/4)**
>
> We first clarify what kinds of biases and data-generation errors may arise in our setting, and how this notion of “bias” differs from the one commonly discussed in traditional reinforcement learning. Then we explain our filtering mechanisms and present additional experiments that validate the effectiveness of this filtering; finally, we describe the design of our loss function, which further mitigates any residual biases that are not fully removed by the data filtering step.
>
> ### **Bias Analysis**
>
> 1. **No Iterative Self-Bias**
>    Similar concerns about self-reinforcement or confirmation bias have been extensively discussed in prior work on self-training and LLM-based annotation [1]. These works show that such bias typically emerges through *iterative* feedback loops, where a model repeatedly reuses its own predictions as training signals. In contrast, P²-DPO's resulting preferences are then used once for training, without feeding the updated policy back into the data-generation pipeline. In our case, although some bias in the self-generated preferences is inevitable, but behaves like moderate label noise, rather than leading to a runaway self-reinforcement effect.
>
> 2. **Visually Intervened Targets Instead of Reinforcing the Original Distribution**
>    In classical self-training setups, the model is often trained to imitate its own distribution
> $$
> \pi_{\text{ref}}(y \mid I, P)
> $$
> under the *same* input \((I, P)\), which makes it easy to fall into a self-reinforcement loop. In P²-DPO, we deliberately break this symmetry by first putting the reference model into different **visual states** for the same semantic input \((I, P)\). For example, we obtain a “strong-vision” distribution under enhanced evidence,
>
> $$
> p_{\text{strong}}(y) = \pi_{\text{ref}}(y \mid I_{\text{aug}}, P_{\text{enh}})
> $$
>
> and a “weak-vision” distribution under the original or degraded image,
>
> $$
> p_{\text{weak}}(y) = \pi_{\text{ref}}(y \mid I, P)
> \quad \text{or} \quad
> \pi_{\text{ref}}(y \mid I_{\text{deg}}, P)
> $$
>
> The preference pairs are then constructed so that the **strong-vision response** (e.g., \(y_{\text{focus}}^w\) from \(p_{\text{strong}}\)) is treated as the winner, and our training objective explicitly encourages the policy
> $$
> \pi_\theta(\cdot \mid I, P)
> $$
> to move closer to this strong-vision behavior. In other words, we are not simply reinforcing the original distribution
> $$
> \pi_{\text{ref}}(\cdot \mid I, P)
> $$
> instead, we use visually *intervened* distributions as targets and align the weak-vision state to the strong-vision state, thereby reducing self-reinforcement on the original \((I, P)\) distribution.
>
> 3. **Residual Bias from the Guidance Mechanism**
>
>    From this perspective, the core risk in our method is not that “ordinary model biases will be automatically amplified,” but rather whether the **guidance mechanism** is correct. Since our preference pairs are defined via visual interventions (crops, erasures, noise) that distinguish strong-vision from weak-vision states, any error in this guidance—such as mislocalized attention leading to a poor crop, or noise that unintentionally alters the image semantics—can introduce incorrect supervision signals for individual samples.
>
>
> ### **Filtering out low-quality data**
>
> Even with the above structural design, mis-specified guidance can still introduce residual noisy preferences, in particular when (i) aggressive visual perturbations (crops, occlusions, noise) destabilize the language generation itself, leading to disfluent or degenerate text, or (ii) the reference model assigns almost indistinguishable or extremely skewed probabilities to the two responses, making the resulting preference a poor learning signal. Our quality filtering mechanism that is an additional safety gate (see Appendix *Data Filtering Implementation Details* for detailed statistics). For each raw preference pair, we compute two metrics with a frozen reference model $(\pi_{\text{ref}})$: a fluency-based perplexity (PPL) score for both responses, and a log-probability margin between the winning and losing responses, which are then used to discard low-quality or uninformative pairs.

---

> > ### Author Response · Authors · 2025-11-20
> > **## Reply for the second question (Part 2/4)**
> >
> > - **Perplexity (PPL).**
> > This measures the fluency and grammatical plausibility of the winning/losing answers. For each sample we take
> > $$
> > \max(\text{PPL}(y_w), \text{PPL}(y_l)),
> > $$
> > and set an upper threshold based on the empirical distribution. Samples exceeding this threshold are treated as low-quality text and directly discarded.
> >
> > - **Log-probability Margin (Log-Prob Margin).**
> > This captures the “learnability” of the preference signal:
> > $$
> > M = \log p_{\text{ref}}(y_w) - \log p_{\text{ref}}(y_l).
> > $$
> >
> > Intuitively, if attention-guided cropping truly fails and falls on noise or irrelevant background, the resulting answers typically behave as follows:
> >
> > - either the semantics are off-topic or incoherent, leading to clearly elevated PPL;
> > - or the reference model assigns an extreme probability gap between the two answers, pushing the sample into the tail of the Margin distribution.
> >
> > Such samples are very likely to be removed by the above PPL + Margin filtering and thus do not enter the training set. Therefore, even if a small number of severe attention failures occur, they are unlikely to silently accumulate at the data level, which substantially reduces the risk of reinforcing spurious regions or noise.
> >
> >
> > **Additional experiment: Verifying the effectiveness of filtering**
> >
> > Our preference pairs are constructed from the prompts and images in the RLHF-V dataset. During training, we only use the prompt + image and **do not** use the human preference annotations provided by RLHF-V. To validate the quality of our self-constructed preference pairs and to check whether the PPL + Margin filter truly identifies “problematic pairs”, we design an additional control experiment that uses RLHF-V’s human high-/low-quality answers as an external evaluation standard.
> >
> > **Construction of an evaluation task with human references.**
> > In RLHF-V, each sample, in addition to the image and question, is accompanied by:
> >
> > - a high-quality human answer (High-Quality Answer, denoted $A_{\text{HQ}}$), and
> > - a low-quality human answer (Low-Quality Answer, denoted $A_{\text{LQ}}$).
> >
> > In this additional experiment, these human annotations are used only as *evaluation references* and do not participate in training. For each self-constructed preference pair $(y_w, y_l)$, we build an evaluation prompt (i.e., `PROMPT_TEMPLATE`) that contains:
> >
> > - **[Question]**: the original question;
> > - **High-Quality Answer**: the human high-quality answer as a golden reference;
> > - **Low-Quality Answer**: the human low-quality answer as a negative reference;
> > - **Model Response A/B**: our generated $y_w$ and $y_l$, randomly assigned to A and B.
> >
> > We then use a strong general-purpose language model (Qwen3-Max) to assign a quality score in $\{1, 2, 3\}$ to A and B respectively, based on their semantic proximity to the high-/low-quality human answers:
> >
> > - **3 points**: semantically close to the High-Quality Answer, with only minor descriptive differences, and correctly capturing the main idea;
> > - **2 points**: basically reasonable but not fully aligned, e.g., missing some information or adding harmless verbosity;
> > - **1 point**: closer to the Low-Quality Answer, or clearly wrong / hallucinatory.
> >
> > **Defining invalid samples under the “standard-answer filter”.**
> > Within this framework, we label a preference pair as an *invalid sample that should be removed* if either of the following holds:
> >
> > - the answer we mark as the winner, $y_w$, receives the lowest score (1 point), indicating that it is semantically closer to the human low-quality answer;
> > - or $y_w$ receives a lower score than $y_l$, indicating that the evaluator believes the true preference direction is opposite to our constructed win/lose relation.
> >
> > This step depends only on the question + human high/low-quality answers + our generated responses, and is completely independent of PPL, Margin, or attention maps. It therefore acts as an independent, semantic-level “standard-answer filter”.
> >
> > **Quantitative comparison with PPL + Margin filtering.**
> > According to this standard-answer filter:
> >
> > - the Qwen evaluator marks **510** preference pairs as invalid samples;
> > - our PPL + Margin filtering alone removes **287** samples;
> > - the intersection between these two sets contains **236** samples.
> >
> > Thus, among the 287 samples removed by PPL + Margin, we have
> > $$
> > \frac{236}{287} \approx 82\%
> > $$
> > that are *also* judged invalid by the independent standard-answer filter (Qwen + human high/low-quality answers).
> >
> > This leads to two key observations:

---

> > > ### Author Response · Authors · 2025-11-20
> > > **## Reply for the second question (Part 3/4)**
> > >
> > > - **High overlap indicates that our rules are not arbitrary.**
> > > Among the samples removed by PPL + Margin, more than 80% are simultaneously flagged as problematic by an independent evaluator that explicitly compares against human high-quality / low-quality answers. This suggests that the PPL + Margin filter is not randomly pruning examples, but is highly consistent with a more semantic, human-anchored evaluation criterion.
> > >
> > > - **Our thresholds are intentionally conservative.**
> > > Although the standard-answer filter identifies 510 suspicious samples in total, we only remove a subset of them (287 samples) via PPL + Margin and retain roughly 95% of the overall data. In other words, we deliberately let PPL + Margin play the role of a **“fallback cleaner for extremely bad samples”**, rather than aggressively deleting all borderline cases, thereby balancing data quality and diversity.
> > >
> > > Therefore, this additional experiment, from an attention-independent and human-anchored external perspective, confirms that our PPL + Margin filtering strategy can indeed identify and remove a large portion of low-quality or preference-direction-incorrect samples, while remaining relatively conservative in scale. This provides strong evidence that it is a reliable fallback filtering mechanism.
> > >
> > > ### Down-weighting bias in the loss function
> > > To address the potential distributional bias introduced when the model incorrectly crops the supposed key region, and because the filtering procedure cannot completely remove all low-quality samples, we explicitly introduce a CLIPScore-based dynamic weighting mechanism into the loss function.
> > >
> > >
> > > Specifically, the model simultaneously constructs two preference signals: a focus preference from the attention-guided cropped region
> > >
> > > $\mathcal{L}_{\text{focus}}$
> > >
> > > and a global robust preference from the full original image
> > >
> > > $\mathcal{L}_{\text{dpo-rob}}$.
> > >
> > > For each sample, we first use a pretrained CLIP model to compute a Perceptual Gain Ratio $r$, which measures whether cropping actually brings a semantic gain:
> > >
> > > $$
> > > r = \frac{\mathrm{CLIPScore}(P, I_{\text{crop}})}{\mathrm{CLIPScore}(P, I)} ,
> > > $$
> > >
> > > where $P$ is the text prompt, and $I$ and $I_{\text{crop}}$ denote the original image and the cropped image, respectively. If $r > 1$, it indicates that the alignment between the image and the prompt is improved after cropping (the main issue is insufficient focus); conversely, if $r < 1$, it indicates that cropping has damaged the original semantics (the main issue is lack of robustness / erroneous cropping).
> > >
> > > We then map $r$ into a smooth weighting factor
> > >
> > > $$
> > > \alpha = \alpha_{\text{max}} \cdot \tanh\left(\frac{r - 1.0}{\tau}\right),
> > > $$
> > >
> > > and use it to dynamically allocate weights between the two loss branches:
> > >
> > > $$
> > > w_{\text{focus}} = w_{\text{base}} + \alpha, \quad
> > > w_{\text{robust}} = w_{\text{base}} - \alpha.
> > > $$
> > >
> > > The final unified objective is
> > >
> > > $\mathcal{L}_{\text{total}}$ =$\mathbb{E}$ [
> > >
> > > $w_{\text{focus}} \cdot \mathcal{L}_{\text{focus}}$ +
> > >
> > > $w_{\text{robust}} \cdot \mathcal{L}_{\text{dpo-rob}}$
> > >
> > > ]

---

> > > > ### Author Response · Authors · 2025-11-20
> > > > **## Reply for the second question (Part 4/4)**
> > > >
> > > > Under this mechanism, if a particular crop mostly falls on irrelevant regions, we have $\mathrm{CLIPScore}(P, I_{\text{crop}}) < \mathrm{CLIPScore}(P, I)$, which yields $r < 1$ and thus $\alpha < 0$. As a result, $w_{\text{focus}}$ is automatically decreased and $w_{\text{robust}}$ is increased: the influence of preference pairs generated from erroneous crops is significantly down-weighted during training, while the global preference term $\mathcal{L}_{\text{dpo-rob}}$ is not affected by this kind of problem, preventing the model from being systematically pulled toward incorrect regions.
> > > >
> > > >
> > > > **Additional experiment: reasonableness of using CLIPScore as a dynamic weighting signal**
> > > > To more intuitively verify the reasonableness of using CLIPScore as a dynamic weighting signal, we design a controlled experiment. We randomly sample 500 examples from the training set and compute the CLIPScore between the prompt $P$ and the original image $I$, as well as between the prompt and the attention-guided cropped image $I_{\text{crop}}$. We define
> > > >
> > > >
> > > > $$
> > > > \text{diff} =
> > > > \text{CLIPScore}(P, I_{\text{crop}})-
> > > > \text{CLIPScore}(P, I).
> > > > $$
> > > > $$
> > > > \text{ratio} =
> > > > \frac{\text{CLIPScore}(P, I_{\text{crop}})}{\text{CLIPScore}(P, I)}.
> > > > $$
> > > >
> > > > At the same time, we construct a "reverse cropping" control group: for the same set of examples, we crop from the complementary (low-attention) region of the attention map to obtain $I_{\text{inv}}$, and compute the corresponding $\mathrm{diff}$ and $\mathrm{ratio}$ in the same way. We denote the attention-guided cropping group as `att_crop`, and the reverse cropping group as `att_inverse`.
> > > >
> > > > According to the statistics (see Table 1) and the distribution plots (Figure 1 for `improvement_diff` and Figure 2 for `improvement_ratio` in the PDF (in supplementary materials)) in the `att_crop` group, the mean of `improvement_ratio` is about $1.22$, and the median is about $1.16$, which indicates that, for the vast majority of samples, attention-guided cropping systematically improves the CLIPScore, yielding roughly a $20\%$ relative gain on average. The mean of `improvement_diff` is about $3.64$, and the median is about $3.09$, showing that the semantic similarity between the cropped image and the prompt is consistently increased for most samples.
> > > >
> > > > In contrast, in the `att_inverse` group, the mean of `improvement_ratio` is about $0.93$, and the median is about $0.93$, both lying in the region below $1$, which means that when we deliberately crop from regions not attended by the model, the CLIPScore decreases systematically in a statistical sense. The mean of `improvement_diff` is about $-1.76$, and the median is about $-1.29$, also exhibiting a clear negative shift.
> > > >
> > > > Although the standard deviations indicate that there is some variation across individual samples, the signs and magnitudes of the mean and median make the pattern clear: when cropping along high-response attention regions (`att_crop`), the improvement in CLIPScore is significant and directionally consistent; when we intentionally avoid attention regions and perform reverse cropping (`att_inverse`), the CLIPScore tends to decrease overall.
> > > >
> > > > These results show that CLIPScore is highly sensitive to whether the crop truly focuses on semantically relevant regions. When the crop aligns with high-response attention regions, most samples satisfy $\mathrm{ratio} > 1$ and $\mathrm{diff} > 0$; when the crop falls into regions complementary to attention, most samples satisfy $\mathrm{ratio} < 1$ and $\mathrm{diff} < 0$. Therefore, using `ratio` and `diff` to drive the dynamic weighting of $w_{\text{focus}}$ and $w_{\text{robust}}$ allows us, in a statistical sense, to amplify the contribution of "well-cropped" samples and suppress the impact of "badly cropped" samples, which is precisely the effect we aim to achieve with this mechanism.
> > > >
> > > > **Table 1: CLIPScore statistics for attention-guided vs. inverse cropping.**
> > > > | Group        | Metric             | Count | Mean     | Median   | Std      | Min        | Max        |
> > > > |-------------|--------------------|-------|----------|----------|----------|------------|------------|
> > > > | `att_crop`  | improvement_ratio  | 500   | 1.222227 | 1.160330 | 0.206298 | 1.060866   | 3.165598   |
> > > > | `att_crop`  | improvement_diff   | 500   | 3.636841 | 3.087014 | 1.847164 | 1.848955   | 12.247755  |
> > > > | `att_inverse` | improvement_ratio | 500   | 0.928378 | 0.933745 | 0.180881 | 0.219223   | 1.550033   |
> > > > | `att_inverse` | improvement_diff  | 500   | -1.764821| -1.293989| 3.749693 | -15.150490 | 8.948589   |
> > > >
> > > >
> > > >
> > > > **References for this reply**
> > > > [1] Xia Y, Mukherjee S, Xie Z, et al. From selection to generation: A survey of llm-based active learning[C]//Proceedings of the 63rd Annual Meeting of the Association for Computational Linguistics (Volume 1: Long Papers). 2025: 14552-14569.

---

> ### Author Response · Authors · 2025-11-20
> **## Reply for the third question (Part 1/5)**
>
> Your main concern is that, while you acknowledge the final performance gains of our method, you remain skeptical about *why* the proposed mechanisms are effective and do not yet fully see how they work in practice. In particular, you point out that the current draft lacks intuitive visual evidence of how Calibration Loss and Dynamic Deficit-Weighting reshape attention and semantic grounding, and explicitly suggest that “visualizations such as attention map changes” would make the effectiveness of these mechanisms clearer and more convincing. We fully agree with this assessment and have accordingly revised the paper.
>
> Concretely, in the revised version we will add direct **attention-map visualizations over the input images**, comparing the model’s attention before and after applying our method, as well as representative qualitative cases where the attention shifts from spurious regions to the true object areas and the answers are correspondingly corrected. These figures are intended to provide exactly the kind of intuitive, image-level evidence the reviewer requested.
>
> Beyond these qualitative visualizations, we also recognize that understanding *why* the mechanisms work benefits from more systematic analysis. To this end, we have added a series of complementary experiments that track how the model’s attention behavior changes under our training objective. First, we examine the **global shift from textual to visual attention** across training epochs. Second, we analyze the **layer-wise redistribution of visual attention** between shallow and mid/high layers. Third, we study the **Attention Focusing Ratio (AFR)** on datasets with ground-truth bounding boxes, showing how attention concentrates on true object regions. These analyses, presented below, are designed to augment the new attention-map visualizations and to make the effect of our mechanisms more transparent and convincing.
>
>
> ### (a) Global shift from textual to visual attention
> Following the reviewer’s suggestion, we compute how much attention the answer tokens allocate to image tokens versus text tokens from the prompts for the base model, all intermediate checkpoints saved during training, and the final model.
>
> From a global perspective, we do observe that, as training progresses, the model tends to rely more on visual tokens rather than purely textual tokens.
>
> Concretely, for the base model and the intermediate checkpoints after 1–4 epochs, we measure how much attention the answer tokens place on visual tokens vs. text tokens using two aggregated metrics (see Table R2 below and Figure 3 in the provided PDF (in supplementary materials)):
>
> 1. **Layer-wise mean ratio** (see Table R3):
>    For each layer, we compute an `image_sum_ratio` (the fraction of attention mass directed to image tokens), and then average this ratio across all layers.
>
> 2. **Global ratio across all layers**:
>    We sum the attention weights to image tokens and text tokens over **all** layers and heads, and compute
>
> $$
> \text{Global Image Ratio} = \frac{\text{image-sum}}{\text{image-sum} + \text{text-sum}}
> $$
>
> Both aggregation schemes show a consistent trend:
>
> - The **mean image ratio** increases from **0.1971 (base)** to **0.2055 (epoch 4)**, a relative gain of about **+4.3%**.
> - The **global ratio** `image_sum / (image_sum + text_sum)` increases from **0.2204** to **0.2291**, a relative gain of about **+4.0%**.
>
> This trend is consistent with the findings of PAI [1], which proposes a training-free method to alleviate hallucinations in LVLMs by explicitly rebalancing attention at inference time. Concretely, PAI identifies the “text inertia” phenomenon—LVLMs can produce similar outputs with or without the image—and counters it by (i) adaptively amplifying the attention weights assigned to image tokens, and (ii) subtracting the logits of pure-text predictions from those of multi-modal inputs. In this way, PAI directly boosts visual evidence and suppresses text-only biases without updating model parameters.
>
> In contrast, our method integrates this balance into the training process itself. We introduce a Calibration Loss and a Dynamic Deficit-Weighting term, which are *intended* to encourage the model to reallocate attention between image and text tokens in a data-driven way. We do not manually rescale attention matrices at inference time; instead, the attention mechanism keeps the same parametric form as in the original backbone, and the redistribution emerges from optimization over our training data.

---

> > ### Author Response · Authors · 2025-11-20
> > **## Reply for the third question (Part 2/5)**
> >
> > Regarding the magnitude of the change, the increase from 0.2204 to 0.2291 may look small in absolute terms (≈0.0087), but this metric is a global average over the [0, 1] range, aggregating attention distributions across all layers, all heads, and all answer tokens. Importantly, these statistics are computed over the entire training split of our preference dataset (more than 5,000 multimodal samples), rather than a few illustrative cases. A ≈4% relative shift at this scale therefore reflects a consistent redistribution of attention probability mass from text tokens to image tokens across the model, rather than a fluctuation on a small subset. Moreover, from the base model to epochs 1–4, this global ratio consistently stays above the base level, and its epoch-wise trend is aligned with the overall decrease in hallucination metrics reported in the main results table. Taken together, this suggests that the learned rebalancing of attention is a stable phenomenon that co-occurs with improved visual grounding, rather than incidental noise.
> >
> > **References**
> > [1] Liu S, Zheng K, Chen W. Paying more attention to image: A training-free method for alleviating hallucination in lvlms[C]//European Conference on Computer Vision. Cham: Springer Nature Switzerland, 2024: 125-140.
> >
> > ### (b) Layer-wise redistribution of visual attention (shallow vs. mid/high layers)
> >
> > To better understand how the attention pattern is reshaped, we analyze changes at the level of individual layers (see Figure 5 in the provided PDF (in supplementary materials)). We group the 32 Transformer layers into three segments and, for each layer, measure the attention ratio from answer tokens to visual tokens (see Figure 4 in the provided PDF (in supplementary materials)):
> >
> > - **Low layers**: 0–7
> > - **Mid layers**: 8–17
> > - **High layers**: 18–31
> >
> > The results show that the change is not a uniform “flat boost” of visual attention across layers, but a structured redistribution:
> >
> > - In the **low segment (layers 0–7)**, the average visual ratio increases only modestly (about **+1.5%** relatively), and in some shallow layers (e.g., layers 2–5), the visual attention ratio even slightly decreases.
> > - In the **mid segment (8–17)** and **high segment (18–31)**, the average visual ratio increases more strongly, by roughly **+5.5%** and **+6.1%** respectively.
> > - Several mid/high layers receive the most prominent gains, e.g.
> >   - layer 21: **0.2535 → 0.2710**,
> >   - layer 30: **0.1058 → 0.1283**,
> >   where the relative increase can reach **≈+21%**.
> > - For some upper layers such as layers 22, 27, and 30, the visual attention ratio exhibits a nearly monotonic increase from epoch 0 to epoch 4, while the shallow layers remain comparatively stable.
> >
> > Taken together, these observations suggest that our method does not simply amplify image attention everywhere. Instead, we observe a shift of attention budget from shallow layers to mid/high layers, which recent work identifies as key stages for visual–language fusion [2,3,4], indicating a layer-aware reallocation rather than a layer-agnostic scaling. Notably, this shift appears even though we do not apply any manual layer-specific rescaling during training.
> >
> > This empirical pattern is also compatible with how our training signal is constructed. Our Visually-Grounded Contrastive Preference Generation (VCPG) builds preference pairs through visual grounding rather than post-hoc textual editing, so the difference between $y^w$ and $y^l$ largely reflects how much visual evidence is used rather than purely textual paraphrasing (Sec.~3.1).
> >
> > In the same section, our analysis based on Visual Information Dependency (VID) and Fisher information shows that such visually grounded pairs are expected to increase the gradient norm
> > $$
> > \mathbb{E}\big[\|\Delta(\theta_1)\|\big]
> > $$
> > and concentrate optimization on the vision-dominant parameters $\theta_1$, providing a more targeted training signal for vision-related weights than typical vision-agnostic strategies. In other words, the data construction is designed to bias updates toward parameters that mediate visual processing, rather than purely language-only components. The fact that the strongest attention changes occur in mid/high layers, where visual and textual information are fused, is consistent with this theoretical prediction and suggests that the learned redistribution is plausibly linked to updates in the model’s visual pathway rather than purely text-only corrections.

---

> > > ### Author Response · Authors · 2025-11-20
> > > **## Reply for the third question (Part 3/5)**
> > >
> > > This layer-wise redistribution pattern is also consistent with recent analyses of LVLM internals.
> > > Jiang et al. (“Devils in Middle Layers of Large Vision-Language Models”) use an attention-lens metric to show that **middle layers are the critical stages where visual information is enriched and semantically refined**, and that modifying visual attention in these layers effectively mitigates object hallucinations [2]. Ye et al. (“CLAIM”) further find that **cross-modal attention divergence is most prominent in intermediate layers** in multilingual settings, highlighting their central role in multilingual object hallucination [3]. Li et al. (“Have the VLMs Lost Confidence?”) analyze the attention distribution of visual tokens and report that **the ability to resist sycophancy is predominantly associated with stronger high-layer visual attention**, and that enhancing image attention at higher layers helps mitigate this phenomenon [4].
> > >
> > > Overall, our layer-wise analysis follows the same qualitative trend: the visual-attention gain is concentrated in mid/high layers, while shallow layers remain relatively stable or even slightly reduced, indicating that the model allocates more attention budget to layers where semantic decisions and cross-modal alignment are known to play a central role, rather than uniformly amplifying image tokens across the network.
> > >
> > > **References**
> > > [2] Jiang Z, Chen J, Zhu B, et al. Devils in middle layers of large vision-language models: Interpreting, detecting and mitigating object hallucinations via attention lens[C]//Proceedings of the Computer Vision and Pattern Recognition Conference. 2025: 25004-25014.
> > > [3] Ye Z, Li Q, Feng X, et al. CLAIM: Mitigating Multilingual Object Hallucination in Large Vision-Language Models with Cross-Lingual Attention Intervention[J]. arXiv preprint arXiv:2506.11073, 2025.
> > > [4] Li S, Ji T, Fan X, et al. Have the VLMs Lost Confidence? A Study of Sycophancy in VLMs[C]//The Thirteenth International Conference on Learning Representations.
> > >
> > > **Table R2. Global image attention ratios across training epochs for LLaVA-1.5-7B.**
> > > | Checkpoint | Mean `image_sum_ratio` (over all samples) | Overall visual proportion `image_sum / (image_sum + text_sum)` |
> > > | ---------- | ----------------------------------------- | -------------------------------------------------------------- |
> > > | Base       | 0.197109                                  | 0.220413                                                       |
> > > | Epoch 1    | 0.205424                                  | 0.227467                                                       |
> > > | Epoch 2    | 0.206031                                  | 0.228646                                                       |
> > > | Epoch 3    | 0.205454                                  | 0.229081                                                       |
> > > | Epoch 4    | 0.205478                                  | 0.229134                                                       |

---

> > > > ### Author Response · Authors · 2025-11-20
> > > > **## Reply for the third question (Part 4/5)**
> > > >
> > > > **Table R3. Layer-wise image attention ratios across training epochs for LLaVA-1.5-7B.**
> > > > | Layer | Base     | Epoch 1  | Epoch 2  | Epoch 3  | Epoch 4  |
> > > > | ----- | -------- | -------- | -------- | -------- | -------- |
> > > > | 0     | 0.582635 | 0.593329 | 0.593449 | 0.593631 | 0.593519 |
> > > > | 1     | 0.366500 | 0.391864 | 0.393070 | 0.393238 | 0.392990 |
> > > > | 2     | 0.190149 | 0.189333 | 0.189212 | 0.189637 | 0.189796 |
> > > > | 3     | 0.196239 | 0.191296 | 0.190189 | 0.189791 | 0.189928 |
> > > > | 4     | 0.191965 | 0.187599 | 0.187344 | 0.187056 | 0.187289 |
> > > > | 5     | 0.203474 | 0.204195 | 0.202961 | 0.201720 | 0.201750 |
> > > > | 6     | 0.089091 | 0.090028 | 0.089853 | 0.089228 | 0.089272 |
> > > > | 7     | 0.084844 | 0.089449 | 0.089285 | 0.088600 | 0.088717 |
> > > > | 8     | 0.101540 | 0.110526 | 0.110479 | 0.109903 | 0.110045 |
> > > > | 9     | 0.101719 | 0.112581 | 0.112729 | 0.111575 | 0.111484 |
> > > > | 10    | 0.111041 | 0.118794 | 0.118870 | 0.118319 | 0.118108 |
> > > > | 11    | 0.159806 | 0.174256 | 0.174747 | 0.174087 | 0.173995 |
> > > > | 12    | 0.179874 | 0.188957 | 0.189014 | 0.188362 | 0.188550 |
> > > > | 13    | 0.129479 | 0.135149 | 0.135432 | 0.134765 | 0.135033 |
> > > > | 14    | 0.245604 | 0.253697 | 0.254489 | 0.254164 | 0.253848 |
> > > > | 15    | 0.126483 | 0.130899 | 0.131937 | 0.131362 | 0.131381 |
> > > > | 16    | 0.081543 | 0.083173 | 0.083282 | 0.082433 | 0.082661 |
> > > > | 17    | 0.181153 | 0.187317 | 0.190350 | 0.189816 | 0.190467 |
> > > > | 18    | 0.176871 | 0.181448 | 0.181708 | 0.180133 | 0.180290 |
> > > > | 19    | 0.211094 | 0.220113 | 0.222175 | 0.221381 | 0.221776 |
> > > > | 20    | 0.276292 | 0.285806 | 0.287980 | 0.287570 | 0.287934 |
> > > > | 21    | 0.253457 | 0.267468 | 0.270626 | 0.270038 | 0.271004 |
> > > > | 22    | 0.219176 | 0.227209 | 0.229050 | 0.229339 | 0.229935 |
> > > > | 23    | 0.220085 | 0.233589 | 0.235633 | 0.235134 | 0.235488 |
> > > > | 24    | 0.207460 | 0.218768 | 0.221832 | 0.220961 | 0.220983 |
> > > > | 25    | 0.225522 | 0.238937 | 0.239690 | 0.238254 | 0.238611 |
> > > > | 26    | 0.268594 | 0.277129 | 0.275000 | 0.274295 | 0.273532 |
> > > > | 27    | 0.092683 | 0.100136 | 0.101913 | 0.102185 | 0.102223 |
> > > > | 28    | 0.186419 | 0.198663 | 0.199423 | 0.199156 | 0.199256 |
> > > > | 29    | 0.196246 | 0.206421 | 0.206900 | 0.206131 | 0.205888 |
> > > > | 30    | 0.105821 | 0.119179 | 0.126292 | 0.127698 | 0.128254 |
> > > > | 31    | 0.190546 | 0.203599 | 0.206782 | 0.206987 | 0.206594 |
> > > >
> > > >
> > > >
> > > > ### (c) Attention Focusing Ratio (AFR): focusing on key object regions
> > > >
> > > > To further verify that the model is not only “looking more at the image” but also focusing more on the **correct regions**, we revisit the **Attention Focusing Ratio (AFR)** analysis introduced in Sec.5.2 of the paper. We use TextVQA, which provides ground-truth bounding boxes, and for each layer $l$ we define $\text{AFR}_l$ as the ratio between the mean attention that answer tokens assign to visual tokens **inside** the ground-truth answer box and the mean attention they assign to visual tokens **outside** that box. A higher AFR therefore indicates that attention is more tightly concentrated on the true object region rather than on background or irrelevant areas; the full protocol and motivation are detailed in Sec.5.2.
> > > >
> > > > In the main paper, we reported AFR only for the base model and the final model to validate mitigation of the Perceptual Bottleneck. Now, we extend that analysis by additionally including the intermediate checkpoints at epochs 1–3, allowing us to track how AFR evolves over the whole training trajectory rather than only comparing two endpoints. As shown below, AFR steadily improves from the base model through epochs 1–4, especially in mid/high layers, indicating that the model progressively shifts attention toward ground-truth regions instead of merely increasing undirected visual attention.
> > > >
> > > > We compute AFR for all 32 layers and compare the base model with the checkpoints during training:
> > > >
> > > >
> > > > - The **average AFR over all layers** increases from **5.54 (base)** to **5.64 (epoch 4)**, a relative gain of about **+1.8%**, indicating a moderate but consistent overall strengthening of attention to key object regions.
> > > > - When we group layers into low / mid / high segments, the average AFR changes are:
> > > >   - **Low**: 0.75 → 0.79 → 0.79 → 0.78 → 0.78 (absolute change ≈ **+0.03**)
> > > >   - **Mid**: 7.05 → 7.03 → 7.10 → 7.12 → 7.16 (absolute change ≈ **+0.11**)
> > > >   - **High**: 7.20 → 7.30 → 7.32 → 7.32 → 7.33 (absolute change ≈ **+0.13**),
> > > >   showing that the largest absolute gains appear in the mid/high segments, while the low segment fluctuates slightly around its base value.

---

> > > > > ### Author Response · Authors · 2025-11-20
> > > > > **## Reply for the third question (Part 5/5)**
> > > > >
> > > > > - Importantly, the overall shape of the AFR curve over layers (i.e., which layers act as peaks and valleys) remains almost unchanged (Figure 6). For the LLaVA architecture, certain “visual-semantic layers” (e.g., layers 10–15, especially layer 14) already had high AFR values in the base model. Under our method, these layers remain the AFR peaks, but their peak values are further amplified. For example,
> > > > >   - **Layer 14**: AFR increases from **14.74 → 17.10 → 18.02 → 18.44 → 18.72**, a relative gain of about **+27%**. Notably, layer 14 already exhibits the highest AFR in the base model and acts as one of the “visual–semantic” peak layers in LLaVA; under our method, this existing peak is further sharpened rather than shifted, meaning that the layer that was originally most responsible for focusing on core object regions becomes even more concentrated on ground-truth boxes and less on background noise during answer generation.
> > > > >
> > > > >
> > > > > These layer-wise statistics and AFR analyses together show that our method preserves the original hierarchical division of labor across layers, while gently but directionally reallocating attention from shallower layers and background regions toward mid/high layers and ground-truth object regions. Rather than collapsing the architecture into an “always-look-at-image” regime, the model sharpens the layers that were already responsible for semantic visual grounding.
> > > > >
> > > > > From the training perspective, this behavior is also compatible with how our **Calibration Direct Preference Optimization objective** is designed to shape the updates. In particular, the Focus-and-Enhance preference pairs and the composite focus loss
> > > > >
> > > > > $$
> > > > > \mathcal{L}_{\text{focus}} = \mathcal{L}_{\text{dpo\_focus}} + \lambda_{\text{calib}} \mathcal{L}_{\text{Calib}}
> > > > > $$
> > > > >
> > > > > explicitly prefer responses whose probability increases when the image is enhanced or cropped to the key region (Sec.~3.2). As discussed in our VID-based analysis, this calibration term encourages the model to attribute preference differences to the visual intervention, which in turn tends to increase the gradient signal on vision-dominant parameters and bias updates toward weights that mediate visual processing rather than purely language-only components.
> > > > >
> > > > >
> > > > > Taken together, the observed redistribution of attention toward mid/high layers and ground-truth object regions is consistent with this optimization picture: the calibration loss encourages the model to rely more on informative visual cues in its on-policy preferences, and the layer-wise/AFR statistics provide complementary evidence that this pressure is reflected in stronger, more focused visual grounding in the internal attention patterns. These changes are also in line with the reductions in hallucination reported in the main results.
> > > > >
> > > > >
> > > > > ### Qualitative case studies
> > > > >
> > > > > To illustrate how our method reshapes the model’s attention, we provide qualitative case studies that visualize attention patterns before and after training. Specifically, for each representative example, we show (i) the initial attention maps of the base model, (ii) the guided crop region produced by our attention-based visual intervention, and (iii) the final attention maps after training with our method, where attention becomes more concentrated on the truly relevant visual evidence. Please refer to the supplementary PDF (in supplementary materials) for these visualizations and detailed commentary.
> > > > >
> > > > > Concretely, in the supplementary PDF (in supplementary materials) we include image-level examples such as:
> > > > >
> > > > > - **Original attention heat map**: the base model’s attention is relatively diffuse, with weak activations scattered over multiple irrelevant regions of the image.
> > > > > - **Crop box**: our attention-guided cropping algorithm identifies a salient region (e.g., the area containing the potato chips in the example “*What kind of potato chips are on the plate?*”), which is then used as a visual condition to elicit a more grounded response distribution.
> > > > > - **Attention heat map after training**: after training with Calibration Loss and Dynamic Deficit-Weighting, the attention mass shifts and becomes sharply concentrated on the true object region, while spurious regions receive much less attention.
> > > > >
> > > > > These cases visually demonstrate the intended behavior of our method: the crop-based condition first helps the model simulate a “strong-vision” state focused on the correct region, and training then aligns the default attention pattern towards this visually grounded distribution. For further implementation details on how attention maps are computed and rendered, please refer to Appendix B of the main paper.

---

> ### Comment · Reviewer_m9BD · 2025-11-22
> **Responses to authors' rebuttal**
>
> For question 1, thanks for providing additional results.
>
> For question 2, while the authors propose several bias mitigation mechanisms, such as strong/weak-vision contrastive design, CLIPScore-based dynamic weighting, and PPL+Margin filtering, the response lacks direct evidence that these methods actually reduce early-stage hallucination or self-reinforcement bias accumulation. The presented results demonstrate plausibility but do not establish a clear causal relationship between the proposed mechanisms and bias suppression.
>
> Moreover, although CLIPScore is used to measure semantic consistency, it may not reliably reflect hallucination when the model outputs short or low-descriptive responses. Incorporating such an unstable metric directly into loss weighting could introduce additional noise rather than consistently mitigating bias.
>
> For question 3, thanks for providing the additional analysis and the qualitative results are still needed besides these analysis.
>
> Therefore, I maintain the original scores.

---

> > ### Author Response · Authors · 2025-11-23
> > **Clarifying some misunderstandings**
> >
> > Thank you for your reply！I apologize for the excessive amount of information I provided earlier. This might have affected your reading and caused some misunderstandings.
> >
> > ## **For question2**
> > We believe there is an important misunderstanding: it may look as if we first let the model freely generate some outputs, then apply a few filtering rules to filter out hallucinations.
> >
> > In fact, our pipeline is designed to bias the generated data toward a **low-hallucination** distribution already at the generation stage, while filtering and weighting serve only as additional safeguards rather than a post-hoc attempt to “wipe out hallucination” from arbitrary samples.
> >
> > ### **Generated data**
> > (1) Focus-and-Enhance preference pairs.
> > Our Focus-and-Enhance preference pairs are built directly on the mechanism described in Sec.~4.1.1. We first use answer-conditioned attention maps of the reference model to extract a salient crop from the image, and then construct an enhanced view  $I_{\text{aug}}$(original image plus crop) and a degraded view $I_{\text{deg}}$ (erasing the crop region). Prior training-free work on attention-guided cropping has shown that supplementing the input with such salient regions can mitigate perceptual hallucinations in LVLMs (Zhang et al., 2025, *MLLMs Know Where to Look*). In our framework, this insight is turned into preference supervision: for the same question, the response from the enhanced view  $(I_{\text{aug}}, P_{\text{enh}})$ is treated as the winner and the response from the degraded view  $(I_{\text{deg}}, P)$  as the loser, explicitly biasing the learned policy toward the low-hallucination, visually clarified state.
> >
> >
> > (2) Robust preference pairs from strong/weak-vision contrast.
> > We construct weak-vision inputs by adding Gaussian noise to the original image and treat the model’s output under this degraded view as the losing response. For the strong-vision side, we apply visual contrastive decoding between the clean-image model (expert) and the noisy-image model (amateur), a decoding scheme that has already been shown to effectively reduce object hallucinations in LVLMs at inference time (Leng et al., 2024, *Mitigating Object Hallucinations in LVLMs through Visual Contrastive Decoding*). In our setting, the contrastively decoded strong-vision output is always chosen as the winner, so the resulting preference distribution is explicitly biased toward low-hallucination, visually grounded answers rather than arbitrary self-generated text.
> >
> > ### **PPL+Margin filtering.**
> > In an additional experiment, we use a commercial model to compare human-labeled answers with our generated answers, filter out low-quality samples, and then measure the overlap with our PPL+Margin-based filtering (additional experiment). The large overlap shows that PPL+Margin reliably detects erroneous responses (see Construction of an evaluation task with human references in Reply for the second question (Part 2/4)). By construction, once a sample is filtered out, it no longer serves as training supervision, so these erroneous or hallucinatory answers cannot further contribute to self-reinforcement bias.
> >
> >
> > ### **Use of CLIPScore**
> > In our method, CLIPScore is computed only between the question and the original / cropped images to assess crop quality; it is **never computed on the generated answers**. We use it solely to down-weight samples in the focus branch when the crop is likely incorrect, while the robust (global strong/weak-vision) branch remains untouched, and CLIPScore is not a training target. Intuitively, the two types of preference pairs are complementary—focus pairs are more suitable for local, region-specific questions, whereas robust pairs are more suitable for global questions—and CLIPScore simply provides a principled way to adjust their relative influence depending on whether the crop truly improves image–text alignment. As shown in our **additional experiment** (Reply for the second question (Part 4/4), Table 1), CLIPScore reliably decreases under intentionally wrong crops, so it serves as a targeted safety signal for bad crops rather than an unstable source of noise.
> >
> > ## **For question 3**
> > Regarding the attention map changes you previously requested, we have already included these visualizations in the updated supplementary material. If you have reviewed them and still find them unsatisfactory, we would sincerely appreciate it if you could indicate the specific types of qualitative evidence or cases you are most interested in, and we will do our best to provide them.

---

### Meta-Review · Area_Chair_NkLv · 2026-01-11

**Summary:**

The reviewers expresses several concerns, which are listed in the following.

1. limited experiments. Missing baselines. [Reviewer m9BD, Reviewer cpy5]. The experiments are primarily conducted on LLaVA-1.5-7B and Qwen-2.5VL-3B. Missing qualitative interpretation of the experiments. [Reviewer m9BD], Missing comparisons with recent work [Reviewer cpy5], Missing computational complexity analysis. [Reviewer LaWn]

2. The limitation of the proposed method. Potential bias in on-policy data [Reviewer m9BD], the effect of quality of attention maps and cropping heuristics [Reviewer LaWn].

After reading the reviewers' comments as well as the authors' rebuttal, the concerns on the experiments, comparing with existing work and baselines such Qwen-vl-7B, has been addressed. The quality interpretation of the work is also provided. The limitations of the work are discussed, although some information is not clearly provided and addressed. Most of the comments have been addressed and the paper does have merits. As such, I am suggestion acceptance of the paper.

**Reviewer Concerns:**

The experiments on Qwen-VL-7B have been conducted.

**Reviewer Scores:**

I think that the reviewers may maintain their previous rating.

---

### Decision · Program_Chairs · 2026-01-26

Accept (Poster)